# Temporal, spatial and gender-based dietary differences in middle period San Pedro de Atacama, Chile: A model-based approach

William J. Pestle[1]*, Mark Hubbe[2,3], Christina Torres-Rouff[3,4], Gonzalo Pimentel[5,6]

1 Department of Anthropology, University of Miami, Coral Gables, Florida, United States of America,
2 Department of Anthropology, The Ohio State University, Columbus, Ohio, United States of America,
3 Instituto de Arqueología y Antropología, Universidad Católica del Norte, San Pedro de Atacama, Chile,
4 Department of Anthropology and Heritage Studies, University of California Merced, Merced, California,
United States of America, 5 Fundación Patrimonio Desierto de Atacama, San Pedro de Atacama, Chile,
6 Universidad de Tarapacá, Arica, Chile

* w.pestle@miami.edu

**Data Availability Statement:** All relevant data are within the paper and its Supporting Information files.

## Abstract

To explore the possible emergence and lived consequences of social inequality in the Atacama, we analyzed a large set (n = 288) of incredibly well preserved and contextualized human skeletons from the broad Middle Period (AD 500–1000) of the San Pedro de Atacama (Chile) oases. In this work, we explore model-based paleodietary reconstruction of the results of stable isotope analysis of human bone collagen and hydroxyapatite. The results of this modeling are used to explore local phenomena, the nature of the Middle Period, and the interaction between local situations and the larger world in which the oases were enmeshed by identifying the temporal, spatial, and biocultural correlates and dimensions of dietary difference. Our analyses revealed that: 1) over the 600-year period represented by our sample, there were significant changes in consumption patterns that may evince broad diachronic changes in the structure of Atacameño society, and 2) at/near 600 calAD, there was a possible episode of social discontinuity that manifested in significant changes in consumption practices. Additionally, while there were some differences in the level of internal dietary variability among the *ayllus*, once time was fully considered, none of the *ayllus* stood out for having a more (or less) clearly internally differentiated cuisine. Finally, sex does not appear to have been a particularly salient driver of observed dietary differences here. While we do not see any *de facto* evidence for complete dietary differentiation (as there is always overlap in consumption among individuals, *ayllus*, and time periods, and as isotopic analysis is not capable of pinpointing different foods items or preparations), there are broad aspects of dietary composition changing over time that are potentially linked to status, and foreignness. Ultimately, these stand as the clearest example of what has been termed "gastro-politics," potentially tied to the emergence of social inequality in the San Pedro oases.

**Funding:** WJP received National Science Foundation Grant BCS-1358753, https://www.nsf.gov/. CTR, MH, and GP received National Science Foundation, Grant BCS-1359644, https://www.nsf.gov/. CTR and MH received Vicerrectoría de Investigación y Desarrollo Tecnológico of the Universidad Católica del Norte Grant 22/2011, https://www.ucn.cl/sobre-ucn/vicerrectorias/vicerrectoria-de-investigacion-y-desarrollo-tecnologico/. The funders had no role in study design, data collection and analysis, decision to publish, or preparation of the manuscript.

**Competing interests:** The authors have declared that no competing interests exist.

## Introduction

San Pedro de Atacama, Chile, lies at the northern end of the Atacama salt flat, where the San Pedro and Vilama Rivers supply a series of small oases in the otherwise inhospitable expanse of the Atacama Desert, the driest non-polar desert in the world (Fig 1). These oases, occupied since at least the Formative Period (ca. 1200 BC), saw a florescence of human activity in the Middle Period (AD 500–1000), a time that is often described as one of unparalleled regional prosperity and peace. Many of this period's benefits have been ascribed to the incorporation of local societies within the sphere of influence of Tiwanaku—an expansive polity that arose in modern-day Bolivia—and the growing role of Atacama societies in a system of regional exchange and mobility. However, such a broad telling of the Middle Horizon likely masks an unequal distribution of the benefits of the age and could similarly obscure social inequality within and among the different Atacama oases. Indeed inequality has been noted to be particularly pronounced in those instances where aspirant leaders are acquiring power and access to the exotic is increasing [1–8], as seems to have been the case during the Andean Middle Period.

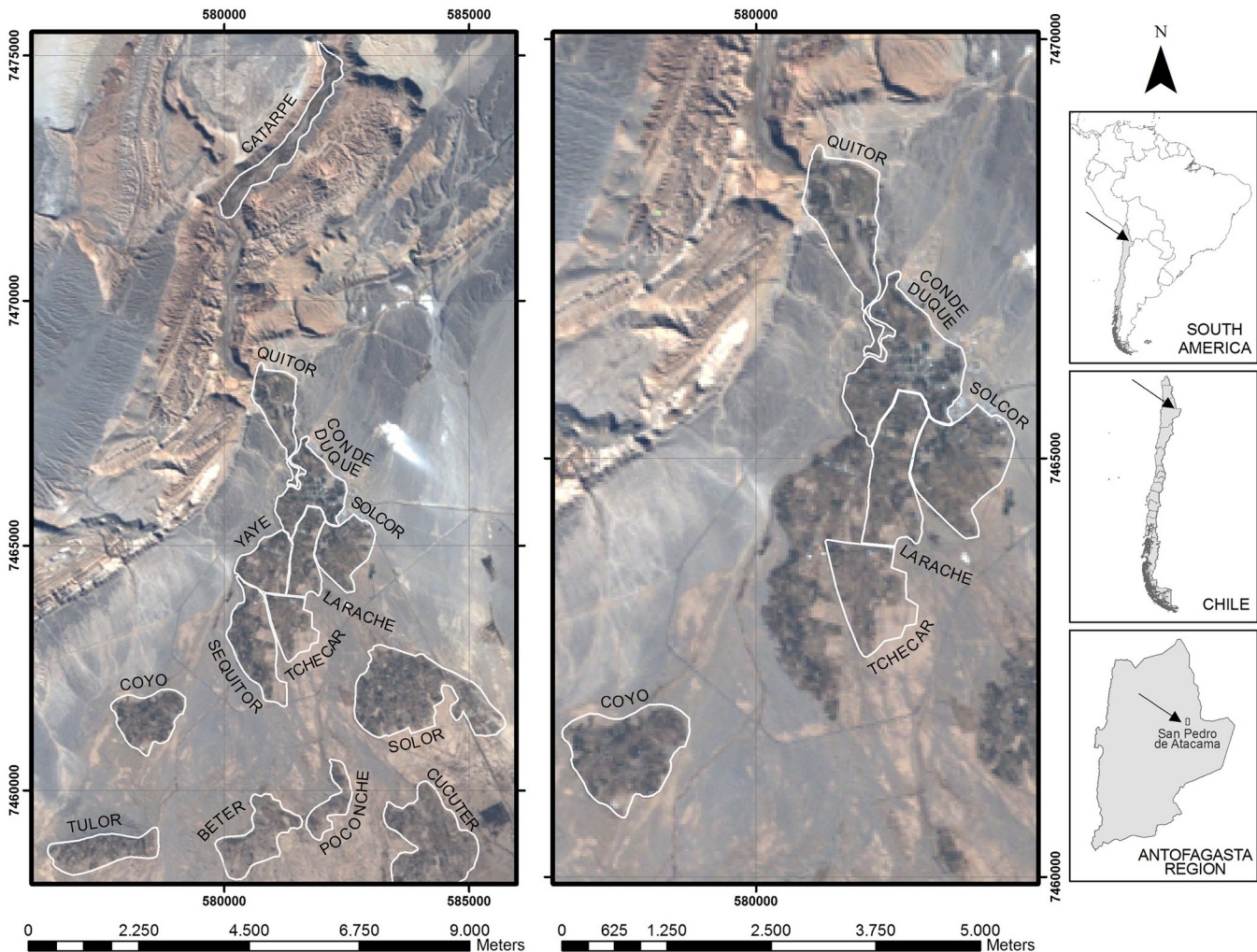

**Fig 1. Map of San Pedro de Atacama, with locations referenced in texts noted.**

To explore the possible emergence and lived consequences of social inequality in the Atacama, we recently completed intensive bioarchaeological and biogeochemical analysis of a large set (over 600 individuals) of incredibly well preserved and contextualized human skeletons recovered from a series of cemeteries dated to the Middle Period of the San Pedro oases. In the present work, we explore one subset of our data in detail, the model-based paleodietary reconstruction accomplished through stable isotope analysis of human bone collagen and hydroxyapatite. We use this to explore local phenomena as well as the nature of the Middle Period, stressing the importance of the indigenous responses, as well as the interaction between local situations and the larger world in which the oases were enmeshed. Given that the distribution of, and access to, varied kinds or quantities of food is a common mechanism and/or consequence of intra-societal differentiation [9, 10], knowledge of how patterns of consumption may have differed over time, among oases, and between various subsets of the population under study (e.g., males and females) offers unique insights into past processes of inequality. Combining these data with a rigorous program of radiocarbon dating and detailed analysis of grave contents, among other sources of information, has the potential to reveal how resource access may have varied along multiple axes of difference in the oases. Importantly, we argue that this integrative approach comes with a precision unobtainable by other archaeological means and allows us to explore in far greater detail the complex shifts occurring in local prehistory.

## Geographical and archaeological background

The Atacama Desert comprises over 100,000 $km^2$ of northern Chile and southern Peru between ca. 18˚ and 30˚ South latitude. The Pacific Ocean and the Andes, which flank the Atacama on the west and east, respectively, both contribute to the desert's hyperarid conditions. Among the world's deserts, the Atacama receives a classification of "Ea23" [11], denoting extreme aridity, a lack of seasonal precipitation, and temperatures between 10˚ and 30˚ C. Instrumental data indicate 1–6 mm/yr of rainfall at weather stations across the region [12], multi-year studies are consistent with measurable rain only on a decadal scale [13], and the region's aridity seems to have been a persistent feature for millennia [14], if not millions of years [15, 16]. Indeed, a recent review of paleoclimatic conditions in the region suggests that aridity equal to, if not exceeding, that seen in the present prevailed throughout the Holocene [17]. The exception to this hyperaridity was the existence of a somewhat wetter time between 1000–2000 years ago, encompassing the period examined here. As discussed below, this may have facilitated the spread of agriculture and the permanent settlements that characterize the Middle Period.

Within this arid landscape, the oases of San Pedro de Atacama offer access to stable conditions for the long-term settlement of humans and have sustained agropastoral societies for millennia. The oases are located at the northern edge of the Atacama salt flat, in the alluvial delta formed by the San Pedro and Vilama Rivers, before they drain into the subsurface water table that forms the salt flat. This flatter region concentrates humidity that allows the development of natural oases which have been managed and even expanded by humans in the past. The oases have been continuously occupied for over 4000 years as populations moved down from the surrounding *Puna* (the cold arid plateaus of the high Andes; [18–22]). With the adoption of horticulture as well as arboriculture and silvopastoralism in the oases, populations aggregated into larger settlements, taking advantage of the rivers and arable land [20, 23, 24]. From the Formative Period (~1200 BC) onwards, there is strong evidence of aggregated settlements, large and small habitation sites, as well as substantial cemeteries that document the long-term occupation of the oases (e.g., [20, 24–26]).

Settlement pattern analyses have revealed a substantive concentration of sites, both cemetery and habitation, in the central and northern oases during and just before the Middle Period (AD 500–1000) [20, 27]. Agüero [27] argues that this demonstrates the considerable growth and ultimate stability of local villages during the Middle Period. This time also witnessed the widespread integration of the Atacama oases into the interregional networks of the southern Andes. Nuʹñez [21] has argued that this time also sees the earliest evidence for the production of surplus, a phenomenon that facilitates the growing caravan trade and is a key factor in the rise of persistent inequality. This is bolstered by a surge in population as well as incipient craft specialization, evidenced in new metallurgy practices, the growth of agriculture, and individuals who focused their energies on the caravan trade [21, 28–32].

In the San Pedro de Atacama oases, this rise in complexity and affluence appears to have manifested in different fashions across the oases and *ayllus*. The *ayllu* is the traditional Andean form of kin-based social organization that can also denote territorial boundaries (ascriptive descent groups [33–35]). Interestingly, Goldstein [36] has argued for the *ayllu* as an ethnic community that, while based on kin, is bounded territorially by *huacas* or ancestral shrines. For example, in 1642, an idol ("ydolo") or huaca was identified as belonging to Sequitur (today's ayllu of Sequitor); it consisted of a black stone that was worshipped by the ayllu's residents and was hidden in a type of vault. Similarly, in the ayllu of Contituques Cantal Acapana (today's Conde Duque or San Pedro) was an idol called Tocol, which was buried in the courtyard of the ayllu's church, having been previously burned in the public square [35]. Locally, *ayllus* have typically been interpreted as naturally differentiated oases (e.g., Solor, Tulor, or Coyo) as well as the divisions of the major oasis into internal sectors marked by territorial features and contemporary boundaries (Fig 1).

## Food and intra-societal differences

Anthropologists have expended countless pages on the ways in which dietary differences can mirror, reinforce, or even establish social inequalities. Arnold [37] asserts that "food was, and continues to be, power in a most basic, tangible and inescapable form." The collusion of the basic sustaining power of food, its necessity for life, and the demands of systems of social inequality, means that "variation in what people eat reflects substantive variation in status and power and characterizes societies that are internally stratified into rich and poor, sick and healthy, developed and underdeveloped, overfed and undernourished" [38]. In this way, the consumption of, or access to, particular foodstuffs has come to mark membership in empowered or subjugated intra-societal groups, and/or as an indicator of internal social schisms.

In spite of the inherent tendency of food to homogenize difference and bring people closer together [9], many stratified societies have developed means by which this innate homogenizing effect of food is overridden through a process of semantic inversion and serves as a tool that, in Appadurai's words, can "serve to regulate rank, reify roles, and signify privileges," and which can, "sustain relations characterized by rank, distance, or segmentation," [9]. Goody [10] maintained that differential control of, and access to, food can play a central role in the creation and maintenance of multiple intra- and inter-societal hierarchies. While food may have a central social role in even the least stratified society, increasing social stratification offers the opportunity (and perhaps establishes the demand) for even more intense uses of food in the negotiation of relations of power [39]. In such societies, the power of food as a symbol or code for status differentiation can be fully exploited as a means of establishing and furthering inequality. In, perhaps, the purest expression of the furtherance of social inequality through food, elite individuals will sometimes go so far as to effectively monopolize certain foodstuffs, especially if that food is already an established symbol of their position in society, for example,

the supposedly exclusive consumption of freshwater fish and wildfowl by the nobility of 15th century England [40]. This monopolization gives rise to an internally differentiated cuisine, a food system in which not all people have equal access to all foodstuffs, as a consequence of differences in social status [10]. The specifics of the consumption of restricted foods: who can, and cannot, have access, when, how much, and under what circumstances, are the means by which elite individuals can attempt to build and maintain their higher status position within a given society. It is this use of food, as de-facto ammunition in internal conflicts over cultural, economic, or political resources, that Appadurai aptly labels "gastro-politics" [9].

That not all societies stratify their food is undoubtedly a reflection of the fact that there are many possible paths by which a society may become stratified [2], and that it is not obligatory that dietary practices follow broader patterns of social stratification, especially in societies without formal stratification. Nevertheless, it is quite often the case that stratified societies do possess some form of differentiated cuisine and the development of this differentiation seems often to be coeval with the development of early stratified societies [41].

## Research questions

Building on this exploration of social differentiation and diet, our research into the paleodietary practices of the Middle Period inhabitants of San Pedro de Atacama was guided by a series of questions that sought to identify the temporal, spatial, and biocultural correlates and dimensions of dietary difference.

**Time.** As the individuals included in this analysis represent nearly a millennium of the occupation of the San Pedro oases, our initial interrogation of the dietary data was diachronic in nature. Specifically, using the corpus of individuals for whom we possess both isotopic and radiometric data (n = 167), we sought to determine if there were any coherent temporal trends (correlations) in the consumption of any of the modeled food groupings that might have resulted from diachronic changes in resource availability/access. Such changes in access or consumption may, in turn, be connected to the establishment, institutionalization, or intensification of social inequality in the San Pedro oases over time.

Furthermore, in the temporal realm, we sought to test the dietary dimensions of a hypothesis advanced in Pestle and colleagues [42], which proposes a disjuncture between what we term the early/incipient (pre-600 calAD) and late/established (post-600 calAD) Middle Period. As above, the large corpus of dated individuals allows for a specific examination of whether there existed dietary correlates of this proposed inflection point in San Pedro's history.

**Space.** In our previous examination of the chronology of the six *ayllus* under consideration here [42], we noted the apparent independence of the use-life patterns of different *ayllus*. Based on this, we contend that the dynamics of human activity in each *ayllu* would appear to have been particularly sensitive to local socio-cultural and environmental dynamics, a conclusion that reinforces previous archaeological and bioarchaeological findings of significant differences among *ayllus* during the Middle Period [43–46]. Here, we attempt to quantify any differences in dietary practices among the *ayllus* through comparison of modeled dietary contributions. This line of inquiry flows from the notion that differences in access to different foodstuffs among *ayllus* could be a consequence of different positioning vis-à-vis the inter-regional exchange systems of the period, or indeed of differences in positions in the status hierarchy of the *ayllus*. As the different *ayllus* sometimes belong to different portions of the Middle Period, we augmented our comparisons of the "raw" modeled foodstuff proportions with a second comparison of the residuals of modeled contributions after regression against calibrated radiocarbon dates (derived from the Inter-*Ayllu* date model of [42]). These time-

corrected dietary differences should allow us greater insights into the mechanics of life in the various *ayllus* after the variable of time is fully accounted for.

**Sex.** Finally, working from the complementary theories that: 1) differences in diet are often associated with, or pattern on, sex and understandings of gender, and 2) that inequality (broadly construed) is experienced differently by men and women, we next examined the relationship between biological sex (as determined osteologically) and diet. First, at the coarsest level, we compared modeled diet between all males and females in the study sample (grouping all individuals from all *ayllus* together). Next, in light of the aforementioned differences among the *ayllus*, we compared male and female diet from each of the four *ayllus* for which we possessed sufficiently large samples of sexed individuals. This comparison was made on both raw modeled foodstuff estimates and with the date-regressed residuals of foodstuff values to control for time (see above). Finally, in an attempt to integrate considerations of both social/ spatial (*ayllu*-based) and sex-based dimensions of dietary difference simultaneously, we compared the modeled diets of males and females from each of the four best-represented *ayllus* with each other. As above, this comparison was made on raw modeled foodstuff estimates and with the date-regressed residuals of foodstuff values.

The three vectors of difference explored here are not meant to be an exhaustive list of the possible ways that inequality may have been partitioned in the San Pedro societies. However, they are the ones where the information available in the bioarchaeological record allow us to explore the differentiation on this spatial scale with a high degree of confidence. For instance, another important vector of social differentiation in many human societies is age, but this is an aspect that we are unable to explore with the current available skeletal record, since 1) the study sample included less than 6% subadult individuals, and 2) most of the individuals included in our study were represented by cranial remains alone, rendering precise age determinations too unreliable to be useful in statistical models.

Of relevance to the three aforementioned vectors of difference, in the present work, we examine diet as a proxy for underlying social, cultural, and political dynamics, remaining conscious that the observed changes in consumption are likely reflections of changes within society. Put differently, when confronted with diachronic changes in consumption, we are aware that time itself is not changing diet, but rather that something integral to the arrangement of Atacameño society is changing with time, which, in turn, is affecting diet. While such proxy evidence can be challenging given that it is not a direct measure of the phenomenon of interest (social change), the accumulated decades of anthropological work on the linkages between consumption and social structure add support to the validity of our investigation along these three lines.

## Isotopic basics

Stable isotope analysis remains the best means available to archaeologists for the estimation of the diet of ancient individuals. Indeed, nearly fifty years of the application of this method in archaeology has built a corpus of experimentally validated methods that permit accurate estimation of broad aspects of human paleodiet (see [47, 48] for general reviews of the method). Assuming sufficient preservation of target biomolecules (bone collagen and hydroxyapatite) and accurate knowledge of the local foodweb and fractionation (the offsets between, for instance, consumed foods and consumer tissues), stable isotope analysis can be used to estimate broad aspects of individual paleodiet ($C_3$ vs. $C_4$ plants, marine vs. terrestrial protein) with a high degree of accuracy. The basics of this method have remained effectively unaltered since the earliest archaeological applications of the technique [49, 50], and the real power of the method remains in the scalability of the data it generates, which permits inference of

everything from individual diets to the consumption practices of broad subsets of a past society. Given the incredible preservation of skeletal biomolecules in the Atacama Desert, and the *relative* simplicity of the local foodweb (which lacks marine or freshwater foods), stable isotope analysis is particularly well suited for paleodietary reconstruction in the region.

Classical applications of stable isotope analysis to archaeological questions of diet have tended to proceed as follows. Collagen (and sometimes hydroxyapatite) are extracted from human bone samples, and mass spectrometry is used to generate estimates of stable isotope ratios (expressed using the delta (δ) notation, a ratio of heavier to lighter isotope relative to an international standard) of carbon ($^{13}C/^{12}C$) and nitrogen ($^{15}N/^{14}N$) for the extracted biomolecules. These consumer values are then compared, typically on an *ad hoc* basis, with measured or stipulated values of potential food groups as to generate bounding (more than/less than), ordering, or rough proportional estimates of past consumption. Only rarely are the effects of fractionation, disparate macronutrient composition of different foodstuffs, and the effects of differential routing of different food fractions to different consumer tissues accounted for systematically and sufficiently.

Fortunately, recent decades have witnessed the development of powerful new tools for the interpretation of isotopic data, in particular mixing models, which permits probabilistic quantification of source contribution. Such models greatly improved the explanatory power of isotopic estimation of paleodiet because these approaches use a complex suite of user-stipulated inputs to model bounded source contributions to a given consumer's diet, rather than relying on basic visual comparisons of plotted source and consumer data, or making simplistic assumptions (for example, that hydroxyapatite isotope values reflect plant diet). While the earliest mixing models [51, 52] could only solve for a limited number (*n*+1) of food sources, later developments [53–55] expanded calculations to non-determined systems, and provided means for dealing with uncertainty and incorporating priors based on a Bayesian approach.

Most recently, models such as FRUITS [56] have been built to accommodate the input of consumer, foodweb, macronutrient composition, and routing parameters, as well as user-specified priors, to determine, probabilistically, bounded source contributions while accounting for uncertainty in all input data. These recent Bayesian approaches "offer a powerful means to interpret data because they can incorporate prior information, integrate across sources of uncertainty and explicitly compare the strength of support for competing models or parameter values" [53]. The substantial potential of these methods for addressing questions of significant paleodietary interest has consequently resulted in their increased use in the South-Central Andes (e.g., [57–62]).

## Methods and materials

### Skeletal collections

The skeletal sample included in this study represent individuals excavated from 13 cemeteries dated to the Middle Period. The sites were excavated over several decades by different researchers, including the amateur archaeologist Father Gustavo LePaige, who recovered most of the skeletal collections in the region. A detailed review of the history of research and the preservation status of the collections can be found in [24, 63]. The chronological contextualization of the cemeteries has been detailed in [42].

The accumulated body of archaeological work on the region suggests that the cemeteries were not precincts reserved for specific classes of individuals, as there is not systematic bias in sex representation in the sites [63], and as individuals of all ages are found in the burial spaces [28, 64]. Therefore, they are assumed to be representative of local societies. However, the different archaeological recovery practices employed over time have resulted in a skeletal

collection that is very biased towards adult individuals (which is reflected in our data), and most of the collections are represented only by human skulls, as LePaige, in particular, did not save postcranial remains from his excavations [63]. Moreover, during the military government in Chile, the skeletal collections were separated from their funerary context, resulting in the loss of individual context information for most of the collections excavated by LePaige. Therefore, detailed archaeological information and complete skeletons are available only in cemeteries excavated more recently, such as Solcor 3, Coyo 3, Quitor 6 Tardío, and Casa Parroquial. For these reasons, the analysis of vectors of inequality within cemeteries is not possible across all of our samples.

Demographic characteristics of the individuals included in this study (age at death and sex) were estimated based on the analysis of skulls and pelvis, following traditional bioarchaeological methods of analysis [65]. However, as mentioned before, for most of the samples only skulls are available, which makes the estimates of age at death unreliable [66, 67]. For this reason, the age distribution of the local populations cannot be estimated properly, and this demographic aspect was not explored in the current study.

The cemeteries were grouped inside *ayllus*, as they represent long-standing geo-political units in the Atacama oases (see above), far predating the colonial period. Cemeteries are included in each *Ayllu* based on their geographic location within their boundaries (Fig 1), given that these boundaries have been stable since at least the beginning of colonial period [68], if not far earlier. In total the cemeteries represent the human presence of six *ayllus*, as detailed in Table 1 and Fig 1.

**Sampling.** Sampling for isotopic analysis took place in the Museo Gustavo Le Paige in San Pedro de Atacama, Chile. We selected samples from a total of 288 individuals representing six *ayllus* (Conde Duque, Coyo, Larache, Quitor, Solcor, and Tchecar). The selection criteria for the individuals attempted to generate representative samples from each the cemeteries and the sexes, while at the same time minimizing the damage to complete bones. For this reason, whenever possible, samples were collected from fragmented skeletal remains. The sample composition is heavily biased towards adults, as subadults are not common in the collections excavated by LePaige. While chemical preservation reduces the sample under consideration to 257 individuals (89.2%), this still represents an enormous study sample by isotopic standards. In all cases, ~1.0 g (target weight) samples of dense cortical bone were removed from available skeletal elements using a diamond cut-off wheel (Dremel #545) mounted on a handheld Dremel Rotary Tool. The rotational speed of the cutoff wheel was kept at a minimum effective speed as to avoid causing any thermal degradation of bone collagen, and wheels were sterilized between individuals to avoid cross-contamination. The element and location of sampling varied from *ayllu* to *ayllu*, and on an individual level, based on completeness and the state of preservation. All sampling was performed in consultation with museum curatorial and conservation staff. Following extraction, samples were bagged in pre-labeled sterile sample bags. Permits for sample export were granted by the Chilean *Consejo de Monumentos Nacionales* (Permits 3682/12, 3219/15, 4276/16, and 5084/17).

**Extraction and analysis.** The extraction of collagen and hydroxyapatite from human bone samples was performed at the Archaeological Stable Isotope Laboratory at the University of Miami. Initially, each sample was ground by hand using a sterilized ceramic mortar and pestle and separated, using geological screens, into different size fractions, which were stored in sterilized scintillation vials until extraction.

Collagen extraction followed the protocol of Longin [69], as modified by Pestle [70]. Briefly, 0.5 g of the 0.5–1.0 mm size fraction was weighed and placed in a 50 ml centrifuge tube. Samples first were demineralized in 30 mL of 0.2 M HCL on a constantly spinning rotator for 24 h, at which time those samples that had achieved neutral buoyancy were then rinsed to neutral

**Table 1. Archaeological, bioarchaeological, chemical, isotopic, and FRUITS modeled foodstuff data for all 257 well-preserved San Pedro individuals included in the present study.**

| Sample | Ayllu | Cemetery | Individual | Sex | Collagen yield (wt%) | Apatite yield (wt%) | wt% C | wt% N | Atomic C:N | δ13Cco (‰) | δ15Nco (‰) | δ13Cap (‰) | Δ13Cap-co (‰) | C3 mean | C3 sd | C4/CAM mean | C4/CAM sd | Bean mean | Bean sd | Terrestrial meat mean | Terrestrial meat sd |
|---|---|---|---|---|---|---|---|---|---|---|---|---|---|---|---|---|---|---|---|---|---|
| L160 | Conde Duque | Casa Parroquial | 11, 418 | | 5.2 | 67.5 | 30.6 | 10.2 | 3.5 | -11.8 | 12.2 | -6.0 | 5.8 | 17.1% | 11.5% | 36.7% | 15.3% | 18.3% | 10.5% | 27.9% | 15.9% |
| L167 | Conde Duque | Casa Parroquial | 18, 418 | M | 17.3 | 56.7 | 42.0 | 15.0 | 3.3 | -11.0 | 13.4 | -6.4 | 4.6 | 23.5% | 14.6% | 39.0% | 12.2% | 11.4% | 8.4% | 26.1% | 16.7% |
| L162 | Conde Duque | Casa Parroquial | 19, 419 | M | 15.3 | 57.9 | 41.1 | 15.0 | 3.2 | -12.5 | 12.1 | -7.6 | 4.9 | 20.2% | 11.6% | 32.2% | 15.5% | 22.8% | 12.0% | 24.8% | 16.1% |
| L161 | Conde Duque | Casa Parroquial | 2, 414 | | 14.4 | 59.8 | 40.8 | 14.5 | 3.3 | -13.7 | 11.4 | -7.9 | 5.8 | 24.7% | 14.8% | 29.9% | 11.2% | 24.5% | 13.7% | 21.0% | 14.5% |
| L163 | Conde Duque | Casa Parroquial | 20, 420 | F | 14.2 | 59.7 | 40.2 | 14.4 | 3.3 | -15.4 | 11.8 | -9.4 | 6.1 | 25.6% | 15.8% | 16.1% | 9.3% | 32.1% | 18.6% | 26.2% | 13.5% |
| L166 | Conde Duque | Casa Parroquial | 22, 421 | M | 11.9 | 61.7 | 35.3 | 12.4 | 3.3 | -13.7 | 11.2 | -9.1 | 4.6 | 27.3% | 15.4% | 17.0% | 10.1% | 24.6% | 13.1% | 31.1% | 11.7% |
| J74 | Conde Duque | Casa Parroquial | Cuerpo 05 | F | 6.1 | 54.9 | 38.5 | 13.4 | 3.4 | -13.6 | 11.7 | -6.9 | 6.7 | 21.5% | 12.7% | 33.3% | 14.7% | 19.6% | 11.5% | 25.6% | 15.3% |
| J75 | Conde Duque | Casa Parroquial | Cuerpo 06 | | 12.4 | 55.6 | 42.7 | 15.5 | 3.2 | -14.1 | 12.3 | -8.2 | 5.9 | 29.7% | 13.4% | 30.0% | 10.9% | 17.6% | 10.7% | 22.8% | 13.3% |
| J71 | Conde Duque | Casa Parroquial | Cuerpo 08 | F | 13.9 | 67.5 | 41.6 | 15.1 | 3.2 | -13.3 | 12.0 | -6.7 | 6.7 | 19.3% | 12.7% | 33.5% | 13.2% | 21.3% | 11.8% | 25.9% | 15.8% |
| J72 | Conde Duque | Casa Parroquial | Cuerpo 09 | | 5.4 | 54.8 | 39.7 | 13.8 | 3.3 | -14.7 | 12.6 | -7.9 | 6.8 | 26.9% | 14.8% | 23.3% | 13.2% | 18.7% | 11.7% | 31.1% | 14.7% |
| J73 | Conde Duque | Casa Parroquial | Cuerpo 10 | M | 15.9 | 47.3 | 44.2 | 15.7 | 3.3 | -12.8 | 13.5 | -6.7 | 6.1 | 19.9% | 13.5% | 33.7% | 12.6% | 17.2% | 10.8% | 29.2% | 13.9% |
| L80 | Coyo | Coyo Oriente | 3974 | F | 21.4 | 43.5 | 44.2 | 15.5 | 3.3 | -12.3 | 11.3 | -7.0 | 5.3 | 14.7% | 11.0% | 37.1% | 9.6% | 26.9% | 12.8% | 21.4% | 13.7% |
| L93 | Coyo | Coyo Oriente | 3978 | F | 20.5 | 41.5 | 42.9 | 14.7 | 3.4 | -14.0 | 11.7 | -8.5 | 5.5 | 27.5% | 14.8% | 22.4% | 10.3% | 20.3% | 13.4% | 29.8% | 13.5% |
| L88 | Coyo | Coyo Oriente | 3981 | M | 10.2 | 47.5 | 40.0 | 14.0 | 3.3 | -14.0 | 11.4 | -8.8 | 5.3 | 25.6% | 14.2% | 24.0% | 10.0% | 26.2% | 12.8% | 24.3% | 14.6% |
| L90 | Coyo | Coyo Oriente | 3996 | F | 23.0 | 41.9 | 44.6 | 15.5 | 3.4 | -15.4 | 10.5 | -9.9 | 5.5 | 32.5% | 19.4% | 18.9% | 8.8% | 30.4% | 18.3% | 18.3% | 10.3% |
| L81 | Coyo | Coyo Oriente | 4012 | F | 21.4 | 45.4 | 44.2 | 15.4 | 3.4 | -15.3 | 10.9 | -10.2 | 5.1 | 31.5% | 18.2% | 17.1% | 9.2% | 29.5% | 15.3% | 21.8% | 12.8% |
| L92 | Coyo | Coyo Oriente | 4013 | F | 19.9 | 38.6 | 42.1 | 14.7 | 3.3 | -13.4 | 11.6 | -8.2 | 5.2 | 22.0% | 12.6% | 29.9% | 11.5% | 23.4% | 13.3% | 24.6% | 14.8% |
| L87 | Coyo | Coyo Oriente | 4028 | M | 20.5 | 46.4 | 42.1 | 14.7 | 3.3 | -14.2 | 10.9 | -8.7 | 5.5 | 22.6% | 13.4% | 20.3% | 10.7% | 29.2% | 14.1% | 27.9% | 13.5% |
| K18 | Coyo | Coyo Oriente | 4054 | F | 8.2 | 56.7 | 40.6 | 14.3 | 3.3 | -14.4 | 11.6 | -7.5 | 6.9 | 19.8% | 15.5% | 29.6% | 13.5% | 24.0% | 14.3% | 26.6% | 14.6% |
| L83 | Coyo | Coyo Oriente | 4085 | M | 20.9 | 43.1 | 42.7 | 14.7 | 3.4 | -14.8 | 11.8 | -10.4 | 4.4 | 33.5% | 15.7% | 15.9% | 10.2% | 24.6% | 15.6% | 26.0% | 12.6% |
| L79 | Coyo | Coyo Oriente | 4090 | M | 19.7 | 47.2 | 43.8 | 14.9 | 3.4 | -14.3 | 11.1 | -8.8 | 5.5 | 28.8% | 14.9% | 23.0% | 12.7% | 26.8% | 14.3% | 21.4% | 12.7% |
| L96 | Coyo | Coyo Oriente | 4111 | M | 19.7 | 43.5 | 43.2 | 14.9 | 3.4 | -13.4 | 11.2 | -8.4 | 4.9 | 29.0% | 14.6% | 29.1% | 11.8% | 21.5% | 12.8% | 20.5% | 12.7% |
| L94 | Coyo | Coyo Oriente | 4150 | M | 19.4 | 47.2 | 42.5 | 14.8 | 3.4 | -13.3 | 11.7 | -8.9 | 4.3 | 29.7% | 15.6% | 20.9% | 10.5% | 21.5% | 13.4% | 27.9% | 14.4% |
| L95 | Coyo | Coyo Oriente | 4194 | F | 15.5 | 50.6 | 43.5 | 15.0 | 3.4 | -15.2 | 11.8 | -10.3 | 4.9 | 34.1% | 17.8% | 14.7% | 8.8% | 26.5% | 15.3% | 24.8% | 13.5% |
| L86 | Coyo | Coyo Oriente | 5291 | | 22.6 | 41.9 | 41.6 | 14.7 | 3.3 | -14.0 | 12.4 | -8.8 | 5.2 | 34.4% | 12.9% | 23.6% | 12.7% | 18.8% | 12.5% | 23.3% | 14.5% |
| L85 | Coyo | Coyo Oriente | 5340 | F | 22.8 | 43.9 | 43.2 | 15.1 | 3.3 | -13.1 | 12.5 | -7.5 | 5.6 | 28.7% | 18.0% | 27.4% | 12.7% | 19.0% | 13.5% | 25.0% | 15.7% |
| L82 | Coyo | Coyo Oriente | 5344 | F | 16.3 | 37.6 | 42.3 | 14.5 | 3.4 | -13.7 | 11.2 | -8.7 | 5.0 | 28.7% | 17.2% | 24.9% | 11.1% | 24.6% | 15.0% | 21.8% | 11.8% |
| L91 | Coyo | Coyo Oriente | 5373 | M | 18.5 | 45.3 | 44.3 | 15.4 | 3.4 | -14.7 | 11.9 | -10.1 | 4.5 | 38.1% | 18.1% | 13.9% | 8.9% | 21.4% | 16.7% | 26.7% | 12.8% |
| L97 | Coyo | Coyo Oriente | 5375 | F | 19.6 | 41.3 | 42.4 | 14.8 | 3.3 | -15.0 | 9.9 | -9.5 | 5.5 | 25.3% | 15.2% | 20.0% | 10.3% | 30.9% | 16.0% | 23.8% | 13.8% |
| L89 | Coyo | Coyo Oriente | 5377 | F | 21.6 | 45.2 | 45.4 | 15.2 | 3.5 | -16.2 | 9.7 | -10.8 | 5.4 | 36.2% | 18.9% | 13.4% | 7.9% | 33.4% | 18.1% | 17.0% | 11.7% |
| L84 | Coyo | Coyo Oriente | 5383 | M | 21.4 | 40.7 | 42.3 | 14.5 | 3.4 | -13.5 | 11.1 | -8.8 | 4.8 | 23.0% | 14.2% | 23.0% | 11.8% | 26.8% | 14.9% | 27.2% | 13.9% |
| H33 | Coyo | Coyo 3 | 13291, T16, #93 | M | 14.4 | 51.7 | 42.0 | 15.3 | 3.2 | -11.1 | 14.5 | -7.1 | 4.0 | 22.4% | 13.4% | 39.4% | 13.9% | 13.3% | 10.4% | 25.0% | 18.2% |
| H31 | Coyo | Coyo 3 | 13298, T1, #89 | | 4.8 | 51.0 | 28.6 | 10.1 | 3.3 | -11.5 | 12.9 | -7.1 | 4.4 | 25.1% | 15.3% | 35.7% | 11.8% | 13.7% | 10.4% | 25.5% | 16.6% |
| H38 | Coyo | Coyo 3 | 13304, T13, #91 | | 9.7 | 45.7 | 36.1 | 13.3 | 3.2 | -14.1 | 10.6 | -9.3 | 4.8 | 32.5% | 17.6% | 19.3% | 10.2% | 25.8% | 15.1% | 22.4% | 14.4% |
| H32 | Coyo | Coyo 3 | 13335, T21, #90 | F | 16.9 | 43.3 | 42.3 | 15.5 | 3.2 | -11.8 | 11.5 | -6.4 | 5.4 | 17.1% | 11.6% | 34.2% | 14.5% | 21.1% | 10.5% | 27.6% | 14.3% |
| H42 | Coyo | Coyo 3 | 13363, T23, #86 | F | 13.4 | 45.1 | 37.8 | 14.1 | 3.1 | -12.4 | 11.6 | -7.7 | 4.7 | 22.6% | 15.2% | 25.8% | 11.3% | 21.3% | 13.4% | 24.7% | 13.3% |
| H34 | Coyo | Coyo 3 | 13608, T35, #94 | M | 8.9 | 60.4 | 38.3 | 13.8 | 3.2 | -12.7 | 12.2 | -9.0 | 3.7 | 34.2% | 14.8% | 22.6% | 13.3% | 17.1% | 11.8% | 22.9% | 15.1% |
| H44 | Coyo | Coyo 3 | 13639, T36, #85 | F | 16.1 | 44.7 | 38.6 | 14.4 | 3.1 | -14.4 | 11.1 | -9.0 | 5.4 | 21.5% | 14.9% | 21.2% | 9.4% | 32.6% | 14.4% | 26.2% | 14.6% |
| H37 | Coyo | Coyo 3 | 13688, T42, #87 | M | 1.5 | 59.6 | 14.7 | 5.0 | 3.4 | -14.4 | 10.9 | -9.7 | 4.7 | 30.9% | 17.6% | 18.2% | 9.9% | 25.1% | 15.1% | 24.7% | 14.0% |
| H39 | Coyo | Coyo 3 | 13746, T46, #92 | M | 11.5 | 58.2 | 35.9 | 13.1 | 3.2 | -12.9 | 12.9 | -8.9 | 4.0 | 33.3% | 15.9% | 21.2% | 10.3% | 17.6% | 11.8% | 25.9% | 13.8% |
| H41 | Coyo | Coyo 3 | 13794, T57, #84 | M | 2.4 | 51.3 | 15.3 | 5.1 | 3.5 | -15.8 | 10.7 | -10.0 | 5.8 | 29.8% | 16.2% | 14.2% | 7.8% | 30.1% | 16.5% | 25.9% | 12.5% |
| k72 | Coyo | Coyo Oriente | 3904 (11886), #313 | M | 18.1 | 40.9 | 45.1 | 16.1 | 3.3 | -15.8 | 10.3 | -9.9 | 5.9 | 32.5% | 20.2% | 13.5% | 8.4% | 32.4% | 17.0% | 21.7% | 13.6% |
| G94 | Coyo | Coyo Oriente | 3913 (12,059), #234 | | 15.5 | 50.2 | 42.9 | 15.6 | 3.2 | -13.6 | 11.9 | -8.3 | 5.3 | 30.7% | 17.5% | 23.3% | 10.7% | 22.2% | 14.4% | 23.8% | 14.2% |
| I67 | Coyo | Coyo Oriente | 3948 (9,965) | F | 18.5 | 41.7 | 36.4 | 13.4 | 3.2 | -15.7 | 10.7 | -10.1 | 5.6 | 32.7% | 17.5% | 14.1% | 9.9% | 32.9% | 16.9% | 20.3% | 13.2% |
| k71 | Coyo | Coyo Oriente | 3956 (9987), #312 | F | 9.6 | 51.3 | 43.8 | 15.4 | 3.3 | -15.0 | 10.9 | -8.9 | 6.0 | 22.2% | 15.3% | 23.5% | 10.3% | 29.6% | 15.0% | 24.7% | 13.2% |
| G99 | Coyo | Coyo Oriente | 3957 (9,799), #241 | M | 13.8 | 54.3 | 35.5 | 13.0 | 3.2 | -15.0 | 10.4 | -9.7 | 5.3 | 27.2% | 17.6% | 19.2% | 10.9% | 30.8% | 16.3% | 22.9% | 12.4% |
| G96 | Coyo | Coyo Oriente | 3959 (9971), #260 | F | 16.0 | 54.9 | 38.8 | 14.2 | 3.2 | -14.5 | 11.2 | -9.8 | 4.7 | 31.1% | 16.6% | 15.8% | 9.7% | 23.8% | 14.6% | 29.3% | 12.6% |
| k73 | Coyo | Coyo Oriente | 3973 (11497), #314 | | 18.3 | 49.6 | 45.9 | 16.5 | 3.3 | -12.9 | 11.9 | -7.5 | 5.4 | 26.0% | 14.2% | 31.8% | 13.6% | 20.6% | 13.9% | 21.6% | 13.5% |
| I75 | Coyo | Coyo Oriente | 3979 (13553) | F | 19.4 | 41.5 | 43.9 | 16.0 | 3.2 | -14.5 | 11.4 | -8.3 | 6.2 | 24.5% | 14.4% | 24.8% | 13.1% | 27.2% | 14.3% | 23.6% | 16.9% |

(*Continued*)

**Table 1.** (Continued)

| ID | Region | Site | Sample | Sex | | | | | | | | | | | | | | | | | | |
|---|---|---|---|---|---|---|---|---|---|---|---|---|---|---|---|---|---|---|---|---|---|---|
| k74 | Coyo | Coyo Oriente | 3984 (9791), #315 | F | 17.0 | 39.4 | 44.7 | 15.7 | 3.3 | -13.9 | 11.4 | -8.1 | 5.8 | 23.0 | 13.5 | 28.5 | 12.9 | 28.2 | 15.1 | 20.3 | 15.9% |
| I71 | Coyo | Coyo Oriente | 4003 (9825) | | 2.5 | 61.4 | 37.1 | 12.6 | 3.4 | -13.8 | 12.5 | -7.7 | 6.1 | 19.4 | 14.1 | 34.5 | 11.5 | 24.2 | 13.1 | 21.9 | 11.7% |
| I69 | Coyo | Coyo Oriente | 4005 (9836) | F | 20.1 | 45.9 | 43.7 | 15.6 | 3.3 | -14.7 | 11.4 | -8.4 | 6.3 | 25.2 | 16.1 | 28.1 | 12.3 | 26.3 | 15.5 | 20.5 | 13.4% |
| I73 | Coyo | Coyo Oriente | 4011 (13598) | M | 12.4 | 35.0 | 43.2 | 15.3 | 3.3 | -13.0 | 10.7 | -7.8 | 5.3 | 17.5 | 12.1 | 36.7 | 10.1 | 26.6 | 13.4 | 19.1 | 13.4% |
| k75 | Coyo | Coyo Oriente | 4020 (9838), #316 | M | 13.7 | 42.1 | 45.3 | 15.6 | 3.4 | -14.9 | 11.1 | -9.7 | 5.1 | 27.1 | 15.3 | 20.9 | 9.3 | 31.5 | 13.7 | 20.5 | 12.1% |
| I52 | Coyo | Coyo Oriente | 4031 (9961), #255 | F | 23.9 | 17.6 | 38.8 | 14.2 | 3.2 | -11.8 | 12.2 | -7.3 | 4.5 | 20.7 | 11.6 | 36.1 | 12.5 | 17.9 | 12.2 | 25.4 | 13.9% |
| I56 | Coyo | Coyo Oriente | 4046 (9940) | F | 7.5 | 51.7 | 39.7 | 14.8 | 3.1 | -14.5 | 12.0 | -8.6 | 5.9 | 24.0 | 16.8 | 23.8 | 12.0 | 29.3 | 16.3 | 22.9 | 14.7% |
| I66 | Coyo | Coyo Oriente | 4049 (9870) | M | 19.0 | 74.0 | 40.8 | 15.1 | 3.1 | -14.3 | 11.1 | -9.1 | 5.2 | 29.5 | 17.4 | 20.1 | 11.9 | 27.6 | 14.3 | 22.7 | 13.1% |
| I57 | Coyo | Coyo Oriente | 4052 (10031) | M | 19.7 | 41.1 | 42.4 | 16.0 | 3.1 | -15.3 | 10.9 | -9.8 | 5.5 | 27.4 | 16.6 | 20.3 | 10.3 | 33.4 | 16.0 | 18.9 | 9.6% |
| I59 | Coyo | Coyo Oriente | 4053 (9828) | F | 9.8 | 54.0 | 37.9 | 13.6 | 3.2 | -15.9 | 10.8 | -10.2 | 5.6 | 26.6 | 17.6 | 13.3 | 7.9 | 34.7 | 17.2 | 25.4 | 12.7% |
| I77 | Coyo | Coyo Oriente | 4054 (9872) | F | 4.9 | 51.6 | 42.7 | 14.5 | 3.4 | -16.3 | 11.3 | -8.5 | 7.9 | 27.8 | 16.2 | 20.8 | 10.9 | 25.3 | 13.8 | 26.1 | 13.6% |
| k76 | Coyo | Coyo Oriente | 4059 (11500), #317 | | 13.3 | 41.2 | 44.7 | 14.7 | 3.6 | -15.6 | 10.8 | -9.4 | 6.2 | 34.1 | 18.8 | 17.3 | 8.6 | 28.6 | 16.8 | 20.0 | 13.5% |
| G103 | Coyo | Coyo Oriente | 4060 (11458), #271 | M | 11.5 | 57.8 | 34.4 | 12.6 | 3.2 | -13.2 | 11.4 | -8.0 | 5.2 | 19.2 | 15.5 | 29.5 | 12.1 | 24.9 | 14.3 | 26.4 | 15.5% |
| k77 | Coyo | Coyo Oriente | 4064 (9860), #318 | M | 16.4 | 28.4 | 44.4 | 15.8 | 3.3 | -12.3 | 12.8 | -7.3 | 5.0 | 17.8 | 11.7 | 36.4 | 11.0 | 19.6 | 11.7 | 26.2 | 13.8% |
| I76 | Coyo | Coyo Oriente | 4065 (9991) | F | 20.1 | 48.2 | 29.1 | 10.4 | 3.3 | -17.4 | 12.7 | -11.6 | 5.7 | 42.8 | 17.4 | 10.2 | 7.5 | 26.2 | 16.2 | 20.8 | 12.9% |
| I64 | Coyo | Coyo Oriente | 4067 (9963) | M | 8.8 | 51.4 | 33.3 | 11.8 | 3.3 | -15.4 | 10.5 | -9.6 | 5.9 | 22.6 | 15.1 | 21.8 | 10.2 | 35.5 | 17.8 | 20.1 | 14.7% |
| I53 | Coyo | Coyo Oriente | 4069 (9986), #259 | | 8.3 | 41.1 | 26.4 | 9.4 | 3.3 | -13.3 | 10.9 | -8.3 | 5.1 | 24.9 | 16.6 | 27.0 | 13.0 | 23.3 | 13.2 | 24.8 | 14.5% |
| k78 | Coyo | Coyo Oriente | 4077 (9785), #319 | F | 14.7 | 45.7 | 44.6 | 15.8 | 3.3 | -15.4 | 10.4 | -9.9 | 5.4 | 36.8 | 20.3 | 14.6 | 7.8 | 28.1 | 16.7 | 20.5 | 12.1% |
| G102 | Coyo | Coyo Oriente | 4093 (10,041), #274 | M | 11.5 | 57.1 | 37.2 | 13.5 | 3.2 | -13.0 | 12.1 | -8.9 | 4.0 | 27.7 | 15.4 | 26.2 | 10.6 | 20.8 | 12.9 | 25.3 | 14.8% |
| I61 | Coyo | Coyo Oriente | 4098 (9937) | M | 20.5 | 31.1 | 43.5 | 16.7 | 3.0 | -14.6 | 12.1 | -9.0 | 5.6 | 31.0 | 17.3 | 22.8 | 10.7 | 24.7 | 14.3 | 21.5 | 14.3% |
| G93 | Coyo | Coyo Oriente | 4102 (9873), #245 | M | 19.6 | 51.1 | 42.6 | 15.5 | 3.2 | -14.4 | 10.8 | -9.4 | 5.0 | 27.1 | 17.3 | 20.7 | 9.3 | 30.0 | 15.6 | 22.2 | 13.7% |
| k79 | Coyo | Coyo Oriente | 4109 (9946), #320 | M | 14.0 | 45.0 | 45.5 | 16.0 | 3.3 | -12.1 | 11.1 | -6.3 | 5.8 | 23.3 | 13.6 | 38.7 | 15.0 | 17.0 | 11.5 | 21.0 | 16.8% |
| G95 | Coyo | Coyo Oriente | 4132 (9787), #248 | M | 14.0 | 48.2 | 43.2 | 15.6 | 3.2 | -14.2 | 11.3 | -8.8 | 5.3 | 22.8 | 13.6 | 23.2 | 10.3 | 29.1 | 16.0 | 25.0 | 12.6% |
| I81 | Coyo | Coyo Oriente | 4151 (9790), #322 | F | 12.4 | 43.2 | 44.4 | 15.6 | 3.3 | -13.4 | 12.1 | -8.1 | 5.3 | 28.5 | 14.6 | 27.4 | 13.8 | 21.9 | 13.3 | 22.2 | 16.7% |
| I68 | Coyo | Coyo Oriente | 4154 (9984) | | 4.4 | 46.9 | 43.7 | 15.3 | 3.3 | -14.1 | 10.7 | -6.5 | 7.6 | 19.2 | 12.2 | 38.0 | 11.9 | 21.0 | 11.8 | 21.8 | 14.7% |
| k82 | Coyo | Coyo Oriente | 4158 (9945), #323 | | 2.9 | 59.4 | 39.8 | 13.7 | 3.4 | -16.9 | 10.3 | -11.4 | 5.5 | 33.4 | 17.9 | 9.0 | 6.7 | 37.1 | 19.7 | 20.4 | 11.5% |
| I54 | Coyo | Coyo Oriente | 4163 (9871), #261 | | 15.2 | 42.5 | 34.3 | 12.4 | 3.2 | -14.6 | 10.9 | -10.0 | 4.6 | 33.6 | 18.2 | 17.8 | 10.7 | 26.0 | 15.8 | 22.6 | 14.6% |
| I58 | Coyo | Coyo Oriente | 4164 (9779) | M | 22.6 | 33.7 | 41.8 | 15.8 | 3.2 | -14.7 | 16.1 | -9.8 | 5.0 | 46.6 | 17.8 | 25.8 | 11.8 | 12.5 | 9.8 | 15.2 | 8.9% |
| I63 | Coyo | Coyo Oriente | 4175 (9974) | F | 18.8 | 40.4 | 39.6 | 15.3 | 3.0 | -13.9 | 11.7 | -7.9 | 6.0 | 24.0 | 13.3 | 20.6 | 11.5 | 24.2 | 13.3 | 31.2 | 12.7% |
| I55 | Coyo | Coyo Oriente | 4178 (12252), #254 | F | 22.1 | 27.5 | 36.6 | 13.3 | 3.2 | -12.4 | 11.3 | -7.8 | 4.7 | 15.6 | 11.6 | 37.4 | 13.8 | 26.6 | 12.9 | 20.4 | 13.9% |
| G98 | Coyo | Coyo Oriente | 4190 (12.2,43), #275 | M | 17.5 | 51.2 | 40.6 | 14.5 | 3.3 | -13.1 | 12.1 | -8.8 | 4.3 | 23.8 | 15.4 | 20.5 | 10.1 | 25.0 | 14.3 | 30.7 | 13.6% |
| G97 | Coyo | Coyo Oriente | 4193 (12247), #273 | M | 14.8 | 56.7 | 40.4 | 14.7 | 3.2 | -13.7 | 11.0 | -8.5 | 5.3 | 21.7 | 14.9 | 24.8 | 11.0 | 26.9 | 14.3 | 26.7 | 13.1% |
| I60 | Coyo | Coyo Oriente | 5300 (9856) | F | 1.7 | 30.4 | 36.7 | 12.7 | 3.4 | -14.5 | 11.5 | -8.5 | 6.0 | 32.4 | 16.2 | 26.1 | 10.0 | 22.0 | 13.8 | 19.5 | 12.4% |
| I51 | Coyo | Coyo Oriente | 5308 (9845), #252 | M | 6.4 | 66.4 | 31.0 | 10.9 | 3.3 | -10.6 | 12.7 | -6.1 | 4.5 | 16.7 | 12.8 | 48.1 | 11.6 | 13.3 | 10.1 | 21.8 | 14.7% |
| I72 | Coyo | Coyo Oriente | 5310 (10003) | M | 6.4 | 44.8 | 41.5 | 14.4 | 3.4 | -14.1 | 11.6 | -7.5 | 6.6 | 21.4 | 14.1 | 29.3 | 12.5 | 21.7 | 11.7 | 27.7 | 14.3% |
| G101 | Coyo | Coyo Oriente | 5316 (9.842), #250 | F | 3.1 | 64.5 | 23.7 | 8.5 | 3.3 | -12.1 | 11.9 | -7.3 | 4.9 | 21.1 | 14.0 | 36.2 | 10.9 | 17.7 | 11.6 | 25.1 | 15.8% |
| G92 | Coyo | Coyo Oriente | 5317 (9806), #264 | M | 8.2 | 50.2 | 32.8 | 11.8 | 3.3 | -13.9 | 11.5 | -8.2 | 5.7 | 24.5 | 18.0 | 25.7 | 10.8 | 25.3 | 16.2 | 24.5 | 15.0% |
| I65 | Coyo | Coyo Oriente | 5334 (9846) | F | 21.0 | 39.6 | 37.5 | 14.0 | 3.1 | -13.7 | 11.2 | -8.1 | 5.6 | 25.8 | 12.5 | 23.0 | 10.8 | 21.1 | 11.8 | 30.1 | 13.8% |
| k83 | Coyo | Coyo Oriente | 5335 (9952), #324 | F | 13.9 | 42.5 | 44.4 | 15.7 | 3.3 | -15.6 | 10.8 | -10.3 | 5.4 | 39.0 | 18.3 | 14.7 | 9.4 | 25.5 | 15.0 | 20.8 | 13.0% |
| k84 | Coyo | Coyo Oriente | 5343 (13552), #325 | M | 16.8 | 41.4 | 45.6 | 16.2 | 3.3 | -14.3 | 10.7 | -9.0 | 5.3 | 29.1 | 16.2 | 26.5 | 8.5 | 27.6 | 17.0 | 16.8 | 10.7% |
| I62 | Coyo | Coyo Oriente | 5345 (9811) | F | 23.1 | 20.0 | | | 3.1 | -13.9 | 12.6 | -8.6 | 5.3 | 28.9 | 14.3 | 24.4 | 11.3 | 20.3 | 12.0 | 26.4 | 14.3% |
| I70 | Coyo | Coyo Oriente | 5352 (9808) | F | 18.0 | 49.0 | 44.2 | 16.1 | 3.2 | -16.4 | 10.6 | -9.4 | 7.0 | 29.0 | 17.2 | 18.3 | 8.6 | 33.7 | 18.0 | 18.9 | 12.4% |
| L108 | Coyo | Coyo 3 | T. 10 | F | 5.6 | 45.9 | 39.4 | 13.5 | 3.4 | -14.8 | 11.3 | -9.3 | 5.5 | 25.8 | 16.6 | 18.1 | 10.4 | 26.7 | 16.1 | 29.4 | 12.7% |
| L112 | Coyo | Coyo 3 | T. 13 | F | 9.6 | 46.6 | 40.3 | 14.0 | 3.4 | -13.8 | 9.6 | -8.2 | 5.6 | 20.4 | 14.8 | 34.6 | 11.1 | 31.0 | 14.8 | 14.0 | 10.9% |
| L109 | Coyo | Coyo 3 | T. 18 | M | 18.5 | 49.3 | 41.9 | 14.7 | 3.3 | -12.4 | 12.1 | -7.3 | 5.1 | 20.2 | 13.5 | 40.1 | 12.2 | 17.3 | 12.9 | 22.4 | 15.6% |
| L110 | Coyo | Coyo 3 | T. 28 | F | 11.8 | 47.0 | 41.1 | 14.5 | 3.3 | -13.2 | 11.3 | -7.6 | 5.6 | 24.3 | 14.7 | 29.4 | 11.7 | 21.9 | 13.1 | 24.4 | 13.1% |
| L114 | Coyo | Coyo 3 | T. 32 | F | 17.7 | 52.0 | 44.0 | 15.5 | 3.3 | -14.9 | 10.7 | -10.2 | 4.6 | 25.4 | 18.3 | 19.3 | 10.1 | 34.7 | 18.9 | 20.6 | 12.6% |
| L111 | Coyo | Coyo 3 | T. 51 | M | 7.2 | 47.7 | 38.8 | 13.1 | 3.5 | -15.7 | 11.1 | -10.1 | 5.6 | 35.1 | 20.5 | 15.3 | 9.1 | 28.5 | 16.0 | 21.2 | 11.2% |
| H86 | Larache | Larache | 115, 11021, #38 | F | 10.0 | 48.4 | 32.4 | 11.8 | 3.2 | -13.4 | 11.7 | -7.6 | 5.7 | 23.3 | 12.8 | 31.9 | 13.0 | 18.4 | 10.8 | 26.5 | 13.7% |
| H87 | Larache | Larache | 117, 11073, #41 | M | 16.3 | 37.0 | 40.7 | 14.8 | 3.2 | -13.8 | 11.8 | -10.3 | 3.5 | 27.1 | 15.8 | 18.5 | 11.0 | 28.7 | 15.4 | 25.7 | 13.9% |
| H85 | Larache | Larache | 124, 10909, #33 | M | 12.9 | 47.8 | 38.6 | 14.0 | 3.2 | -11.4 | 12.8 | -5.9 | 5.5 | 19.5 | 10.6 | 37.0 | 17.6 | 14.5 | 9.2 | 29.0 | 18.5% |

*(Continued)*

**Table 1.** (Continued)

| | | | | | | | | | | | | | | | | | | | | |
|---|---|---|---|---|---|---|---|---|---|---|---|---|---|---|---|---|---|---|---|---|
| H51 | Larache | Larache | 125, 10412, #31 | F | 14.9 | 44.4 | 38.6 | 14.1 | 3.2 | -13.5 | 12.4 | -7.2 | 6.3 | 14.9% | 13.1% | 32.5% | 11.1% | 24.1% | 12.5% | 28.5% | 14.7% |
| H77 | Larache | Larache | 127, 10926, #30 | F | 2.6 | 69.6 | 24.0 | 8.4 | 3.3 | -13.1 | 11.4 | -8.2 | 4.9 | 28.0% | 13.7% | 26.9% | 13.4% | 21.6% | 12.1% | 23.6% | 14.6% |
| H72 | Larache | Larache | 1583, 10617, #40 | F | 9.9 | 52.3 | 34.2 | 12.4 | 3.2 | -14.7 | 10.9 | -8.9 | 5.8 | 28.3% | 15.3% | 21.5% | 10.8% | 26.7% | 14.6% | 23.5% | 14.5% |
| H83 | Larache | Larache | 3480, 9902, #45 | M | 10.4 | 49.1 | 36.2 | 13.2 | 3.2 | -15.0 | 12.1 | -8.7 | 6.4 | 27.1% | 15.5% | 25.1% | 10.9% | 24.9% | 14.3% | 22.9% | 13.3% |
| H73 | Larache | Larache | 356, 14669, #43 | F | 14.0 | 55.9 | 35.6 | 12.9 | 3.2 | -12.0 | 12.5 | -7.0 | 5.0 | 23.6% | 13.1% | 30.6% | 13.0% | 15.3% | 9.6% | 30.5% | 15.2% |
| H82 | Larache | Larache | 357, 10759, #39 | F | 9.3 | 47.1 | 33.9 | 12.3 | 3.2 | -14.3 | 10.6 | -8.2 | 6.1 | 26.8% | 16.2% | 27.2% | 10.2% | 26.0% | 13.8% | 20.0% | 13.8% |
| H78 | Larache | Larache | 358, 14670, #37 | M | 15.6 | 47.6 | 39.2 | 14.4 | 3.2 | -10.0 | 12.1 | -4.6 | 5.4 | 11.5% | 8.7% | 54.4% | 11.1% | 11.9% | 7.6% | 22.2% | 14.1% |
| H80 | Larache | Larache | 360, 10755, #36 | M | 8.9 | 58.4 | 34.2 | 14.2 | 3.3 | -9.6 | 11.2 | -4.6 | 5.0 | 9.0% | 6.3% | 54.9% | 13.5% | 11.4% | 6.9% | 24.7% | 13.7% |
| H84 | Larache | Larache | 366, 10758, #32 | F | 15.1 | 45.4 | 36.5 | 13.3 | 3.2 | -13.7 | 11.4 | -7.9 | 5.8 | 24.7% | 14.8% | 29.9% | 11.2% | 24.5% | 13.7% | 21.0% | 14.5% |
| H74 | Larache | Larache | 3797, 11081, #44 | F | 9.7 | 61.9 | 34.8 | 12.6 | 3.2 | -15.3 | 8.6 | -8.4 | 6.9 | 17.2% | 12.1% | 23.4% | 12.2% | 40.2% | 14.3% | 19.3% | 15.0% |
| H71 | Larache | Larache | 3802, 11209, #46 | F | 8.0 | 56.9 | 32.6 | 11.9 | 3.2 | -16.6 | 9.1 | -8.4 | 5.2 | 34.2% | 20.0% | 9.3% | 6.2% | 36.7% | 19.8% | 19.8% | 12.1% |
| H75 | Larache | Larache | 3803, 11082, #42 | F | 11.1 | 63.6 | 33.5 | 12.1 | 3.2 | -15.7 | 8.4 | -8.2 | 7.5 | 23.8% | 18.7% | 29.9% | 9.9% | 34.1% | 17.0% | 12.3% | 8.5% |
| H76 | Larache | Larache | 390, 10754, #35 | M | 11.3 | 56.4 | 33.5 | 12.2 | 3.2 | -10.0 | 11.3 | -4.6 | 5.4 | 12.5% | 8.6% | 46.7% | 16.7% | 11.6% | 7.3% | 29.2% | 15.7% |
| H79 | Larache | Larache | 5055, 12014, #34 | F | 12.4 | 56.3 | 34.6 | 12.3 | 3.3 | -16.4 | 10.7 | -10.3 | 6.1 | 42.1% | 19.2% | 10.5% | 6.8% | 28.3% | 16.4% | 19.1% | 10.0% |
| K17 | Larache | Larache | Callejo, segundo piso, T360, C-10755 | M | 13.2 | 53.0 | 42.2 | 14.8 | 3.3 | -11.0 | 11.6 | -3.6 | 7.4 | 9.0% | 7.0% | 49.7% | 15.2% | 11.5% | 7.5% | 29.8% | 14.2% |
| K16 | Larache | Larache | Callejon, segundo piso, T358, C-14670 | M | 22.5 | 55.5 | 41.3 | 14.8 | 3.3 | -10.3 | 12.2 | -4.3 | 6.1 | 8.1% | 6.9% | 56.2% | 13.3% | 9.8% | 7.2% | 25.9% | 16.7% |
| K15 | Larache | Larache | Capa inferior, segundo piso, T390, C-10754 | M | 10.3 | 60.2 | 23.3 | 8.0 | 3.4 | -11.3 | 11.1 | -4.1 | 7.2 | 10.4% | 8.6% | 54.5% | 11.0% | 14.1% | 8.1% | 21.0% | 12.9% |
| J76 | Quitor | Quitor 5 | 1957 | | 11.6 | 50.3 | 44.0 | 15.9 | 3.2 | -17.2 | 9.6 | -10.2 | 7.0 | 32.8% | 20.9% | 12.1% | 7.1% | 37.0% | 18.3% | 18.2% | 11.4% |
| J78 | Quitor | Quitor 5 | 1964 | | 21.0 | 29.0 | 43.9 | 15.8 | 3.2 | -17.7 | 9.4 | -11.6 | 6.1 | 29.9% | 18.8% | 11.6% | 9.5% | 41.5% | 19.6% | 17.1% | 12.1% |
| J77 | Quitor | Quitor 5 | 2021 | M | 17.8 | 46.9 | 44.2 | 15.8 | 3.3 | -17.2 | 11.4 | -10.9 | 6.4 | 33.7% | 20.1% | 11.1% | 7.6% | 33.8% | 17.9% | 21.4% | 11.6% |
| H104 | Quitor | Quitor 6 Tardío | 104, #164, T10 | F | 22.1 | 42.4 | 39.0 | 14.9 | 3.2 | -13.0 | 11.4 | -7.8 | 5.2 | 22.9% | 13.2% | 22.6% | 12.0% | 24.5% | 14.0% | 30.0% | 14.0% |
| J20 | Quitor | Quitor 5 | 1916, 9,741, #100 | | 13.1 | 48.2 | 32.7 | 14.4 | 3.2 | -14.7 | 9.5 | -9.2 | 5.5 | 32.8% | 15.6% | 19.5% | 8.9% | 26.6% | 15.3% | 21.1% | 13.4% |
| J33 | Quitor | Quitor 5 | 1921, 9741, #107 | M | 6.5 | 54.8 | 30.7 | 10.9 | 3.3 | -13.0 | 10.5 | -7.5 | 5.5 | 22.6% | 13.3% | 31.0% | 13.6% | 25.0% | 14.5% | 21.5% | 14.8% |
| J29 | Quitor | Quitor 5 | 1942, 9737, #96 | | 19.1 | 49.1 | 40.5 | 14.9 | 3.2 | -17.0 | 9.0 | -12.3 | 4.6 | 36.4% | 20.5% | 7.1% | 5.0% | 40.7% | 20.0% | 15.8% | 8.8% |
| J36 | Quitor | Quitor 5 | 2020, 9639, #212 | F | 16.2 | 58.5 | 30.4 | 11.0 | 3.2 | -16.6 | 8.3 | -12.2 | 4.4 | 31.0% | 23.0% | 9.6% | 7.6% | 46.8% | 21.2% | 12.6% | 10.2% |
| J28 | Quitor | Quitor 5 | 2026, 9732, #102 | M | 21.3 | 44.3 | 40.4 | 14.5 | 3.2 | -15.4 | 9.1 | -10.8 | 4.6 | 24.9% | 17.5% | 14.1% | 7.6% | 41.4% | 18.1% | 19.7% | 10.6% |
| J26 | Quitor | Quitor 5 | 2055, 9074, #112 | | 24.1 | 30.4 | 42.2 | 15.2 | 3.2 | -16.4 | 9.5 | -10.2 | 6.2 | 30.8% | 22.7% | 15.6% | 8.8% | 39.2% | 20.6% | 14.5% | 10.8% |
| J25 | Quitor | Quitor 5 | 2100, 9775, #97 | M | 15.9 | 47.1 | 40.5 | 14.8 | 3.2 | -12.8 | 10.7 | -7.6 | 5.2 | 21.1% | 14.1% | 29.0% | 12.0% | 23.4% | 13.0% | 26.6% | 14.1% |
| J24 | Quitor | Quitor 5 | 2109, 9766, #104 | | 18.4 | 45.2 | 39.3 | 14.0 | 3.3 | -14.9 | 11.0 | -11.1 | 3.8 | 32.5% | 18.0% | 10.9% | 8.1% | 31.1% | 18.5% | 25.5% | 12.8% |
| J37 | Quitor | Quitor 5 | 2125, 9628, #106 | M | 21.4 | 40.3 | 39.3 | 14.3 | 3.2 | -16.1 | 10.0 | -11.4 | 4.7 | 39.3% | 16.7% | 11.1% | 7.8% | 31.6% | 15.6% | 18.0% | 11.6% |
| J23 | Quitor | Quitor 5 | 2179, 9763, #111 | M | 2.9 | 64.5 | 17.9 | 6.3 | 3.3 | -10.9 | 10.8 | -5.3 | 5.6 | 13.8% | 9.1% | 48.7% | 12.4% | 16.1% | 11.6% | 21.5% | 12.8% |
| J30 | Quitor | Quitor 5 | 2212, 9658, #99 | F | 8.6 | 61.3 | 32.7 | 11.7 | 3.2 | -12.5 | 10.7 | -7.9 | 4.6 | 23.8% | 14.1% | 27.0% | 13.9% | 22.9% | 13.1% | 26.3% | 16.4% |
| H103 | Quitor | Quitor 6 Tardío | 222/223, #165, T17 | F | 21.2 | 37.4 | 36.1 | 13.3 | 3.2 | -13.2 | 10.3 | -7.3 | 5.9 | 19.7% | 12.8% | 40.5% | 11.9% | 24.0% | 13.3% | 15.8% | 11.2% |
| J34 | Quitor | Quitor 5 | 2228, 9773, #103 | | 4.7 | 62.2 | 19.4 | 6.7 | 3.4 | -15.6 | 9.2 | -10.2 | 5.3 | 34.3% | 21.0% | 13.7% | 8.1% | 30.8% | 15.7% | 21.2% | 13.4% |
| J21 | Quitor | Quitor 5 | 2245, 11,485, #109 | M | 18.9 | 44.1 | 38.1 | 13.6 | 3.3 | -11.5 | 10.2 | -6.3 | 5.2 | 13.0% | 9.8% | 42.4% | 12.3% | 18.0% | 10.2% | 26.7% | 16.1% |
| J22 | Quitor | Quitor 5 | 3066, 9754, #98 | F | 6.3 | 63.0 | 30.0 | 10.7 | 3.3 | -16.6 | 9.3 | -12.8 | 3.8 | 31.2% | 19.9% | 6.5% | 5.2% | 45.8% | 19.1% | 16.5% | 11.4% |
| H111 | Quitor | Quitor 8 | 3146, 12,281, #230 | F | 3.4 | 64.9 | 25.2 | 8.8 | 3.4 | -16.4 | 10.3 | -9.8 | 6.6 | 36.1% | 17.7% | 15.0% | 7.8% | 28.2% | 15.9% | 20.8% | 13.6% |
| I2 | Quitor | Quitor 8 | 3202, 12,294, #233 | F | 2.9 | 58.1 | 26.3 | 9.2 | 3.3 | -16.5 | 10.5 | -12.3 | 4.2 | 38.9% | 20.1% | 10.1% | 7.3% | 30.8% | 18.6% | 20.2% | 12.4% |
| H113 | Quitor | Quitor 8 | 3227, s/n, #229 | | 4.3 | 65.3 | 27.1 | 9.4 | 3.3 | -15.8 | 10.9 | -11.3 | 4.5 | 44.6% | 20.6% | 11.2% | 8.1% | 24.1% | 17.3% | 20.1% | 12.6% |
| H91 | Quitor | Quitor 9 | 3236, #223 | M | 14.3 | 47.2 | 38.9 | 14.2 | 3.2 | -11.4 | 12.5 | -6.0 | 5.4 | 22.8% | 13.9% | 42.7% | 12.8% | 13.5% | 9.0% | 21.0% | 15.6% |
| H90 | Quitor | Quitor 9 | 3237, #221 | M | 21.5 | 41.0 | 40.2 | 14.8 | 3.2 | -11.5 | 12.3 | -7.4 | 4.0 | 21.5% | 12.1% | 33.7% | 12.2% | 14.8% | 10.5% | 30.0% | 15.8% |
| H92 | Quitor | Quitor 9 | 3249, #222 | M | 17.4 | 40.3 | 39.2 | 14.6 | 3.1 | -11.7 | 11.2 | -5.9 | 5.8 | 16.8% | 9.6% | 47.9% | 13.4% | 14.5% | 9.0% | 20.8% | 13.1% |
| H88 | Quitor | Quitor 9 | 3250, #220 | F | 2.4 | 38.7 | 18.8 | 6.4 | 3.4 | -13.7 | 12.1 | -8.2 | 5.5 | 25.9% | 17.1% | 25.4% | 10.9% | 22.1% | 13.8% | 26.6% | 16.5% |
| J31 | Quitor | Quitor 5 | 3368, 9697, #105 | F | 2.6 | 67.9 | 16.4 | 5.7 | 3.4 | -16.5 | 8.6 | -12.5 | 4.0 | 38.3% | 23.6% | 7.9% | 5.6% | 39.7% | 21.4% | 14.1% | 8.3% |
| J35 | Quitor | Quitor 8 | 3370(80?), 9727, #108 | | 21.3 | 36.3 | 39.4 | 14.4 | 3.2 | -15.9 | 9.7 | -11.8 | 4.2 | 37.1% | 19.5% | 7.7% | 6.5% | 35.6% | 18.9% | 19.7% | 12.9% |
| J32 | Quitor | Quitor 5 | 3397, 9659, #110 | F | 11.1 | 57.1 | | | | -17.3 | 7.5 | -12.5 | 4.7 | 34.6% | 19.9% | 10.5% | 6.5% | 45.3% | 20.4% | 9.7% | 7.7% |
| J45 | Quitor | Quitor 1 | 3443, #281 | | 7.9 | 59.4 | 33.1 | 12.7 | 3.0 | -13.4 | 11.1 | -7.1 | 6.3 | 19.7% | 12.5% | 33.5% | 11.5% | 20.6% | 11.6% | 26.2% | 15.4% |
| J43 | Quitor | Quitor 1 | 3454, 10612, #282 | F | 14.9 | 50.7 | 34.1 | 12.2 | 3.3 | -13.2 | 11.0 | -7.8 | 5.5 | 19.8% | 15.0% | 26.6% | 11.3% | 24.1% | 13.4% | 29.5% | 14.1% |

(Continued)

**Table 1.** (Continued)

| ID | Region | Subsite | Sample | | Sex | | | | | | | | | | | | | | | | | |
|---|---|---|---|---|---|---|---|---|---|---|---|---|---|---|---|---|---|---|---|---|---|---|
| I42 | Quitor | Quitor 1 | 3486, #280 | 16.5 | | 48.7 | 43.3 | 15.5 | 3.3 | -12.2 | 11.7 | -7.4 | 4.7 | 17.1 | 12.0% | 29.4% | 13.3% | 22.0% | 10.8% | 31.6% | 17.1% |
| I44 | Quitor | Quitor 1 | 3487, 10603, #284 | 6.3 | | 65.5 | 22.3 | 7.6 | 3.4 | -15.2 | 13.0 | -9.3 | 5.8 | 43.7 | 13.8% | 22.6% | 11.9% | 13.9% | 11.3% | 19.8% | 14.0% |
| I47 | Quitor | Quitor 1 | 3493, 10605, #283 | 22.8 | | 43.6 | 38.2 | 13.9 | 3.2 | -14.0 | 9.5 | -7.4 | 6.6 | 17.7 | 13.3% | 28.8% | 10.1% | 32.3% | 16.3% | 21.1% | 12.9% |
| L156 | Quitor | Quitor 6 Tardio | 407, #105, T10 | 21.7 | F | 48.9 | 42.5 | 15.5 | 3.2 | -13.7 | 11.6 | -8.3 | 5.3 | 26.0 | 15.0% | 23.1% | 11.9% | 20.6% | 13.6% | 30.4% | 17.0% |
| L155 | Quitor | Quitor 6 Tardio | 408, #877, T28 | 19.8 | M | 48.6 | 42.3 | 15.5 | 3.2 | -15.9 | 9.7 | -10.5 | 5.3 | 28.9 | 17.7% | 15.6% | 9.3% | 34.1% | 16.4% | 21.4% | 12.5% |
| L157 | Quitor | Quitor 6 Tardio | 411, #425/426, T37 | 2.7 | F | 70.5 | 32.8 | 10.9 | 3.5 | -14.6 | 11.6 | -8.3 | 6.2 | 19.2 | 12.4% | 25.0% | 13.0% | 28.7% | 14.7% | 27.1% | 13.6% |
| L158 | Quitor | Quitor 6 Tardio | 412, #596, T46 | 22.2 | F | 46.1 | 42.8 | 15.6 | 3.2 | -14.2 | 11.3 | -8.7 | 5.6 | 26.1 | 14.9% | 28.2% | 12.5% | 24.0% | 14.0% | 21.8% | 15.3% |
| H95 | Quitor | Quitor 6 Tardio | 469/470, #158, T14 | 19.2 | F | 33.6 | 39.2 | 14.5 | 3.2 | -13.6 | 10.8 | -7.8 | 5.9 | 25.9 | 15.4% | 25.2% | 10.9% | 22.8% | 13.0% | 26.1% | 16.4% |
| H106 | Quitor | Quitor 6 Tardio | 667, #166, T52 | 1.7 | F | 71.1 | 10.9 | 3.5 | 3.6 | -13.7 | 10.9 | -7.9 | 5.8 | 21.6 | 15.2% | 31.1% | 13.9% | 24.8% | 12.8% | 22.4% | 13.4% |
| H107 | Quitor | Quitor 6 Tardio | 685, #161, T54 | 7.0 | F | 52.1 | 31.7 | 11.7 | 3.2 | -11.2 | 11.5 | -6.2 | 5.0 | 12.7 | 8.9% | 43.9% | 14.3% | 17.9% | 10.9% | 25.5% | 17.2% |
| H99 | Quitor | Quitor 6 Tardio | 694, #162, T55 | 14.2 | F | 45.3 | 35.6 | 13.1 | 3.2 | -13.0 | 9.9 | -7.7 | 5.3 | 19.5 | 14.4% | 35.9% | 10.1% | 28.4% | 15.4% | 16.1% | 12.2% |
| H105 | Quitor | Quitor 6 Tardio | 80/81, #163, T8 | 4.8 | F | 65.9 | 28.3 | 9.9 | 3.3 | -14.4 | 10.7 | -8.7 | 5.7 | 24.9 | 15.7% | 23.3% | 11.2% | 27.8% | 13.6% | 24.0% | 14.2% |
| H96 | Quitor | Quitor 6 Tardio | 871, #168, T27 | 19.3 | M | 36.8 | 38.0 | 13.6 | 3.3 | -15.9 | 10.8 | -10.3 | 5.6 | 32.6 | 15.1% | 10.6% | 6.8% | 30.7% | 15.2% | 26.1% | 13.8% |
| H100 | Quitor | Quitor 6 Tardio | 876, #170, T28 | 21.0 | | 36.8 | 38.0 | 13.9 | 3.2 | -15.6 | 9.3 | -8.8 | 6.8 | 21.7 | 14.5% | 20.0% | 10.5% | 33.7% | 17.6% | 24.7% | 13.2% |
| H101 | Quitor | Quitor 6 Tardio | 877, #167, T28 | 20.4 | F | 40.4 | 37.2 | 13.6 | 3.2 | -15.2 | 9.7 | -10.1 | 5.1 | 26.2 | 16.2% | 19.7% | 9.2% | 35.9% | 19.2% | 18.3% | 12.3% |
| H97 | Quitor | Quitor 6 Tardio | 88/89, #169, T9 | 8.3 | F | 55.2 | 32.2 | 11.6 | 3.2 | -13.9 | 10.5 | -8.5 | 5.4 | 25.3 | 16.5% | 27.5% | 11.9% | 28.0% | 14.6% | 19.2% | 13.3% |
| H98 | Quitor | Quitor 6 Tardio | 943, #160, T38 | 3.8 | F | 60.3 | 25.5 | 8.8 | 3.4 | -13.7 | 11.4 | -8.0 | 5.7 | 24.8 | 13.0% | 27.8% | 12.6% | 22.0% | 14.0% | 25.4% | 12.9% |
| L103 | Solcor | Solcor Plaza | 611 | 22.4 | F | 40.4 | 43.8 | 15.2 | 3.4 | -12.5 | 11.9 | -7.0 | 5.4 | 22.6 | 13.8% | 36.8% | 11.7% | 17.1% | 10.7% | 23.5% | 14.6% |
| L100 | Solcor | Solcor Plaza | 621 | 17.6 | M | 49.2 | 44.3 | 15.5 | 3.3 | -15.5 | 12.9 | -11.2 | 4.3 | 37.8 | 23.5% | 10.7% | 7.8% | 26.2% | 16.9% | 25.3% | 15.1% |
| L106 | Solcor | Solcor Plaza | 625 | 20.5 | M | 42.5 | 43.9 | 15.2 | 3.4 | -16.8 | 10.9 | -10.2 | 6.6 | 34.5 | 17.0% | 17.2% | 9.4% | 32.6% | 16.5% | 15.6% | 12.2% |
| L104 | Solcor | Solcor Plaza | 759 | 5.9 | F | 58.3 | 37.0 | 12.5 | 3.5 | -13.1 | 12.8 | -6.8 | 6.3 | 19.1 | 10.7% | 40.5% | 12.6% | 15.2% | 9.0% | 25.2% | 16.3% |
| L105 | Solcor | Solcor Plaza | 1242 | 18.4 | M | 47.0 | 42.8 | 14.6 | 3.4 | -13.6 | 12.3 | -8.7 | 4.9 | 27.9 | 17.9% | 18.9% | 9.6% | 24.9% | 14.6% | 28.3% | 13.7% |
| L99 | Solcor | Solcor Plaza | 1249 | 17.3 | M | 48.8 | 42.8 | 14.9 | 3.4 | -14.2 | 13.0 | -8.3 | 5.9 | 28.1 | 15.0% | 23.0% | 10.3% | 19.9% | 14.6% | 29.0% | 14.9% |
| L98 | Solcor | Solcor Plaza | 1283 | 20.5 | M | 41.7 | 43.1 | 14.6 | 3.4 | -14.5 | 11.7 | -9.3 | 5.3 | 34.3 | 16.8% | 22.9% | 12.5% | 21.2% | 15.5% | 21.6% | 15.4% |
| L101 | Solcor | Solcor Plaza | 1398 | 17.0 | F | 55.7 | 43.2 | 15.2 | 3.3 | -14.9 | 11.5 | -9.5 | 5.4 | 26.8 | 14.8% | 15.6% | 9.8% | 31.8% | 15.8% | 25.7% | 12.0% |
| L61 | Solcor | Solcor 3 | 1737 | 19.6 | F | 50.4 | 45.1 | 16.0 | 3.3 | -12.9 | 11.7 | -6.6 | 6.3 | 16.6 | 10.2% | 35.6% | 12.5% | 20.0% | 12.8% | 27.8% | 15.8% |
| L107 | Solcor | Solcor Plaza | 2938 | 21.3 | F | 31.7 | 45.7 | 15.2 | 3.4 | -13.5 | 12.0 | -7.4 | 6.1 | 22.5 | 13.6% | 34.1% | 11.7% | 21.9% | 11.6% | 21.6% | 14.9% |
| L102 | Solcor | Solcor Plaza | 2940 | 18.5 | F | 37.4 | 43.4 | 15.1 | 3.4 | -14.2 | 11.5 | -8.6 | 5.5 | 25.9 | 14.0% | 23.2% | 12.8% | 23.2% | 13.8% | 27.8% | 14.6% |
| L115 | Solcor | Solcor Nueva Población | 4778 | 20.7 | | 48.8 | 46.0 | 16.3 | 3.3 | -14.3 | 11.5 | -8.0 | 6.3 | 28.5 | 15.4% | 28.2% | 11.5% | 18.6% | 11.6% | 24.7% | 13.9% |
| L117 | Solcor | Solcor Nueva Población | 4789 | 21.8 | M | 47.9 | 44.3 | 15.8 | 3.3 | -16.0 | 11.8 | -10.5 | 5.5 | 36.2 | 19.9% | 11.8% | 7.3% | 29.0% | 18.5% | 23.0% | 13.5% |
| L116 | Solcor | Solcor Nueva Población | 4791 | 22.3 | M | 44.0 | 44.7 | 15.7 | 3.3 | -13.6 | 11.2 | -8.2 | 5.4 | 20.0 | 15.0% | 26.7% | 12.7% | 29.6% | 14.3% | 23.8% | 14.9% |
| G73 | Solcor | Solcor Plaza | 1030, 11,197, #77 | 1.3 | F | 52.6 | 17.5 | 5.9 | 3.4 | -11.6 | 12.4 | -6.8 | 4.8 | 18.0 | 12.9% | 35.1% | 13.7% | 17.7% | 11.8% | 29.3% | 15.0% |
| K70 | Solcor | Solcor Plaza | 1241 (11361), #330 | 20.3 | | 33.6 | 44.1 | 16.0 | 3.2 | -15.0 | 11.4 | -8.7 | 6.3 | 25.5 | 13.9% | 21.7% | 11.2% | 25.2% | 14.5% | 27.5% | 16.2% |
| G78 | Solcor | Solcor Plaza | 1241, 11,361, #82 | 21.6 | | 38.3 | 42.2 | 15.8 | 3.1 | -14.7 | 12.2 | -9.5 | 5.2 | 31.1 | 16.4% | 13.9% | 7.7% | 25.2% | 15.4% | 29.8% | 12.4% |
| K69 | Solcor | Solcor Plaza | 1243 (11351), #329 | 20.4 | M | 35.1 | 43.9 | 15.9 | 3.2 | -14.8 | 11.7 | -8.1 | 6.7 | 28.2 | 18.1% | 27.3% | 10.1% | 23.5% | 14.2% | 21.1% | 13.2% |
| G70 | Solcor | Solcor Plaza | 1244, 11,195, #74 | 18.3 | M | 56.7 | 42.5 | 15.9 | 3.1 | -13.1 | 11.6 | -8.0 | 5.1 | 25.9 | 14.3% | 25.4% | 10.0% | 21.5% | 13.5% | 27.2% | 13.4% |
| K68 | Solcor | Solcor Plaza | 1246 (11353), #328 | 20.3 | F | 44.4 | 44.7 | 16.1 | 3.2 | -13.5 | 11.7 | -7.1 | 6.4 | 23.3 | 15.9% | 34.5% | 12.3% | 23.3% | 13.9% | 18.9% | 12.4% |
| G75 | Solcor | Solcor Plaza | 1286, 11,202, #79 | 10.3 | M | 62.8 | 37.9 | 14.1 | 3.1 | -14.3 | 12.4 | -10.3 | 4.0 | 37.5 | 17.0% | 13.8% | 8.3% | 20.8% | 13.6% | 28.0% | 13.0% |
| K66 | Solcor | Solcor Plaza | 1317 (11215), #326 | 19.5 | | 34.6 | 43.7 | 15.8 | 3.2 | -12.9 | 11.5 | -7.2 | 5.7 | 20.2 | 11.6% | 33.0% | 12.1% | 19.6% | 11.3% | 27.3% | 14.8% |
| G71 | Solcor | Solcor Plaza | 1377, 11,365, #75 | 19.1 | M | 53.1 | 42.0 | 15.8 | 3.1 | -12.4 | 13.0 | -7.5 | 4.9 | 20.0 | 12.9% | 36.0% | 12.6% | 14.9% | 10.4% | 29.1% | 15.2% |
| K67 | Solcor | Solcor Plaza | 1379 (11220), #327 | 20.3 | M | 37.3 | 44.1 | 16.0 | 3.2 | -12.0 | 12.4 | -5.5 | 6.5 | 15.8 | 11.1% | 35.7% | 14.6% | 14.6% | 9.5% | 33.8% | 15.9% |
| G69 | Solcor | Solcor Plaza | 1381, 11,356, #73 | 13.8 | | 40.4 | 38.9 | 14.4 | 3.2 | -13.3 | 12.7 | -7.4 | 5.8 | 23.8 | 13.2% | 25.8% | 14.2% | 18.7% | 12.4% | 31.7% | 16.4% |
| G74 | Solcor | Solcor Plaza | 1386, 11,379, #78 | 16.8 | F | 40.4 | 41.5 | 15.6 | 3.1 | -11.3 | 13.4 | -5.7 | 5.6 | 16.4 | 11.0% | 40.0% | 14.4% | 13.0% | 8.9% | 30.6% | 16.2% |
| G76 | Solcor | Solcor Plaza | 1391, 11,218, #80 | 21.4 | | 38.0 | 42.6 | 16.0 | 3.1 | -13.8 | 14.5 | -9.0 | 4.8 | 24.6 | 14.6% | 22.3% | 10.3% | 19.7% | 13.8% | 33.3% | 14.7% |
| G72 | Solcor | Solcor Plaza | 1394, 11,211, #76 | 16.4 | F | 54.0 | 42.0 | 15.7 | 3.1 | -13.8 | 11.9 | -8.9 | 4.9 | 28.2 | 14.8% | 23.5% | 9.6% | 23.8% | 13.9% | 24.4% | 14.3% |
| G77 | Solcor | Solcor Plaza | 5093, 11,385, #81 | 15.0 | F | 51.3 | 38.7 | 14.4 | 3.1 | -13.6 | 10.8 | -8.6 | 4.9 | 28.7 | 16.1% | 24.2% | 12.7% | 24.5% | 16.0% | 22.7% | 14.5% |
| G68 | Solcor | Solcor Plaza | 629, 11,367, #72 | 19.8 | F | 45.4 | 42.9 | 16.0 | 3.1 | -14.0 | 11.3 | -8.8 | 5.2 | 27.3 | 13.8% | 24.2% | 11.2% | 27.7% | 14.6% | 20.8% | 14.3% |
| L55 | Solcor | Solcor 3 | T. 103, 3599 | 18.3 | M | 52.4 | 45.3 | 15.7 | 3.4 | -12.7 | 11.3 | -6.5 | 6.2 | 17.0 | 11.1% | 36.2% | 12.0% | 20.7% | 11.9% | 26.0% | 14.3% |
| L64 | Solcor | Solcor 3 | T. 106, 13177 | 19.4 | F | 53.8 | 43.2 | 15.7 | 3.2 | -14.7 | 10.5 | -9.5 | 5.2 | 25.8 | 16.9% | 19.3% | 10.6% | 28.6% | 16.1% | 26.3% | 12.2% |

*(Continued)*

**Table 1.** (Continued)

| | | | | | | | | | | | | | | | | | | | | | |
|---|---|---|---|---|---|---|---|---|---|---|---|---|---|---|---|---|---|---|---|---|---|
| L72 | Solcor | Solcor 3 | T. 111, 3604 | M | 19.7 | 52.3 | 42.2 | 15.1 | 3.3 | -12.7 | 12.5 | -7.2 | 5.5 | 26.8 | 14.6 | 29.7 | 12.2 | 16.8 | 11.4 | 26.8 | 14.4% |
| L63 | Solcor | Solcor 3 | T. 111, 3605 | F | 20.7 | 52.0 | 43.7 | 15.6 | 3.3 | -13.1 | 12.0 | -7.3 | 5.7 | 21.2 | 13.7 | 32.6 | 15.1 | 19.3 | 10.8 | 26.9 | 17.3% |
| L66 | Solcor | Solcor 3 | T. 111, 3606 | F | 18.2 | 51.5 | 47.9 | 17.1 | 3.3 | -13.5 | 11.8 | -8.1 | 5.3 | 24.3 | 16.7 | 23.5 | 13.4 | 23.8 | 14.4 | 28.4 | 15.7% |
| L57 | Solcor | Solcor 3 | T. 113, 13120 | F | 23.2 | 46.1 | 41.2 | 14.3 | 3.4 | -14.2 | 10.8 | -9.1 | 5.1 | 26.6 | 15.5 | 23.7 | 8.5 | 29.3 | 15.2 | 20.4 | 12.8% |
| L67 | Solcor | Solcor 3 | T. 115, 3609 | F | 22.8 | 41.2 | 52.4 | 18.9 | 3.2 | -15.0 | 11.0 | -9.4 | 5.6 | 25.9 | 13.9 | 17.0 | 8.9 | 27.0 | 13.7 | 30.2 | 13.0% |
| L62 | Solcor | Solcor 3 | T. 115, 3610 | F | 5.7 | 53.3 | 36.8 | 12.4 | 3.5 | -14.6 | 13.4 | -8.9 | 5.7 | 34.2 | 16.1 | 22.1 | 9.9 | 18.8 | 13.1 | 24.9 | 14.7% |
| L58 | Solcor | Solcor 3 | T. 115, 3611 | F | 21.0 | 49.3 | 44.4 | 15.7 | 3.3 | -14.1 | 10.5 | -9.6 | 4.5 | 29.9 | 16.0 | 19.4 | 9.2 | 27.4 | 15.2 | 23.3 | 11.9% |
| L74 | Solcor | Solcor 3 | T. 116, 13126 | M | 20.3 | 46.3 | 44.9 | 16.1 | 3.3 | -16.1 | 11.2 | -11.3 | 4.8 | 43.3 | 18.1 | 11.8 | 6.9 | 25.0 | 15.6 | 20.0 | 11.3% |
| L78 | Solcor | Solcor 3 | T. 126 | F | 15.8 | 44.8 | 42.9 | 15.3 | 3.3 | -14.0 | 10.4 | -8.7 | 5.4 | 26.9 | 17.2 | 25.0 | 11.1 | 26.4 | 13.8 | 21.7 | 11.6% |
| L56 | Solcor | Solcor 3 | T. 139 indiv. 1 | | 14.7 | 48.9 | 42.3 | 15.8 | 3.1 | -16.1 | 10.9 | -10.2 | 5.9 | 33.4 | 22.9 | 12.4 | 8.4 | 39.1 | 18.8 | 15.0 | 10.8% |
| L60 | Solcor | Solcor 3 | T. 27, 1628 | F | 20.3 | 54.2 | 46.0 | 16.2 | 3.3 | -15.9 | 11.0 | -10.6 | 5.3 | 29.7 | 17.8 | 15.6 | 8.6 | 34.3 | 18.4 | 20.4 | 12.0% |
| L68 | Solcor | Solcor 3 | T. 29, 1666 | M | 20.2 | 54.4 | 41.3 | 14.7 | 3.3 | -15.7 | 10.7 | -10.6 | 5.1 | 35.2 | 18.8 | 11.9 | 7.5 | 30.0 | 19.0 | 22.9 | 13.9% |
| L59 | Solcor | Solcor 3 | T. 30, 1683 | F | 18.8 | 50.9 | 47.4 | 16.9 | 3.3 | -15.3 | 10.5 | -10.0 | 5.3 | 25.7 | 15.5 | 17.5 | 10.9 | 29.6 | 15.6 | 27.2 | 14.6% |
| L75 | Solcor | Solcor 3 | T. 6, 1080 | F | 17.7 | 44.1 | 37.2 | 13.2 | 3.3 | -14.9 | 11.7 | -11.0 | 3.9 | 32.4 | 20.4 | 14.2 | 8.9 | 31.1 | 18.1 | 22.3 | 13.0% |
| L65 | Solcor | Solcor 3 | T. 75, 2607 | F | 19.3 | 53.0 | 43.9 | 15.8 | 3.2 | -15.3 | 11.2 | -9.8 | 5.5 | 32.2 | 20.0 | 18.5 | 9.6 | 27.4 | 17.5 | 21.9 | 14.3% |
| L73 | Solcor | Solcor 3 | T. 78, 2699 | M | 5.2 | 53.7 | 40.2 | 13.9 | 3.4 | -14.4 | 11.2 | -8.8 | 5.6 | 18.2 | 12.9 | 25.0 | 9.7 | 29.4 | 14.3 | 27.4 | 14.2% |
| L77 | Solcor | Solcor 3 | T. 79, 2762 | F | 8.3 | 54.0 | 38.6 | 13.8 | 3.3 | -14.7 | 11.8 | -9.4 | 5.3 | 29.8 | 17.7 | 21.6 | 10.7 | 25.5 | 15.0 | 23.1 | 13.0% |
| L76 | Solcor | Solcor 3 | T. 98, 3593a | F | 16.6 | 38.4 | 41.6 | 14.7 | 3.3 | -12.6 | 12.2 | -7.5 | 5.1 | 26.3 | 12.6 | 32.4 | 13.0 | 17.4 | 12.2 | 23.8 | 14.8% |
| G91 | Solcor | Solcor 3 | T101, 3597, #190 | M | 19.1 | 49.1 | 42.2 | 15.3 | 3.2 | -11.5 | 10.9 | -6.6 | 4.9 | 16.6 | 12.5 | 37.2 | 13.5 | 19.2 | 11.2 | 27.0 | 16.4% |
| H12 | Solcor | Solcor 3 | T107, 13.118, #185 | M | 20.1 | 45.1 | 42.7 | 15.4 | 3.2 | -13.9 | 11.5 | -8.7 | 5.2 | 27.2 | 15.1 | 27.2 | 14.0 | 23.9 | 13.6 | 21.7 | 13.9% |
| G88 | Solcor | Solcor 3 | T112, 13.111, #183 | M | 18.8 | 44.2 | 42.5 | 15.4 | 3.2 | -13.9 | 11.4 | -9.2 | 4.7 | 30.7 | 19.0 | 21.7 | 11.2 | 22.3 | 15.3 | 25.3 | 15.1% |
| G83 | Solcor | Solcor 3 | T117, 13.156, #208 | M | 19.7 | 45.7 | 44.4 | 16.1 | 3.2 | -13.3 | 11.0 | -7.6 | 5.7 | 15.9 | 11.6 | 32.0 | 12.9 | 24.0 | 13.1 | 28.1 | 15.3% |
| H13 | Solcor | Solcor 3 | T132, s/n, #188 | M | 18.4 | 44.7 | 43.4 | 15.6 | 3.2 | -11.5 | 10.9 | -5.8 | 5.7 | 18.0 | 10.2 | 36.1 | 15.3 | 17.3 | 10.2 | 28.7 | 17.5% |
| G89 | Solcor | Solcor 3 | T16, 3061, #177 | F | 21.6 | 42.7 | 44.7 | 16.2 | 3.2 | -15.5 | 11.7 | -10.6 | 4.9 | 34.1 | 19.8 | 16.0 | 9.0 | 29.0 | 17.0 | 20.9 | 12.7% |
| H7 | Solcor | Solcor 3 | T27, 1629, #191 | M | 20.4 | 54.9 | 42.6 | 15.4 | 3.2 | -15.1 | 12.0 | -11.3 | 3.9 | 41.0 | 18.2 | 11.8 | 7.0 | 25.3 | 19.4 | 21.9 | 12.3% |
| H9 | Solcor | Solcor 3 | T30, 1683, #192 | M | 21.4 | 48.3 | 43.5 | 15.9 | 3.2 | -14.7 | 10.6 | -9.9 | 4.8 | 33.0 | 18.9 | 17.7 | 10.2 | 29.7 | 18.2 | 19.7 | 14.1% |
| G90 | Solcor | Solcor 3 | T44, 1871, #186 | F | 21.9 | 36.4 | 37.4 | 12.4 | 3.5 | -11.2 | 15.0 | -7.9 | 3.3 | 23.0 | 12.7 | 30.1 | 14.8 | 14.1 | 10.3 | 32.8 | 16.6% |
| G85 | Solcor | Solcor 3 | T54, 2071, #189 | M | 4.3 | 55.2 | 30.4 | 10.8 | 3.3 | -16.5 | 9.6 | -11.6 | 4.9 | 28.7 | 18.8 | 11.2 | 7.6 | 42.5 | 20.7 | 17.6 | 11.9% |
| G86 | Solcor | Solcor 3 | T60, 2341, #182 | F | 20.5 | 48.3 | 43.9 | 15.9 | 3.2 | -14.6 | 11.5 | -9.5 | 5.1 | 29.7 | 15.7 | 19.4 | 10.1 | 26.7 | 15.2 | 24.2 | 13.7% |
| G84 | Solcor | Solcor 3 | T60, 2342, #184 | M | 17.6 | 52.5 | 42.6 | 15.5 | 3.2 | -15.9 | 9.7 | -11.9 | 3.9 | 32.5 | 19.8 | 10.2 | 6.7 | 36.8 | 18.8 | 20.6 | 11.5% |
| G82 | Solcor | Solcor 3 | T70, 2513, #181 | F | 20.5 | 50.9 | 44.3 | 16.0 | 3.2 | -14.4 | 11.4 | -9.0 | 5.4 | 27.1 | 15.1 | 18.5 | 9.6 | 26.3 | 14.7 | 28.2 | 13.0% |
| H10 | Solcor | Solcor 3 | T70, 2514, #180 | M | 17.3 | 52.0 | 44.3 | 16.1 | 3.2 | -14.5 | 12.8 | -10.5 | 4.0 | 35.7 | 18.3 | 11.0 | 7.9 | 25.8 | 15.8 | 27.6 | 14.6% |
| G81 | Solcor | Solcor 3 | T8, 1161A, #176 | F | 14.3 | 48.9 | 43.0 | 15.6 | 3.2 | -14.7 | 10.3 | -9.8 | 4.9 | 29.3 | 17.7 | 16.2 | 8.8 | 27.7 | 15.6 | 26.8 | 11.4% |
| H8 | Solcor | Solcor 3 | T93, 3.083, #187 | M | 1.5 | 70.6 | 17.4 | 5.7 | 3.6 | -11.4 | 12.1 | -5.8 | 5.5 | 15.7 | 10.6 | 41.8 | 14.1 | 20.3 | 11.9 | 22.2 | 15.8% |
| J81 | Tchecar | Tchecar | 680 | M | 15.3 | 50.3 | 43.6 | 15.7 | 3.2 | -14.0 | 10.7 | -8.8 | 5.2 | 24.1 | 15.0 | 24.4 | 12.9 | 28.1 | 13.9 | 23.4 | 16.0% |
| J80 | Tchecar | Tchecar | 686 | | 1.5 | 59.3 | 37.4 | 12.4 | 3.5 | -15.9 | 11.4 | -9.8 | 6.1 | 34.7 | 20.6 | 18.6 | 10.7 | 26.9 | 17.1 | 19.8 | 13.5% |
| J82 | Tchecar | Tchecar | 807 | | 20.4 | 39.0 | 44.5 | 16.1 | 3.2 | -13.4 | 14.2 | -9.0 | 4.4 | 33.1 | 16.6 | 17.2 | 8.9 | 16.9 | 13.3 | 32.7 | 14.7% |
| J79 | Tchecar | Tchecar | 815 | F | 17.7 | 39.3 | 42.1 | 15.1 | 3.3 | -14.3 | 11.2 | -9.6 | 4.7 | 27.2 | 16.5 | 18.1 | 9.7 | 28.5 | 15.2 | 26.2 | 17.0% |
| J70 | Tchecar | Tchecar | 1158 | | 9.7 | 54.3 | 42.8 | 15.2 | 3.3 | -13.9 | 12.0 | -7.0 | 6.8 | 23.0 | 12.5 | 31.1 | 13.7 | 17.2 | 10.7 | 28.7 | 16.0% |
| J88 | Tchecar | Tchecar | 1105, 11257, #136 | F | 15.6 | 61.7 | 25.8 | 8.9 | 3.4 | -13.8 | 11.7 | -7.7 | 6.1 | 26.6 | 16.3 | 29.2 | 14.3 | 20.3 | 12.2 | 23.8 | 17.0% |
| J81 | Tchecar | Tchecar | 1151 (11.130) #129 | M | 16.8 | 37.6 | 43.4 | 15.8 | 3.2 | -13.9 | 11.0 | -9.5 | 4.4 | 27.1 | 16.7 | 21.6 | 10.6 | 25.6 | 13.9 | 25.7 | 12.4% |
| J90 | Tchecar | Tchecar | 1155, 11304, #140 | F | 9.1 | 50.1 | 40.6 | 14.4 | 3.3 | -14.5 | 11.3 | -8.8 | 5.7 | 31.8 | 16.4 | 24.1 | 12.0 | 21.9 | 14.9 | 22.1 | 13.6% |
| J89 | Tchecar | Tchecar | 1158, S/N, #139 | | 9.9 | 47.5 | 39.7 | 14.0 | 3.3 | -14.2 | 12.2 | -7.8 | 6.4 | 23.9 | 15.1 | 28.1 | 12.0 | 23.9 | 16.8 | 24.1 | 14.6% |
| J84 | Tchecar | Tchecar | 1161 (11.247) #132 | F | 14.3 | 55.2 | 40.9 | 14.8 | 3.2 | -12.9 | 12.0 | -6.4 | 6.6 | 15.9 | 12.2 | 37.3 | 10.4 | 21.8 | 12.5 | 25.1 | 14.6% |
| J79 | Tchecar | Tchecar | 1173, 11283, #127 | M | 18.1 | 62.6 | 39.1 | 13.9 | 3.3 | -14.7 | 12.0 | -9.3 | 5.4 | 31.6 | 18.9 | 17.2 | 9.6 | 25.0 | 15.6 | 26.3 | 15.7% |
| J87 | Tchecar | Tchecar | 1222, 11143, #135 | F | 21.3 | 49.6 | 41.0 | 14.7 | 3.2 | -14.6 | 11.8 | -9.1 | 5.4 | 35.5 | 18.7 | 19.8 | 11.1 | 22.3 | 16.3 | 22.4 | 11.6% |
| J94 | Tchecar | Tchecar | 666, 11308, #145 | | 20.1 | 40.9 | 43.1 | 15.6 | 3.2 | -14.3 | 11.7 | -8.6 | 5.7 | 22.8 | 13.2 | 23.8 | 11.5 | 25.1 | 13.3 | 28.2 | 15.5% |
| J92 | Tchecar | Tchecar | 678, 11160, #143 | F | 13.6 | 50.9 | 42.4 | 15.3 | 3.2 | -14.0 | 11.2 | -8.5 | 5.4 | 23.7 | 16.7 | 21.5 | 9.9 | 26.4 | 15.4 | 28.4 | 13.4% |
| J86 | Tchecar | Tchecar | 680, 1244, #134 | M | 12.0 | 53.5 | 40.8 | 14.7 | 3.2 | -13.9 | 10.8 | -9.3 | 4.5 | 27.0 | 16.0 | 15.4 | 9.2 | 28.7 | 15.2 | 28.9 | 13.8% |
| J95 | Tchecar | Tchecar | 686, 1189, #146 | | 3.1 | 55.9 | 32.7 | 11.0 | 3.5 | -17.8 | 9.0 | -10.5 | 7.4 | 30.7 | 19.0 | 14.0 | 8.6 | 41.2 | 17.3 | 14.1 | 10.6% |
| J80 | Tchecar | Tchecar | 687 (11.281) #128 | F | 18.4 | 37.5 | 42.2 | 15.3 | 3.2 | -14.4 | 12.2 | -8.4 | 6.0 | 24.1 | 14.6 | 22.5 | 10.9 | 25.4 | 13.3 | 28.0 | 13.8% |

(Continued)

**Table 1.** (Continued)

| | | | | | | | | | | | | | | | | | | | | | | |
|---|---|---|---|---|---|---|---|---|---|---|---|---|---|---|---|---|---|---|---|---|---|---|
| 193 | Tchecar | Tchecar | 691, 11499, #144 | M | 18.4 | 50.2 | 42.6 | 15.2 | 3.3 | -13.6 | 11.5 | -8.1 | 5.5 | 24.2% | 14.6% | 25.0% | 12.2% | 21.4% | 11.5% | 29.4% | 14.0% |
| 182 | Tchecar | Tchecar | 692 (11.123) #130 | | 17.8 | 40.6 | 42.7 | 15.4 | 3.2 | -13.9 | 11.2 | -8.1 | 5.8 | 26.3% | 18.7% | 25.7% | 9.8% | 29.7% | 14.3% | 18.2% | 11.9% |
| 198 | Tchecar | Tchecar | 694 (11.141) #137 | F | 13.4 | 41.9 | 43.5 | 15.8 | 3.2 | -13.6 | 11.6 | -8.9 | 4.8 | 27.4% | 16.9% | 19.3% | 10.0% | 22.7% | 14.0% | 30.6% | 14.4% |
| 185 | Tchecar | Tchecar | 801, 11324, #133 | | 21.7 | 51.0 | 41.6 | 14.8 | 3.3 | -13.6 | 12.2 | -8.4 | 5.1 | 24.3% | 13.9% | 18.6% | 9.3% | 21.5% | 14.2% | 35.6% | 12.9% |
| 183 | Tchecar | Tchecar | 814 (11.106) #131 | M | 17.6 | 44.2 | 43.4 | 15.6 | 3.3 | -15.6 | 10.4 | -9.6 | 6.0 | 23.6% | 16.5% | 15.8% | 8.7% | 36.9% | 14.6% | 23.7% | 14.1% |
| 196 | Tchecar | Tchecar | 815, 11307, #147 | F | 17.0 | 42.2 | 41.8 | 15.1 | 3.2 | -14.1 | 11.2 | -9.1 | 5.1 | 25.1% | 16.4% | 21.2% | 9.7% | 28.7% | 15.8% | 25.0% | 17.5% |
| 197 | Tchecar | Tchecar | 824 (11.263) #138 | | 12.3 | 42.7 | 43.2 | 15.5 | 3.3 | -13.2 | 10.9 | -8.1 | 5.1 | 25.8% | 15.4% | 29.2% | 11.8% | 23.5% | 14.2% | 21.5% | 14.5% |
| 191 | Tchecar | Tchecar | 844, 11116, #141 | | 19.3 | 47.1 | 42.2 | 15.2 | 3.2 | -10.3 | 11.6 | -5.5 | 4.9 | 13.1% | 9.2% | 44.6% | 13.1% | 15.3% | 11.2% | 27.0% | 14.2% |
| 178 | Tchecar | Tchecar | 863, 1177, #142 | F | 19.7 | 39.4 | 41.3 | 15.1 | 3.2 | -14.4 | 11.6 | -8.2 | 6.2 | 27.6% | 14.8% | 25.6% | 11.0% | 21.8% | 13.5% | 25.0% | 16.6% |

pH through a repeated process of centrifugation, decanting, and the addition of distilled water. Samples that had not yet been completely demineralized after 24 h had their acid changed and were returned to the rotator for a further 24 h period. Humic treatment consisted of adding 30 mL of 0.0625 M NaOH to each sample for a period of 20 h. After this time elapsed, the samples were again rinsed to neutral pH by the repeated application of distilled water, centrifugation, and decanting. The remaining "collagen" was then gelatinized in $10^{-3}$ M HCL at 90˚C, and filtered using single-use 40 μm Millipore Steriflip® vacuum filters, condensed, frozen, and freeze dried. Start and end weights were recorded (to the nearest 0.1 mg) and were used to calculate collagen yield (wt%) for each sample.

Hydroxyapatite extraction followed a protocol first established in Lee-Thorp [71] and Krueger [72] and modified by Pestle [70]. Approximately 0.1 g (weighed to the nearest 0.1 mg) of the 0.125–0.25 mm fraction was placed in a 50 mL centrifuge tube. After weighing, each sample underwent a 24 h oxidation of organics using 30 mL of 50% bleach (~2.5% NaOCl). The bleach treatment was then repeated, with fresh bleach solution, for an additional 24 h period, for a total of 48 h of treatment. Samples then were rinsed to neutral pH through a repeated process of centrifugation, decanting, and addition of distilled water. Next, labile carbonates were removed by the addition of 30 mL of 0.1 M acetic acid to each centrifuge tube for a total of four hours, with a 5 min vacuum treatment after 2 h. After the acid treatment, each sample was rinsed to neutral pH before being placed in a 50˚C oven overnight to dry the resulting product. Start and end weights were recorded for all hydroxyapatite samples (weighed to the nearest 0.1 mg) and used to calculate the hydroxyapatite yield (wt%).

Collagen and hydroxyapatite isotopic analysis was performed in the Marine Geology and Geophysics Stable Isotope Laboratory at the Rosenstiel School of Marine and Atmospheric Science, University of Miami. Collagen samples were packed into tin capsules and analyzed using a PDZ Europa ANCA-GSL elemental analyzer interfaced to a PDZ Europa 20–20 isotope ratio mass spectrometer (IRMS). This analytical process yields information on elemental carbon and nitrogen composition as well as relative abundance of the stable isotopes of carbon and nitrogen, data which are used in the generation of $\delta^{13}C_{co}$ and $\delta^{15}N_{co}$. Hydroxyapatite samples were analyzed using a Kiel-IV Carbonate Device coupled to a Thermo Finnigan DeltaPlus IRMA, providing $\delta^{13}C_{ap}$ values. Collagen results were calibrated using acetanilide and glycine, and an OCC (optically clear calcite) standard calibrated to NBS-19 was used for hydroxyapatite. Standards were analyzed in every sample set at the beginning and end of the run, as well as in-between the analyzed samples to ensure instrumental stability. Check samples of all three standards were also run as unknowns in every run to verify measurement accuracy [73]. Precision (as determined by replicate analysis of samples included in the present study) averaged 0.1‰ for $\delta^{13}C_{co}$, 0.2‰ for $\delta^{15}N_{co}$, and 0.1‰ for $\delta^{13}C_{ap}$.

As all samples considered herein were extracted and analyzed using the same protocols and instruments, the issue of inter-laboratory differences in resulting stable isotope signatures [74–76] are not of concern.

**Foodweb sampling.** Paleodietary reconstruction requires knowledge of the isotopic composition of both consumers and potential foodstuffs. Thus, in addition to human bone samples, we also built a representative sample of the plants and animals that make up the local foodweb. To make the foodweb appropriate for the reconstruction of ancient dietary practices in San Pedro de Atacama we targeted, inasmuch as possible, archaeological flora and fauna from sites in the region, which would better represent the local foodweb available to the human consumers of interest here. As necessary, however, the archaeological foodweb was augmented through the collection of modern floral and faunal samples, again restricted to the immediate environs of San Pedro de Atacama. Ultimately, this foodweb comprised sixty-two samples representing four distinct groupings: beans, $C_3$ plants, $C_4$/CAM plants, and terrestrial

fauna. Unpublished values for six camelid samples, analyzed and graciously provided by Francisca Santana Sagredo were included in the average values for the terrestrial faunal grouping, but are not detailed individually here. As with human bone samples, permits for foodweb sample export were granted by the Chilean *Consejo de Monumentos Nacionales* (see permit numbers above).

**Modeling.** The Bayesian model known as FRUITS (Food Reconstruction Using Isotopic Transferred Signals; [56]) was used to quantify individual dietary composition. This multi-source mixture modeling technique is one of several developed with the goal of better bounding estimates of food source contribution and, as such, offers clear advantages over both classical approaches to paleodietary reconstruction and other model-based approaches. While such an approach offers notable advantages (in that it permits statistical testing of multiple hypotheses, does not rely on ad hoc explanations of observed dietary differences, and fully accounts for the many sources of error in paleodietary reconstruction) it is far from a panacea. Indeed, such models require user stipulation of a large number of parameters, many of which can be difficult to obtain/estimate. Model outputs are only as good as the data upon which any model iteration is provided built, and while such models generate mathematically viable solutions, any model is invariably specific to the assumptions informing it, while their reliability to broader scenarios are frequently not straightforward or easy to assess. As such, their use requires detailed knowledge of both model mechanics and the many parameters needed for their generation. Therefore it is central to the defense of any model that the specifics parameters used are detailed, and the following paragraphs present the parameters used for the present application.

Only well-preserved consumer samples (human bone; collagen yield >0.5 wt%, carbon yield >4.5 wt%, nitrogen yield >0.9 wt%, atomic C/N ratio between 2.9–3.6) were included in FRUITS modelling. The consumer data for the model were based on the isotope ratios generated by IRMS. To account for fractionation, we first determined the consumer-foodstuff offset (and error) for $\delta^{13}C_{co}$ using the method of [77], which takes into account both the measured $\delta^{13}C_{co}$ and the spacing value ($\Delta^{13}C_{ap-co}$). Fractionation of $\delta^{13}C_{ap}$ was stipulated as 10.1±0.4‰, following Fernandes et al. [78]. Finally, for $\delta^{15}N_{co}$ we employed a trophic fractionation value of 3.6±1.2‰, as recommended by several experimental studies of omnivorous animals [79–84].

Foodweb isotope values comprised the edible portions of the sixty-two samples discussed above. Any modern data included in this reference sample had $\delta^{13}C$ values corrected by +1.5‰ to account for recent fossil fuel burning [85]. Furthermore, the $\delta^{13}C$ value of bone samples were adjusted by -2.0‰ to account for bone collagen-edible tissue offset [86, 87]. Macronutrient (protein, carbohydrate, lipid) composition of each food group was determined by reference to the USDA National Nutrient Database for Standard Reference [88]. Elemental composition (particularly %C) of each foodstuff/macronutrient group was based on formulae provided in Morrison and colleagues [89], and differences in digestibility were accounted for following Hopkins [90].

To account for differential routing, all nitrogen in bone collagen was stipulated as coming from dietary protein, the carbon in hydroxyapatite was stipulated as reflecting all dietary carbon, and the carbon composition of bone collagen was set as reflecting a 3:1 ratio of dietary protein to energy [78]. Carbon isotope offsets between measured bulk food isotope values and the isotopic values of a foodstuff's fats (bulk $\delta^{13}C$-6‰) and carbohydrates (bulk $\delta^{13}C$+0.5‰) were based on data from Tieszen [91]. The carbon isotope signature of a measured bulk foodstuff's protein was determined using a mass-balance equation, such that a proportional/weighted average of the $\delta^{13}C$ of protein and energy (fats and carbohydrates) would equal the measured $\delta^{13}C$ bulk value (corrected for the concentration of carbon in each macronutrient and foodstuff-appropriate macronutrient concentration).

For purposes of this analysis, we divided the available foodstuffs into four groups reflecting taxonomy and photosynthetic pathway (beans, $C_3$ plants, $C_4$/CAM plants [which, in spite of photosynthetic differences, are inseparable isotopically], and terrestrial fauna). Some of the local $C_3$ plants, in particular leguminous trees of the family *Fabaceae* (which includes key local taxa such as *algarrobo* [*Prosopis* spp.] and *chañar* [*Geoffroea decorticans*]) but also the *Cyperaceae*, can exhibit nitrogen-fixing behaviors resulting in very low $\delta^{15}N$ values. Given, however, that these N-fixing behaviors would appear to be related to the age of the tree(s) in question (at least for *Fabaceae*), with more mature trees yielding much higher $\delta^{15}N$ values [92, 93], and as we see a huge range in $\delta^{15}N$ values in our samples of these taxa (from 1.1–15.1‰), we have chosen to include such taxa with the other $C_3$ plants, leaving beans proper (*frijoles/porotos*, with high protein content and very low $\delta^{15}N$ values) in a distinct category.

Consumption of protein was limited to between 10 and 50% of protein as energy (using the FRUITS *a priori* data option), reflecting the lower and upper limit of possible human protein intake [94]. All FRUITS simulations were performed using 10,000 iterations, as recommended by the program's developers.

Given differences in foodweb composition, we note that the results of the modeling presented here are different from, and not directly comparable to, previous dietary modeling we have presented for the region (e.g., [60, 61]).

**Dating.** Temporal control was obtained through a rigorous program of direct radiocarbon dating of bone collagen. Selection and collection of samples for dating were performed concurrently with osteological assessment of remains and the collection of samples for stable isotope analyses. AMS dating for the majority of these samples was performed at the National Science Foundation-Accelerator Mass Spectrometry Laboratory at the University of Arizona (n = 145), with the remaining twenty-nine samples dated by Beta Analytic of Miami, Florida, USA. All told, our sample included 167 individuals (65%) for whom we possessed both isotope values and direct AMS dates. The temporal structure of the dates used in our statistical considerations below are based on Bayesian modeling of available dates conducted at two scales (all *ayllus* and inter-*ayllu*), as fully detailed in [42].

**Statistical analysis.** All statistical analyses were performed in R [95], and, unless otherwise noted, nonparametric methods were employed. An alpha of 0.05 was employed for all analyses.

Correlations were assessed using Spearman's rank correlation coefficient ($\rho$), the nonparametric equivalent of Pearson's r, which has the added capability of assessing the strength of non-linear bivariate correlations. Bootstrapped Spearman's correlation analysis was performed using an R function (UncertaintyCorrelations) to address uncertainty in the calculation of correlations. The function was written by one of the authors (MH) and the script is provided in S1 File. The use of a bootstrapping approach allows for full accounting of uncertainty in both dates and modeled foodstuff contributions, a step that is not taken often enough in such diachronic analyses. Two-sample comparisons of variances were made using Levene's test for the equality of variance, and we tested paired differences using the Wilcoxon signed-rank method, the nonparametric version of the more familiar Student's t-test. Comparisons of multiple group means was accomplished using the Kruskal-Wallis one-way test of variance instead of ANOVA, with *post-hoc* pairwise comparisons made using the Wilcoxon signed-rank test.

In several instances, it was judged appropriate to remove the possibly confounding effect of temporal differences among individuals and groups such that dietary effects of differences in *ayllu* and/or sex could be assessed independently of time. In those cases, we regressed (linearly) the variables of interest (dietary contributions by food group) against median modeled calibrated dates and captured the residuals (by food group) for subsequent analysis. Both the original food group contributions and these residual values were visualized using Metric

Multidimensional Scaling [96] (using the function cmdscale from R(stats)), which permitted us to envision the similarities among groups as "distances" in Cartesian space [97].

## Results

The results of our analyses are presented in five parts. We begin by considering the quality of preservation of the samples under analysis. Next, we present a summary of the human (consumer) isotopic values, which is followed by a rendering of the isoscape of the local foodweb. Subsequently, we discuss the workings of the model by means of analysis of the relationship of model inputs and outputs. Finally, we turn to the primary focus of this analysis, the results of the FRUITS dietary modeling, with considerations of differences and trends on diachronic, all *ayllu*, inter-*ayllu*, and intra-*ayllu* scales. For the reasons stipulated above, we believe that these modeled results are significantly more useful and meaningful than raw isotope (delta) values.

### Quality control

Of the 288 human bones samples analyzed, 257 (89.2%) had sample quality values above accepted cut-offs or within acceptable ranges (collagen yield >0.5 wt%, carbon yield >4.5 wt %, nitrogen yield >0.9 wt%, atomic C/N ratio between 2.9–3.6) commonly employed in archaeological studies [98–100]. It should be noted that for two samples (I-32 and I-62), the elemental analyzer failed to generate elemental yields, and that, lacking those data, atomic C:N ratios also could not be calculated. Based on the high collagen yields of these individuals, they were nonetheless included in later analysis and discussion. Individual data are to be found in Table 1. The summary of the sample quality data is presented in Table 2 and only the 257 samples meeting these stringent quality standards are considered/discussed henceforth. On the whole, the hyperarid conditions of the Atacama lend themselves to excellent preservation of buried organic materials (human skeletons included), and the integrity of the samples in this case is consistent with this. As has been previously observed [60, 101], the only notable exception to this pristine preservation are those individuals from cemeteries located very close to water sources (e.g., Quitor 8), which seem to have been subject to frequent inundation.

Examination of the relationship between sample age and the state of preservation produced heartening results, as the negative consequences of differential preservation would seem to be minimal. As seen in Fig 2, for those 167 individuals for whom we possess both radiocarbon dates and isotopic data, there were no statistically significant correlations between the modeled median calibrated radiocarbon dates and collagen yield, apatite yield, carbon or nitrogen concentration, or atomic C:N ratio. Furthermore, as seen in Fig 3, there are no significant correlations among any of the sample quality variables (collagen yield, apatite yield, wt% C, wt% N, and atomic C:N) and the primary isotopic values of the samples ($\delta^{13}C_{co}$, $\delta^{15}N_{co}$, and $\delta^{13}C_{ap}$).

While there are weak, but still significant, correlations between collagen yield and $\Delta^{13}C_{ap-co}$, on the one hand ($\rho$ = -0.14, p = 0.03, n = 257), and $\Delta^{13}C_{ap-co}$ and atomic C:N on the other ($\rho$ = -0.15, p = 0.02, n = 256), as $\Delta^{13}C_{ap-co}$ is a value derived from two primary isotope values

**Table 2. Summary of sample preservation data for samples included in FRUITS paleodietary modeling.**

|  | n | Mean | sd | Min | Max |
|---|---|---|---|---|---|
| Collagen yield (wt%) | 257 | 14.9 | 6.1 | 1.3 | 24.1 |
| Apatite yield (wt%) | 257 | 48.7 | 9.1 | 17.6 | 74.0 |
| wt% Carbon | 255 | 39.1 | 6.6 | 10.9 | 52.4 |
| wt% Nitrogen | 255 | 14.0 | 2.4 | 3.5 | 18.9 |
| Atomic C:N | 255 | 3.3 | 0.1 | 3.0 | 3.6 |

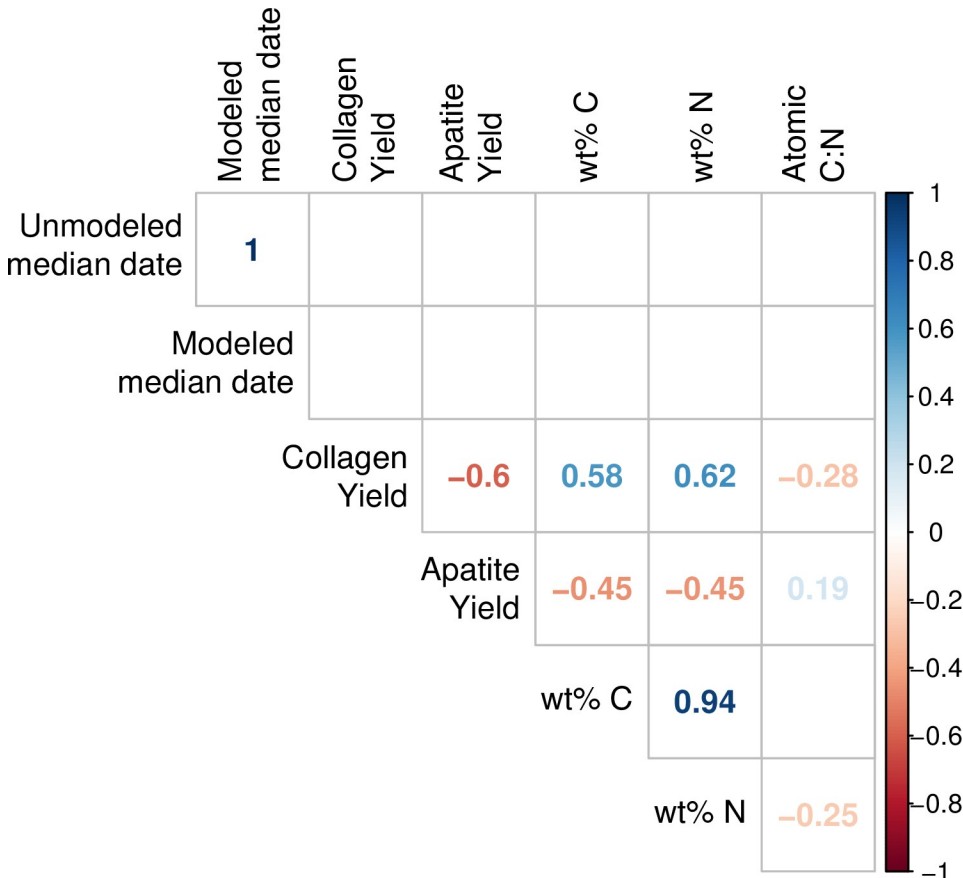

**Fig 2. Correlation matrix of radiocarbon dates and sample quality metrics.**

($\delta^{13}C_{co}$ and $\delta^{13}C_{ap}$), these correlations are more likely a consequence of systematic relationships between $\delta^{13}C_{co}$ and $\delta^{13}C_{ap}$ than any real effect of preservation. The ultimate implication of these findings is that the diachronic trends in isotopic and modeled foodstuff values detailed below are not the consequence of differential preservation due to sample age, or the effects of differences in preservation on isotope values, but rather stem from *bona fide* differences in practices of consumption.

The remaining correlations presented in these matrices are as expected in terms of directionality and strength. For instance, concerning Fig 2, collagen and apatite yields are significantly negatively correlated ($\rho$ = -0.6, $p<0.01$, n = 257) and wt% C and wt% N are significantly positively correlated ($\rho$ = 0.94, $p<0.01$, n = 255). Similarly, as seen in Fig 3, the strong positive correlations between $\delta^{13}C_{ap}$ and $\delta^{13}C_{co}$ ($\rho$ = 0.88, $p<0.01$, n = 257) or $\delta^{13}C_{co}$ and $\delta^{15}N_{co}$ ($\rho$ = 0.5, $p<0.01$, n = 257) are in line with *a priori* understandings of isotope systematics and the nature of the local foodweb (see below). As such, none of these correlations present any cause for concern.

## Isotopes

While the primary focus of the present work is the FRUITS modeling of dietary makeup, rather than raw isotopic values, we briefly discuss those raw isotopic data below. All individual data for the three primary ($\delta^{13}C_{co}$, $\delta^{15}N_{co}$, and $\delta^{13}C_{ap}$) and one derived ($\Delta^{13}C_{ap-co}$) isotope systems are presented in Table 1, and summary data for the 257 well-preserved individuals

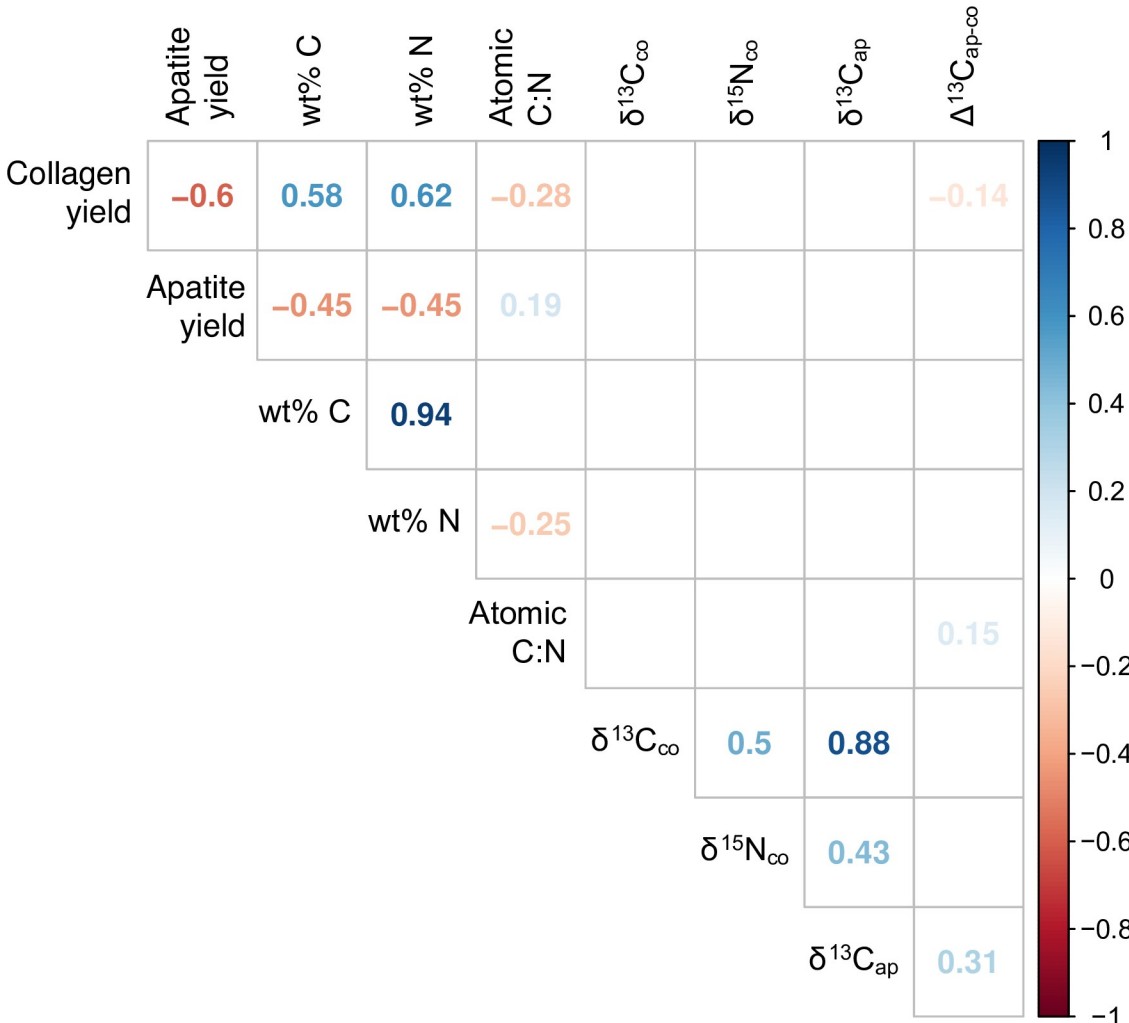

**Fig 3. Correlation matrix of sample quality metrics and measured isotope values.**

included in the present study are presented in Table 3. The fact that all of the primary isotope systems show ranges in excess of 8‰ speaks to substantial variability in dietary practices across time and space, variability that we contend patterns with a number of temporal, spatial, and bio-cultural differences.

As shown in Fig 3, $\delta^{13}C_{co}$, $\delta^{15}N_{co}$, and $\delta^{13}C_{ap}$ are all significantly positively correlated with one-another, as are $\delta^{13}C_{ap}$ and $\Delta^{13}C_{ap\text{-}co}$. Given the positively (but not significantly) correlated $\delta^{13}C$ and $\delta^{15}N$ values of the overall foodweb (see below), and the fact that both collagen and

**Table 3. Summary of isotopic data for bone collagen and hydroxyapatite for n = 257 individuals in FRUITS paleo-dietary modeling.**

|  | Mean | sd | Minimum | Maximum | Range |
|---|---|---|---|---|---|
| $\delta^{13}C_{co}$ (‰) | -14.0 | 1.5 | -17.8 | -9.6 | 8.2 |
| $\delta^{15}N_{co}$ (‰) | 11.3 | 1.1 | 7.5 | 16.1 | 8.6 |
| $\delta^{13}C_{ap}$ (‰) | -8.6 | 1.6 | -12.8 | -3.6 | 9.2 |
| $\Delta^{13}C_{ap\text{-}co}$ (‰) | 5.4 | 0.8 | 3.3 | 7.9 | 4.6 |

**Table 4. Summary of measured δ¹³C and δ¹⁵N of foodgroups used in FRUITS paleodietary modeling.**

| Grouping | n | δ¹³C | | | | δ¹⁵N | | | |
|---|---|---|---|---|---|---|---|---|---|
| | | Mean | sd | Minimum | Maximum | Mean | sd | Minimum | Maximum |
| Beans | 4 | -23.5 | 1.7 | -25.2 | -22.0 | 0.7 | 3.0 | -1.3 | 5.2 |
| C₃ plants | 17 | -23.7 | 2.0 | -27.0 | -20.1 | 8.6 | 5.4 | -1.2 | 16.5 |
| C₄/CAM plants | 13 | -11.2 | 1.6 | -13.5 | -8.9 | 12.0 | 5.6 | 4.6 | 20.8 |
| Terrestrial animals | 28 | -16.7 | 3.1 | -20.9 | -10.2 | 9.5 | 1.8 | 6.6 | 13.2 |

hydroxyapatite derive much of their carbon from shared sources, none of these relationships are particularly surprising.

## Foodweb

Summary stable isotope data for the sixty-two samples included in the local foodweb are presented in Table 4 and Fig 4, with individual values for fifty-six of the samples found in Table 5

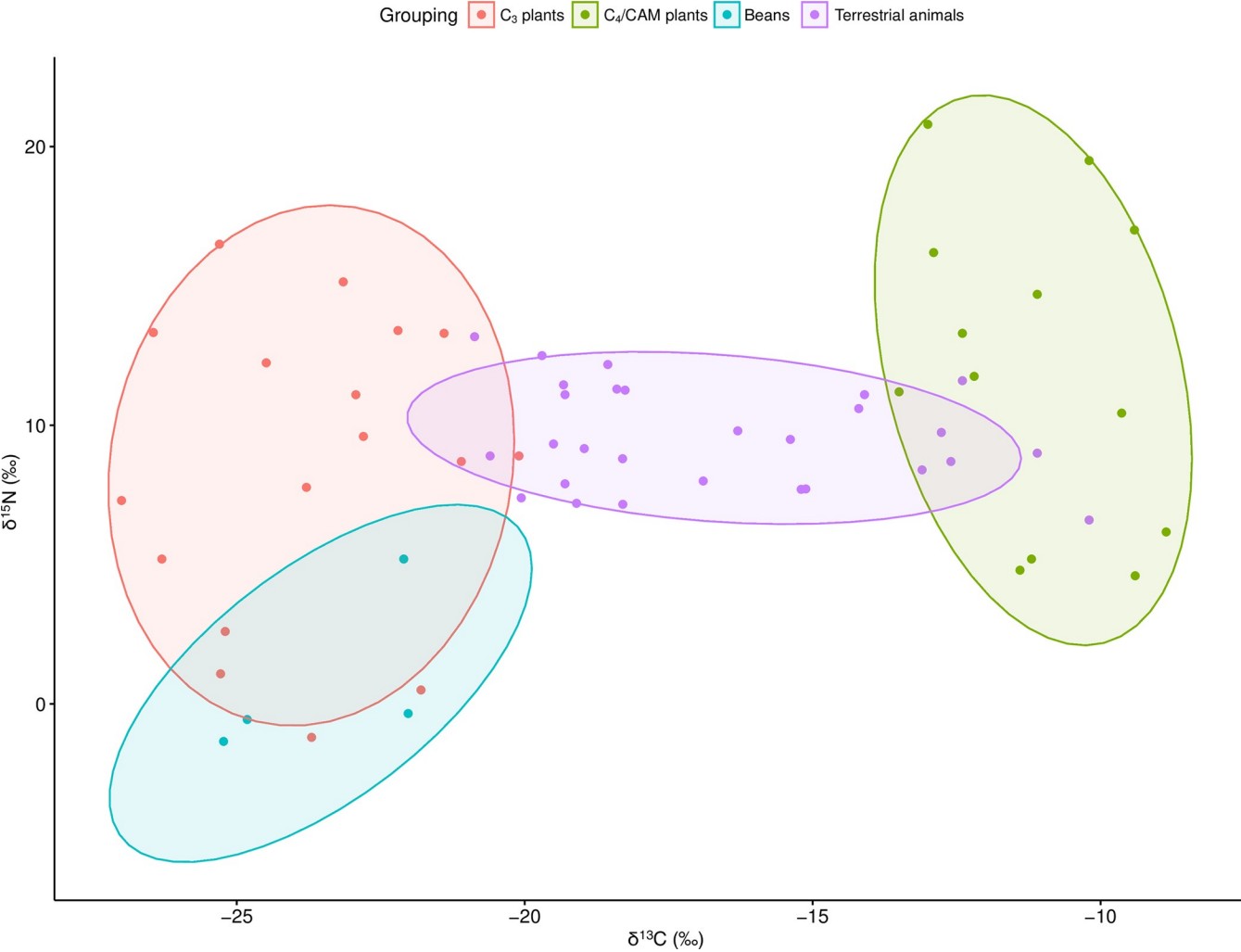

**Fig 4. Scatterplot of δ¹³C and δ¹⁵N of foodgroups used in FRUITS paleodietary modeling.**

**Table 5. Details of measured δ¹³C and δ¹⁵N of foodweb samples used in FRUITS paleodietary modeling.**

| Taxonomic name | Grouping | Ancient/ modern | Site | Tissue | $\delta^{13}$C (‰) | $\delta^{15}$N (‰) |
|---|---|---|---|---|---|---|
| Chenopodium quinoa | C₃ plant | Ancient | Topater | Seed | -26.4 | 13.3 |
| Chenopodium quinoa | C₃ plant | Ancient | Topater | Seed | -27.0 | 7.3 |
| Chenopodium quinoa | C₃ plant | Ancient | Topater | Seed | -26.3 | 5.2 |
| Cucurbitaceae | C₃ plant | Ancient | Solcor 3 | Rind | -24.5 | 12.2 |
| Cucurbitaceae | C₃ plant | Ancient | Solcor 3 | Rind | -25.3 | 16.5 |
| Cyperaceae | C₃ plant | Ancient | 02QUI89 | Rhizome | -23.7 | -1.2 |
| Geoffroea decorticans | C₃ plant | Modern | n/a | Seed | -25.3 | 1.1 |
| Prosopis alba | C₃ plant | Ancient | Topater | Seed | -20.1 | 8.9 |
| Prosopis chilensis | C₃ plant | Ancient | San Salvador | Seed | -21.1 | 8.7 |
| Prosopis chilensis | C₃ plant | Ancient | Solcor 3 | Seed | -23.8 | 7.8 |
| Prosopis chilensis | C₃ plant | Ancient | San Salvador | Seed | -23.2 | 15.1 |
| Prosopis chilensis | C₃ plant | Ancient | Quillagua | Seed | -22.9 | 11.1 |
| Prosopis chilensis | C₃ plant | Ancient | San Salvador | Wood | -22.8 | 9.6 |
| Prosopis chilensis | C₃ plant | Ancient | Quitor 6 Tardio | Seed | -21.4 | 13.3 |
| Prosopis chilensis | C₃ plant | Ancient | Solcor 3 | Seed | -22.2 | 13.4 |
| Prosopis tamarugo | C₃ plant | Modern | n/a | Seed | -21.8 | 0.5 |
| Tessaria absinthioides | C₃ plant | Ancient | San Salvador | Leaf | -25.2 | 2.6 |
| Zea mays | C₄/CAM plant | Ancient | Topater | Kernel | -10.2 | 19.5 |
| Zea mays | C₄/CAM plant | Ancient | Topater | Kernel | -9.4 | 4.6 |
| Zea mays | C₄/CAM plant | Ancient | Topater | Kernel | -11.2 | 5.2 |
| Zea mays | C₄/CAM plant | Ancient | Solcor 3 | Kernel | -12.4 | 13.3 |
| Zea mays | C₄/CAM plant | Ancient | Solcor 3 | Kernel | -12.9 | 16.2 |
| Zea mays | C₄/CAM plant | Ancient | Solcor 3 | Kernel | -13.0 | 20.8 |
| Zea mays | C₄/CAM plant | Ancient | 02QUI93 | Kernel | -11.1 | 14.7 |
| Zea mays | C₄/CAM plant | Ancient | Guatacondo | Kernel | -13.5 | 11.2 |
| Zea mays | C₄/CAM plant | Ancient | Solcor 3 | Kernel | -12.2 | 11.8 |
| Zea mays | C₄/CAM plant | Ancient | Quillagua | Kernel | -9.4 | 17.0 |
| Zea mays | C₄/CAM plant | Ancient | Quillagua | Kernel | -9.6 | 10.4 |
| Zea mays | C₄/CAM plant | Ancient | Quillagua | Kernel | -8.9 | 6.2 |
| Amaranthaceae | C₄/CAM plant | Ancient | Topater | Seed | -11.4 | 4.8 |
| Phaseolus lunatus | Legume | Ancient | Topater | Bean | -22.0 | -0.3 |
| Phaseolus sp. | Legume | Ancient | 02QUI93 | Bean | -22.1 | 5.2 |
| Phaseolus vulgaris | Legume | Ancient | Topater | Bean | -25.2 | -1.3 |
| Phaseolus vulgaris | Legume | Ancient | Topater | Bean | -24.8 | -0.6 |
| Lagidium sp. | Terrestrial | Ancient | RANL | Bone | -19.3 | 11.1 |
| Lagidium sp. | Terrestrial | Modern | n/a | Bone | -14.1 | 11.1 |
| Lama guanicoe | Terrestrial | Modern | n/a | Wool | -19.3 | 7.9 |

(*Continued*)

**Table 5.** (Continued)

| Taxonomic name | Grouping | Ancient/modern | Site | Tissue | $\delta^{13}$C (‰) | $\delta^{15}$N (‰) |
|---|---|---|---|---|---|---|
| Lama guanicoe | Terrestrial | Ancient | Hornitos 01 | Bone | -19.7 | 12.5 |
| Lama guanicoe | Terrestrial | Modern | n/a | Bone | -19.5 | 9.3 |
| Lama guanicoe | Terrestrial | Modern | n/a | Bone | -20.9 | 13.2 |
| Lama sp./Vicugna sp. | Terrestrial | Ancient | Solcor 3 | Bone | -18.3 | 8.8 |
| Lama sp./Vicugna sp. | Terrestrial | Ancient | Solcor 3 | Bone | -12.4 | 11.6 |
| Lama sp./Vicugna sp. | Terrestrial | Ancient | Solcor 3 | Bone | -20.6 | 8.9 |
| Lama sp./Vicugna sp. | Terrestrial | Ancient | San Salvador | Bone | -16.9 | 8.0 |
| Lama sp./Vicugna sp. | Terrestrial | Ancient | San Salvador | Bone | -15.2 | 7.7 |
| Lama sp./Vicugna sp. | Terrestrial | Ancient | San Salvador | Bone | -16.3 | 9.8 |
| Lama sp./Vicugna sp. | Terrestrial | Ancient | Quillagua | Bone | -20.1 | 7.4 |
| Lama sp./Vicugna sp. | Terrestrial | Ancient | VCH-19 | Bone | -15.1 | 7.7 |
| Lama sp./Vicugna sp. | Terrestrial | Ancient | Topater 1 | Bone | -19.3 | 11.5 |
| Lama sp./Vicugna sp. | Terrestrial | Ancient | RANL | Bone | -19.0 | 9.2 |
| Lama sp./Vicugna sp. | Terrestrial | Ancient | Yaye Corral de Toros | Bone | -12.8 | 9.7 |
| Lama sp./Vicugna sp. | Terrestrial | Ancient | Yaye Corral de Toros | Bone | -15.4 | 9.5 |
| Phoenicopterus chilensis | Terrestrial | Modern | n/a | Feather | -10.2 | 6.6 |
| Puma concolor | Terrestrial | Modern | n/a | Fur | -18.3 | 7.2 |
| Rhea sp. | Terrestrial | Modern | n/a | Bone | -18.3 | 11.3 |
| Rhea sp. | Terrestrial | Modern | n/a | Bone | -18.6 | 12.2 |

(see note above in Methods and Materials). Macronutrient concentrations and tissue-specific isotope data for these food groups are found in Table 6. Importantly, we note that the $\delta^{13}$C and $\delta^{15}$N values of the Atacama foodweb overall are not significantly correlated with one another ($\rho = 0.19$, $p = 0.14$, $n = 62$).

There is exclusivity in the $\delta^{13}$C signatures of the two energy-rich plant food groups ($C_3$ plants averaging -23.7±2.0‰ for $\delta^{13}$C versus -11.2±1.6‰ among $C_4$/CAM plants), and in the $\delta^{15}$N values of the two foods likely to have accounted for the majority of dietary protein (terrestrial animals, which have an average $\delta^{15}$N of 9.5±1.8‰, and beans, which average 0.7 ±3.0‰). Such exclusivity, even if it occurs in only one isotope system, is highly useful in attempts to determine the proportional contribution of these food groups to consumer diets. And while the overlapping isotopic values of certain groupings (e.g., beans and $C_3$ plants, which have almost completely overlapping $\delta^{13}$C values, with some greater separation in $\delta^{15}$N) may appear troubling for resolving foodstuff contributions, the inclusion of macronutrient

**Table 6. Macronutrient composition and macronutrient-adjusted $\delta^{13}$C and $\delta^{15}$N values of foodgroups used in FRUITS paleodietary modeling.**

| Food grouping | Macronutrient concentration (%) | | Tissue $\delta^{13}$C (‰) | | | Tissue $\delta^{15}$N (‰) | |
|---|---|---|---|---|---|---|---|
| | Protein | Energy | Bulk | Protein | Energy | Bulk | Protein |
| Beans | 28±6 | 72±8 | -23.5±1.7 | -24.2±1.7 | -23.3±1.7 | 0.7±3.0 | 0.7±3.0 |
| $C_3$ plants | 11±7 | 89±11 | -23.7±2.0 | -22.3±2.0 | -23.8±2.0 | 8.6±5.4 | 8.6±5.4 |
| $C_4$/CAM plants | 11±7 | 89±11 | -11.2±1.6 | -9.8±1.6 | -11.3±1.6 | 12±5.6 | 12±5.6 |
| Terrestrial animals | 77±13 | 23±13 | -16.7±3.1 | -15.0±3.1 | -22.5±3.1 | 9.5±1.8 | 9.5±1.8 |

composition in the FRUITS model nonetheless permits source differentiation. Indeed, in this example, the roughly 1:3 ratio of protein to energy in beans, which is effectively the inverse of that of the animal flesh (which has an approximately 3:1 protein to energy ratio) renders their respective dietary contributions resolvable in spite of their isotopic similarities.

## Model mechanics

Finally, before presenting the results of our modeling, we briefly discuss the relationship between the isotope data used as model inputs and the modeled outputs, in an attempt to better understand and convey the functioning of these iterations of the model.

As seen in Fig 5, if we consider isotope-modeled food relationships on a foodstuff-by-foodstuff basis, several evident patterns emerge. In terms of the consumption of carbohydrate/energy-rich plant foods ($C_3$ and $C_4$/CAM plants), there are strong correlations with the two primary carbon isotope values ($\delta^{13}C_{co}$ and $\delta^{13}C_{ap}$). Specifically, modeled $C_3$ plant consumption is significantly negatively correlated with both $\delta^{13}C_{co}$ and $\delta^{13}C_{ap}$, whereas modeled $C_4$/CAM consumption is significantly positively correlated with both of those isotopic values. Weaker, but still significant, relationships exist for both of these plant groupings and $\delta^{15}N_{co}$ (which is negatively correlated with consumption of $C_3$ plants and positively correlated with the proportion of $C_4$/CAM plants), reflecting the far higher $\delta^{15}N$ values of the $C_4$/CAM grouping. $\Delta^{13}C_{ap-co}$ follows similar patterns of relationships with the modeled consumption of these two plant food groups, being negatively correlated with modeled consumption of $C_3$ plants and positively correlated with the proportion of $C_4$/CAM plants.

Turning to the more protein rich food groups, beans and terrestrial animal meat, there is a rather straightforward set of relationships that would seem to govern the model outputs. Given their $^{13}C$ and $^{15}N$ depleted signatures, it is not surprising that modeled bean consumption is strongly negatively correlated with all three of the primary isotopic values. In contrast, all three of these values are moderately, but still significantly, positively correlated with the modeled consumption of terrestrial meat, a finding that is in line with what we know of the isotope values of that food group.

In the briefest possible terms, the model is performing as one would expect *a priori*, given the isotopic makeup of the locally available foods. Deviations from 1:1 correlations between isotopic inputs and modeled foodstuff contributions can be attributed to the effects of differential routing and the distinct macronutrient composition of the different food groups employed. As mentioned previously, one of the strengths of this model-based approach is that all these effects are fully and honestly accounted for.

## Modeling

In light of the research questions outlined above, we consider the results of the FRUITS modeling along three main potential vectors of difference (time, geography/space, and sex).

**Overall.** As seen in Fig 6, the average modeled contribution of the four food groupings for all 257 individuals included in the present study hover around 25% each. In spite of this apparent similarity, these values nonetheless differ significantly, both in the aggregate (Kruskal-Wallis $\chi^2 = 18.2$, df = 3, p<0.01), and with pairwise comparisons showing that $C_3$ plant consumption differs significantly from all three other groups. Moreover, these average values obscure an enormous amount of individual-level variability in the consumption of each food (3 to 9 times, depending on the food group). Indeed, for beans, mean modeled individual contributions ranged from 9.8–46.8%, for $C_3$ plants 8.2–46.6%, for $C_4$/CAM plants 6.5–56.2%, and for terrestrial meat 9.7–35.6%. It is this broad individual level variation that we ultimately return to in our discussions below.

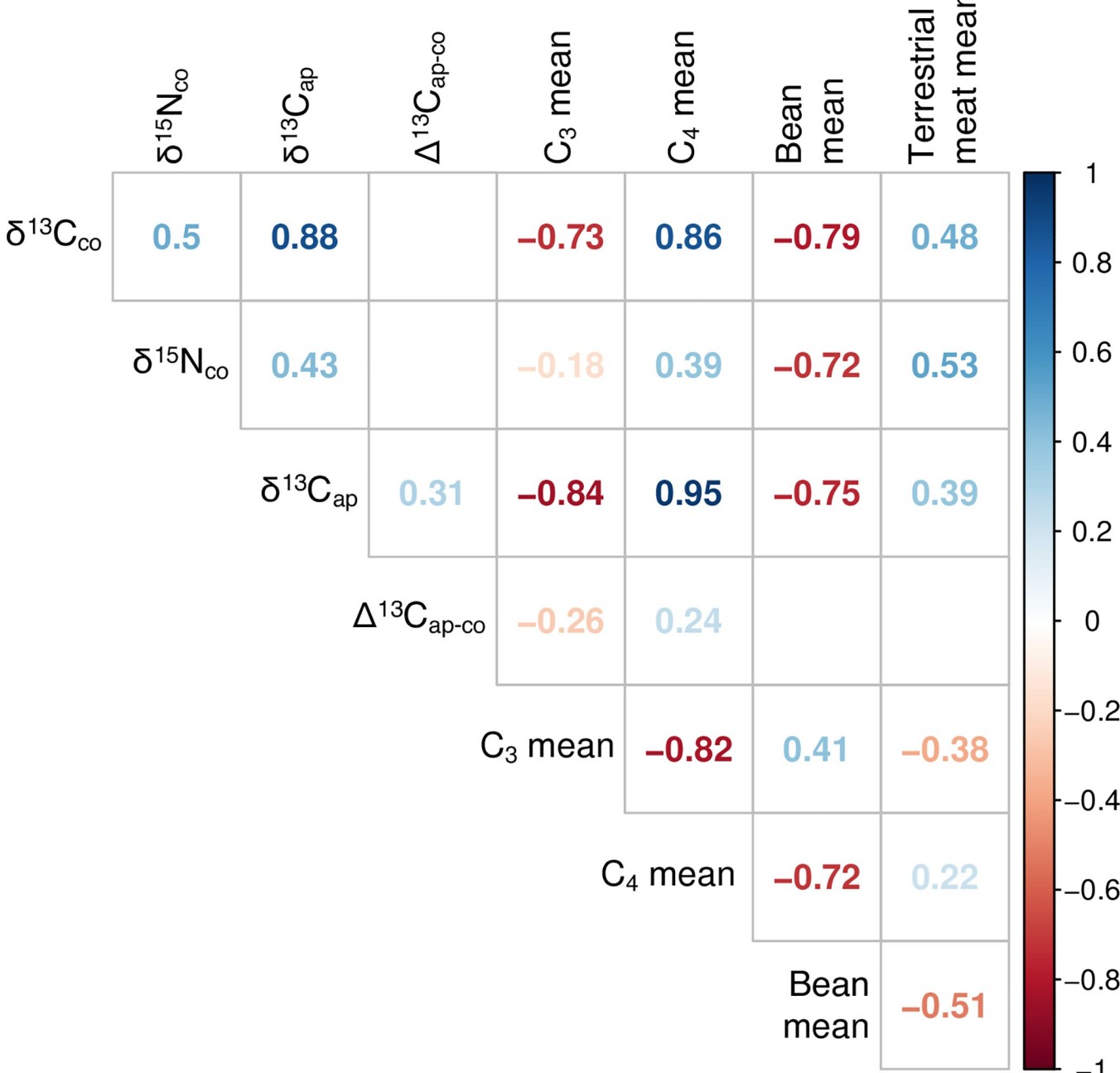

**Fig 5. Correlation matrix of measured isotope values and individual modeled foodgroup contributions.**

**Time.** As seen in Fig 7, based on a correlation analysis of all 167 individuals for whom we possess AMS dates and isotope values, there are statistically significant temporal trends in the consumption of all four food groups. For $C_3$ plants and beans, there are significant decreases over time, whereas for $C_4$/CAM plants and terrestrial meat, modeled consumption increases significantly over time. While we used the median dates of the modeled calibrated ranges for these analyses, given the nearly 1:1 correlation between unmodeled and modeled dates (Fig 8), little changes if the unmodeled median values are used instead.

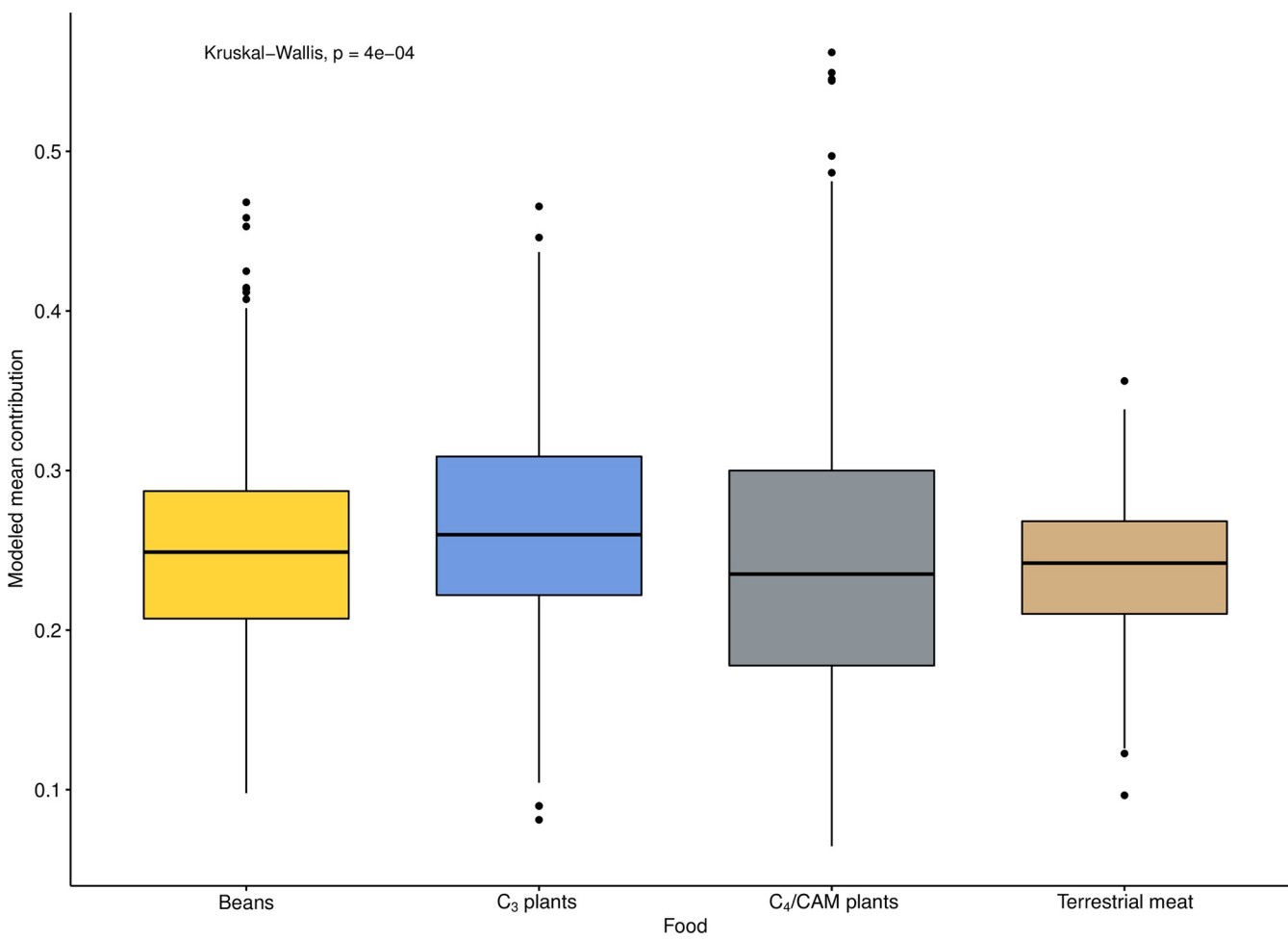

**Fig 6. Boxplot of average mean modeled contribution, by foodgroup.**

Given that there are associated uncertainties in the estimates of both radiocarbon dates and modeled foodstuff contribution values, we also performed a bootstrapped analysis of the observed correlations. As seen in Table 7, at their respective 95% confidence limits, the observed positive correlation between radiocarbon dates and $C_4$/CAM plants still holds (as the 2.5% and 97.5% ρ values are still positive), as does the negative correlation between date and beans. In the case of $C_3$ plants, the previously observed negative correlation with radiocarbon dates is not fully supported, as, at the upper end of the 95% confidence interval, the bootstrapped correlation values extend just beyond/above zero (ρ = 0.01). The same is true with terrestrial meat, as the previously observed positive temporal correlation extends just slightly into negative values (ρ = -0.02) at the lower limits of the 95% confidence interval. However, given that the overwhelming majority of the bootstrapped values for both $C_3$ plants and terrestrial meat support the initial/original correlations, we are inclined to subscribe to their reality, although we must acknowledge that such a conclusion is not supported at a 95% confidence level.

Comparisons of modeled diet were next made between those individuals who had modeled median calibrated AMS dates that preceded and postdated 600 calAD. As seen in Fig 9, this analysis found significant differences (using Wilcoxon rank sum tests) for all four of the modeled food groupings. Pre-600 calAD individuals consumed an average of 33.4±4.7% $C_3$ plants

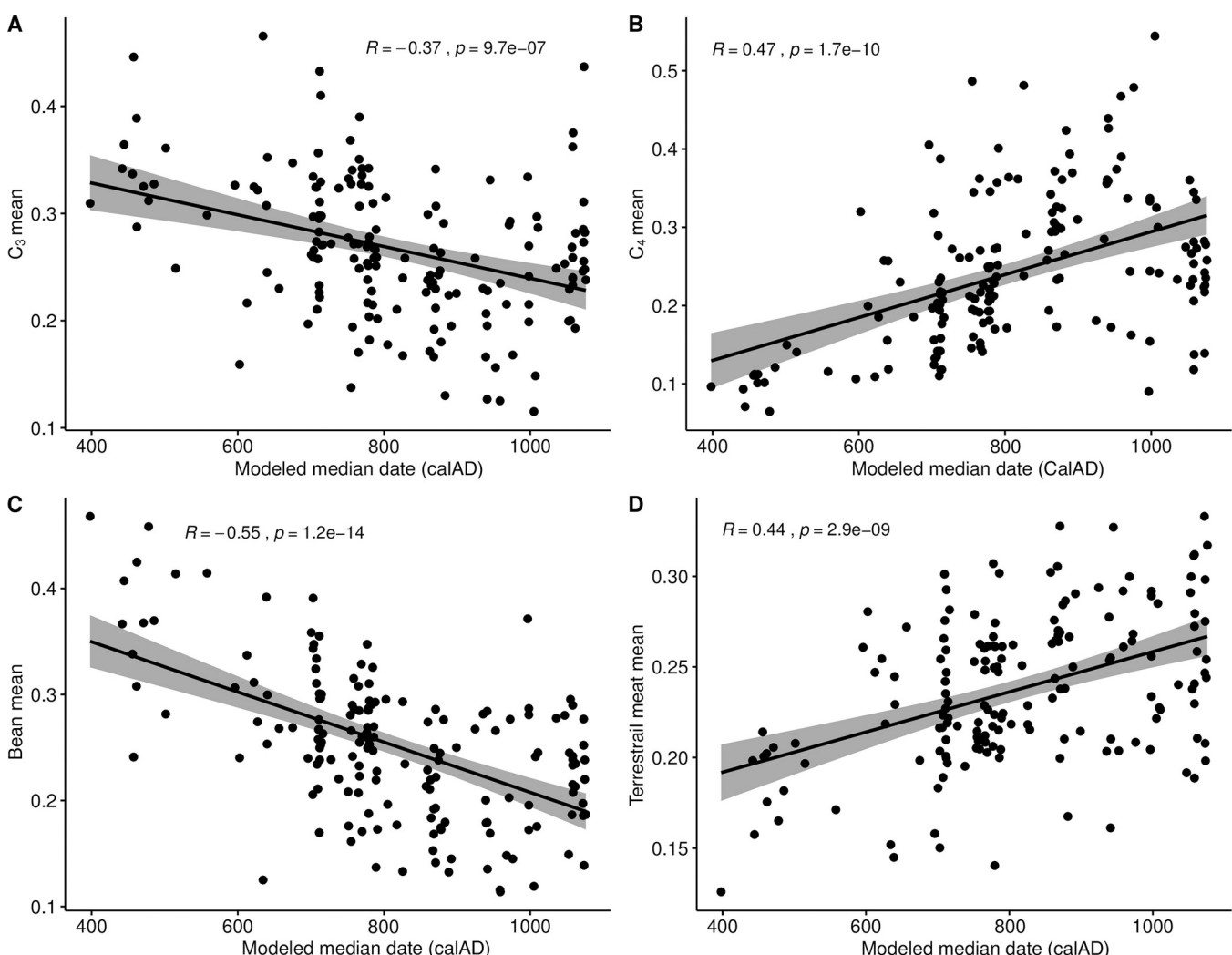

**Fig 7. Scatterplot matrix of modeled median calibrated radiocarbon dates and mean modeled contribution, by foodgroup.**

versus just 26.1±6.4% for the Post-600 calAD individuals. For $C_4$/CAM plants, pre-600 calAD consumption averaged 10.7±2.3%, which more than doubled to an average of 25.6±8.9% in the post-600 calAD individuals. Bean consumption fell from 36.9±6.7% among the pre-600 calAD individuals to 24.1±6.0% after 600 calAD. Correspondingly, terrestrial meat consumption increased from an average of 19±3.2% before 600 calAD to 24.2± 3.9% after that date. These results generally support the argument for behavioral disjuncture at/around 600 calAD.

**Inter-ayllu.** Comparison of modeled diet on an inter-*ayllu* basis found significant differences at an aggregate level for all four modeled food categories, although pairwise comparisons found a somewhat inconsistent pattern of differences, as displayed in Table 8 above the diagonal.

In summary, For $C_3$ plants, the Kruskal-Wallis test indicated significant overall differences ($\chi^2$ = 11.6, df = 5, and p = 0.04), and significant pairwise differences (Wilcoxon rank sum test) were found between five of fifteen pairings. In the case of $C_4$/CAM plants, aggregate differences among the six *ayllus* were significant ($\chi^2$ = 15.1, df = 5, and p = 0.01), and significant pairwise differences were found for seven of fifteen pairs. For beans, there was a larger aggregate difference in consumption ($\chi^2$ = 16.6, df = 5, and p<0.01), and there were accompanying

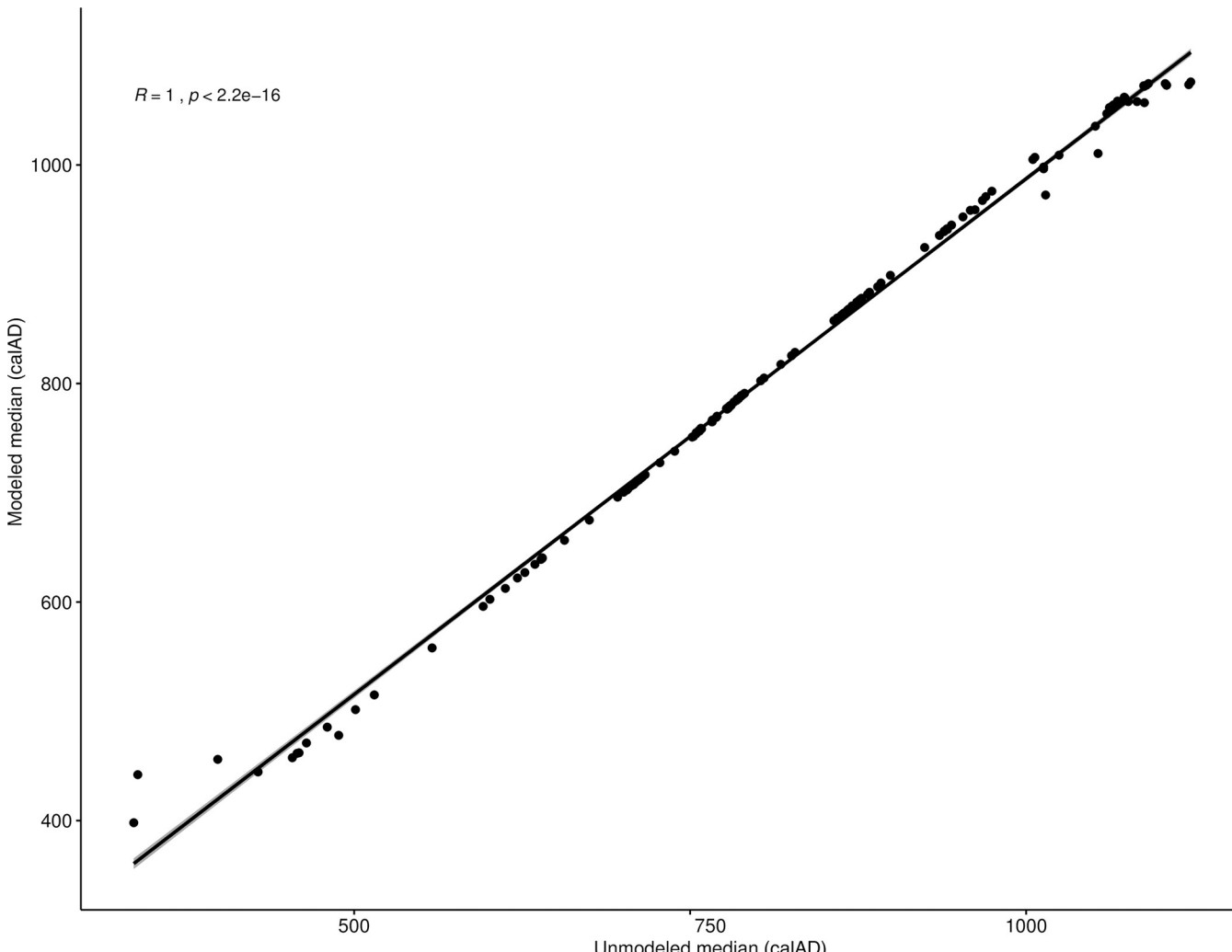

**Fig 8. Scatterplot of unmodeled and modeled median calibrated radiocarbon dates.**

significant pairwise differences among five of fifteen pairings. Finally, for terrestrial meat, we found the largest aggregate difference ($\chi^2$ = 22.3, df = 5, and p<0.01), and significant differences for seven of fifteen pairs.

A graphical summation of these differences is presented in Fig 10, in which we present the results of a multidimensional scaling analysis of all four food groups by *ayllu*. The relative similarity of Tchecar and Solcor is striking, as is the degree of difference between Conde Duque and Larache.

**Table 7. Bootstrapped Spearman's rho correlations of radiocarbon dates and modeled foodstuff contributions.**

|  | C$_3$ plants | C$_4$/CAM plants | Beans | Terrestrial meat |
|---|---|---|---|---|
| **Original ρ** | -0.38 | 0.50 | -0.51 | 0.40 |
| **Original p-value** | 0.00 | 0.00 | 0.00 | 0.00 |
| **2.5% ρ** | -0.26 | 0.19 | -0.34 | -0.02 |
| **50% ρ** | -0.13 | 0.30 | -0.21 | 0.12 |
| **97.5% ρ** | 0.01 | 0.42 | -0.07 | 0.25 |

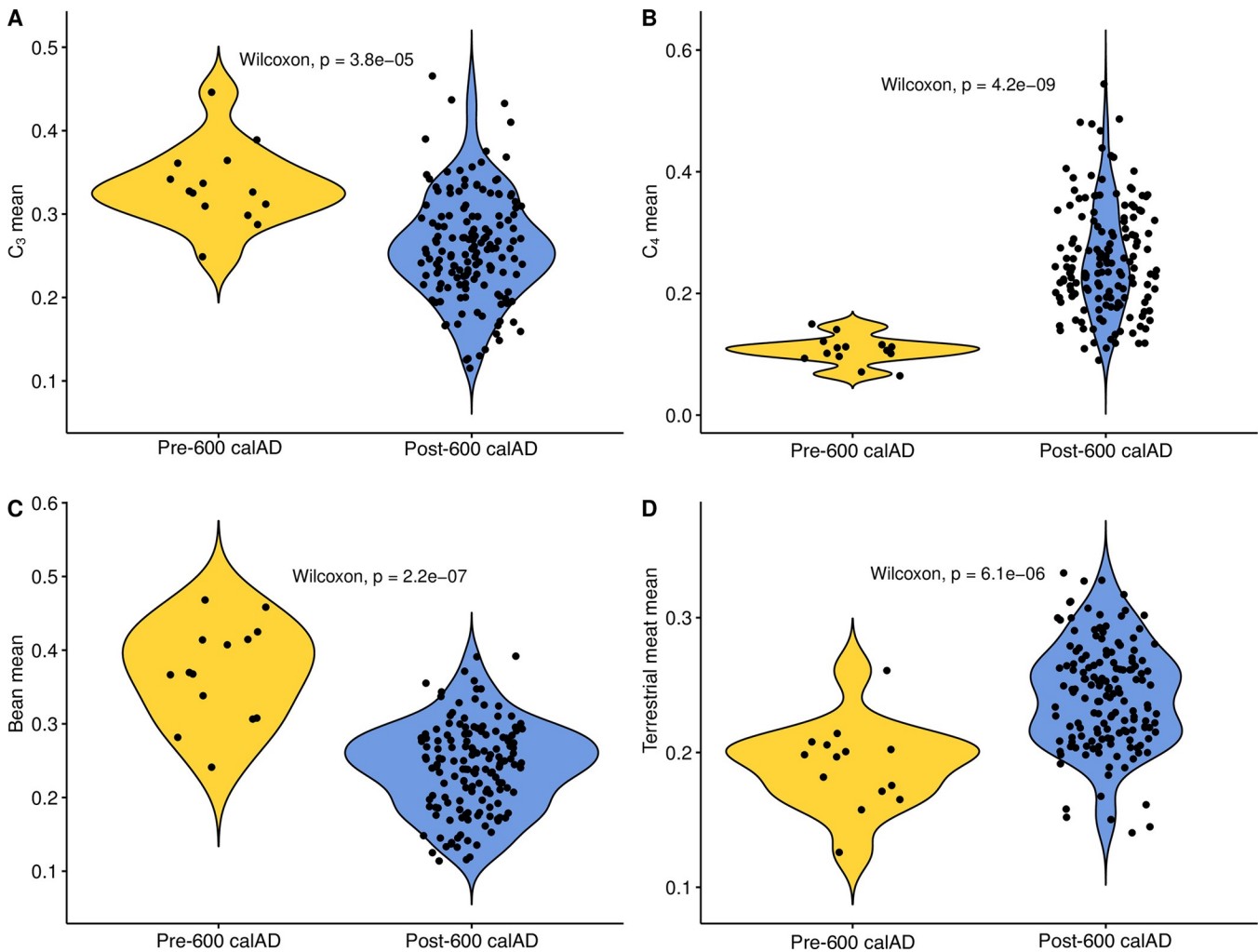

**Fig 9. Violin plot matrix comparing pre- and post-600 calAD mean modeled foodstuff contributions.**

A final consideration of the "raw" modeled foodstuff values relates to the patterns of internal variance in consumption within the different *ayllus*. As seen in Table 9, there were significant differences in variance (assessed using Levene's test for equality of variance) among the *ayllus* for three of the four modeled food groups: $C_3$ plants (F = 3.6, p<0.01), $C_4$/CAM plants

**Table 8. Pairwise comparisons of modeled foodstuff contributions by *ayllu*, with raw values above the diagonal and radiocarbon date-regressed residuals below the diagonal.**

|  | Conde Duque | Coyo | Larache | Quitor | Solcor | Tchecar |
|---|---|---|---|---|---|---|
| Conde Duque |  | C3, C4, Beans, TM |  | C4, Beans, TM |  | C4, Beans |
| Coyo |  |  | C3, C4 | TM | TM | TM |
| Larache |  | C3, C4 |  | C3, C4, Beans | C3, C4 | C3, C4 |
| Quitor |  |  | C3, C4, Beans |  | Beans, TM | TM |
| Solcor |  | C3, C4 | C3, C4, Beans | TM |  |  |
| Tchecar |  | C4 | C3, C4, Beans |  |  |  |

Only significant differences noted.

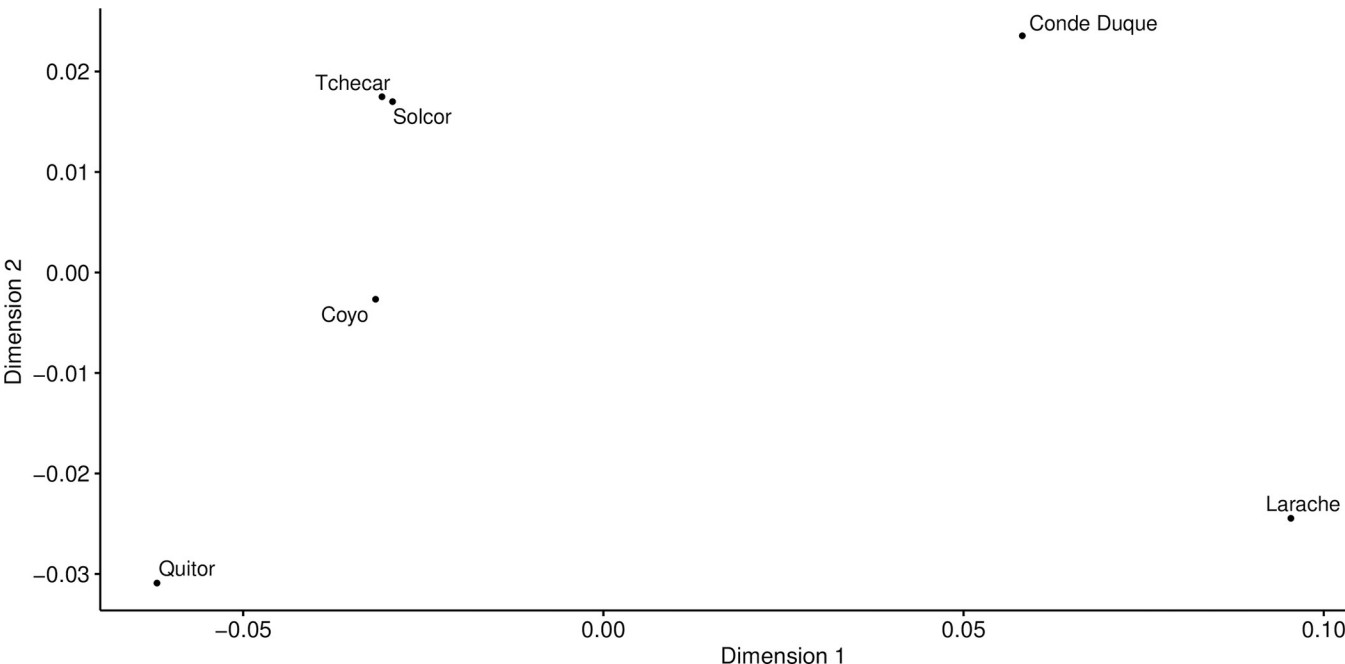

**Fig 10. MDS plot of FRUITS modeled foodgroup contributions, by *ayllu*.**

(F = 5.2, p<0.01), and beans (F = 5.7, p<0.01). Terrestrial meat, on the other hand, did not exhibit significantly different variances among these six *ayllus* (F = 1.4, p = 0.22). The consistent pattern that emerged from this analysis is that Larache and Quitor show the greatest degree of individual-level variability in consumption across the food groupings, whereas Conde Duque, Coyo, and Tchecar show far greater internal consistency in consumption practices.

As the different *ayllus* belong to different portions of the Middle Period, we augmented this comparison of the "raw" modeled foodstuff proportions with a second level of analysis considering the residuals of modeled contributions after regression against Inter-*Ayllu* modeled calibrated radiocarbon dates [42]. Even after accounting for time, some significant differences in modeled consumption remained, as seen in Fig 11 and with pairwise comparisons shown in in Table 8 below the diagonal.

**Table 9. Variance of FRUITS modeled dietary contribution by *ayllu* with results of Levene's test for equality of variance.**

| Ayllu | n | SD of $C_3$ mean | SD of $C_4$ mean | SD of Bean mean | SD of Terrestrial meat mean |
|---|---|---|---|---|---|
| **Conde Duque** | 11 | 3.9% | 7.6% | 5.3% | 3.2% |
| **Coyo** | 86 | 6.1% | 7.9% | 5.4% | 3.5% |
| **Larache** | 20 | 9.3% | 14.7% | 9.2% | 4.6% |
| **Quitor** | 48 | 8.0% | 11.6% | 8.8% | 5.0% |
| **Solcor** | 66 | 6.5% | 8.8% | 6.0% | 4.1% |
| **Tchecar** | 26 | 5.0% | 7.0% | 5.6% | 4.5% |
| **Levene's test for equality of variance** | | | | | |
| **F-statistic** | | 3.62 | 5.22 | 5.65 | 1.43 |
| **p** | | <0.01 | <0.001 | <0.001 | 0.22 |

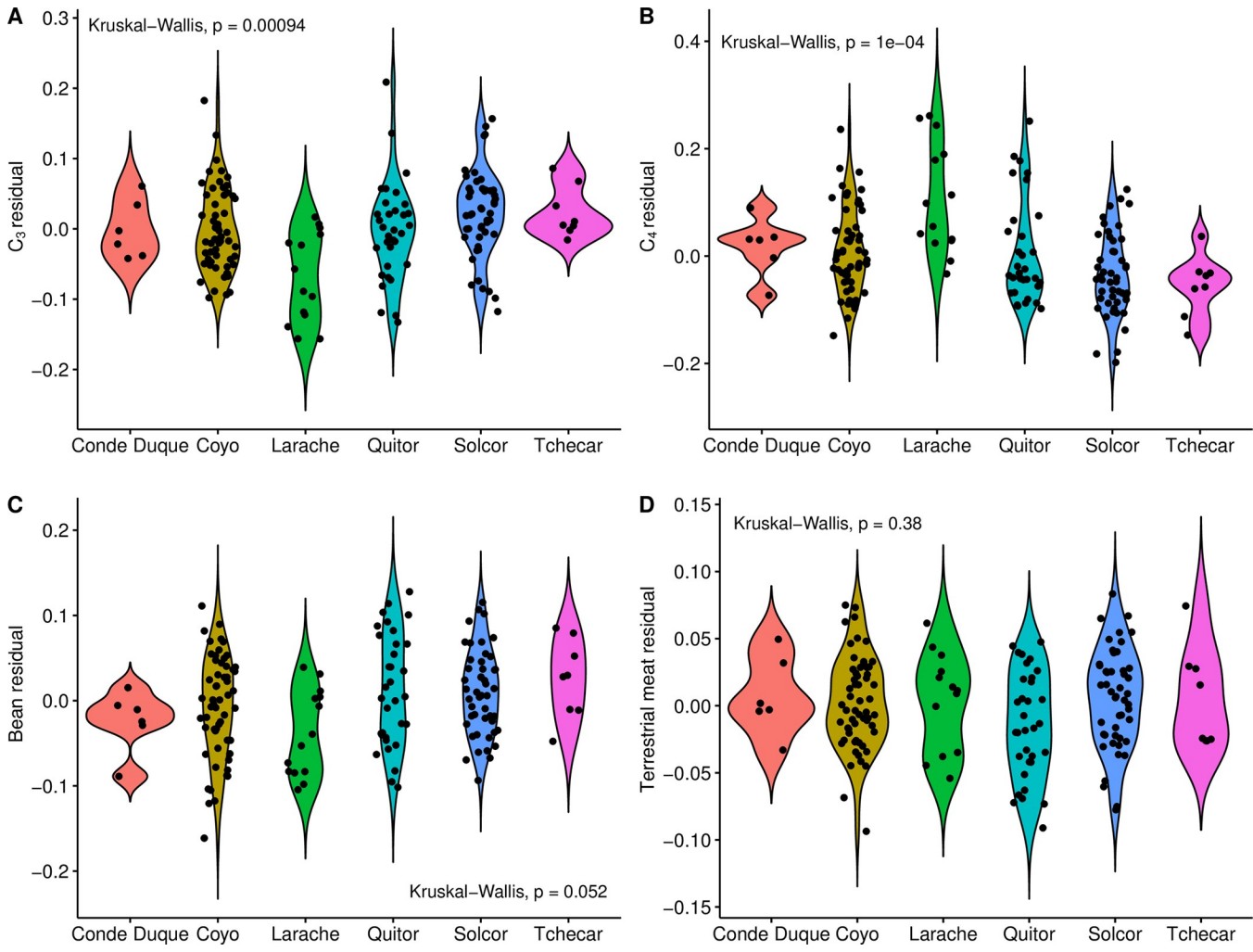

**Fig 11. Violin plot matrix of radiocarbon date-regressed modeled foodgroup contributions by *ayllu*.**

For $C_3$ plant residuals, the Kruskal-Wallis test indicated significant differences overall ($\chi^2$ = 20.7, df = 5, and p<0.01), and significant pairwise differences (Wilcoxon rank sum test) were found for five of fifteen pairings. In the case of $C_4$/CAM plant residuals, aggregate differences among the six *ayllus* were significant ($\chi^2$ = 25.7, df = 5, and p<0.01), and significant pairwise differences were found for six pairs. For bean residuals, there was only a borderline significant aggregate difference ($\chi^2$ = 11.0, df = 5, and p = 0.05), and there were only three accompanying significant pairwise differences. Finally, for terrestrial meat, there was no significant overall difference ($\chi^2$ = 5.3, df = 5, and p = 0.38), and only one significant pairwise difference. Having thus controlled for time, it would appear that inter-*ayllu* differences persist, at least for plant consumption and, potentially, for beans as well, although other differences (e.g., Conde Duque's distinctiveness) moderate once time is accounted for.

These differences are presented in graphical form in Fig 12, which shows the results of a multidimensional scaling analysis of all four food group residuals by *ayllu*. While the pattern, after accounting for time, differs somewhat from the raw value MDS plot by *ayllu* presented above, the overall structure is not markedly different. The relative proximity of Tchecar and Solcor remains a striking feature, as does the continued dissimilarity of Conde Duque versus Larache.

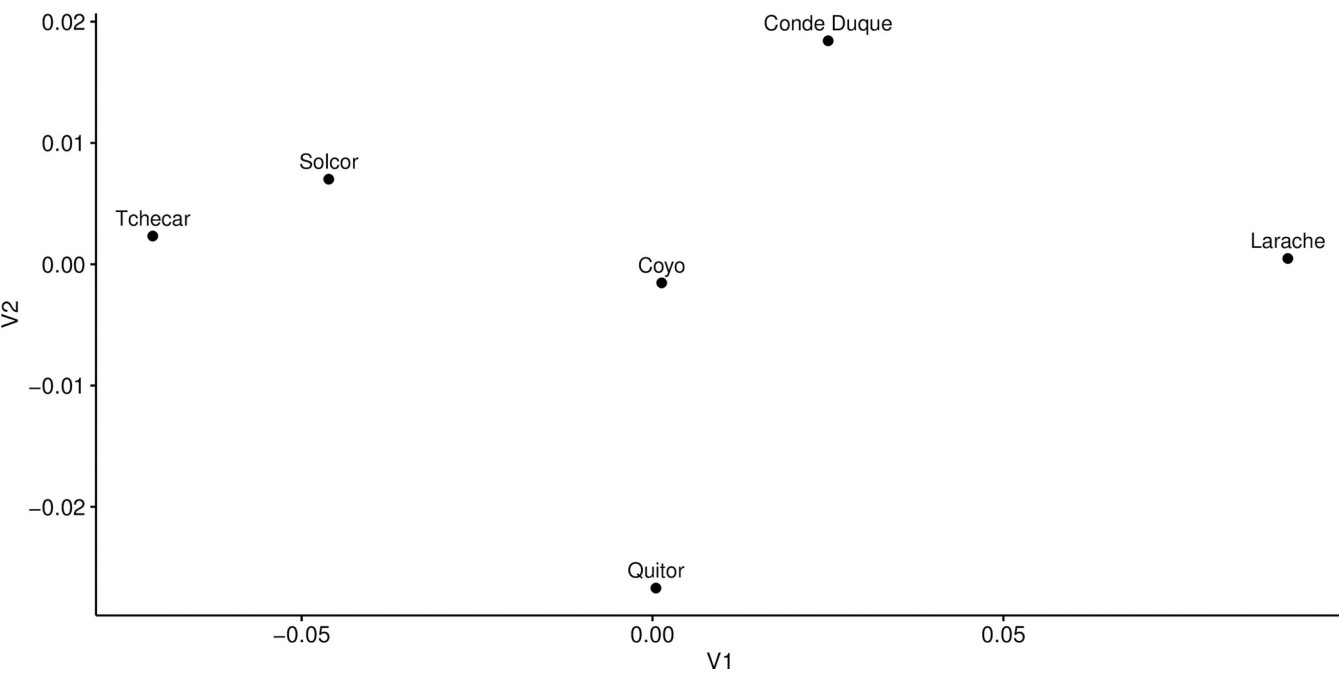

**Fig 12. MDS plot of radiocarbon date-regressed modeled foodgroup contributions by *ayllu*.**

A comparison of the variances of the residuals among the six *ayllus* under consideration found no significant differences for any of the four modeled food groupings once differences in time were fully accounted for by regression. These results are presented in Table 10. Despite the lack of significant differences, the general pattern observed in the raw data holds true, as the individuals from Larache and Quitor show the greatest degree of individual-level variability in consumption across the food groupings, whereas Conde Duque and Tchecar show more internal consistency in dietary makeup. Having controlled for time, however, these differences are not significant.

**Sex.** The first level of comparison of sex-based differences in modeled consumption was made at the level of all sexed individuals (n = 179) across all *ayllus*. As the temporal structure of sexed and dated individuals (n = 112) does not differ significantly (Fig 13), this seemed an appropriate first comparison. As seen in Fig 14, in spite of a working presumption that female and male diets might differ significantly, statistically significant difference was only observed

**Table 10. Variance of FRUITS radiocarbon date-regressed residuals by *ayllu* with results of Levene's test for equality of variance.**

| Ayllu | n | SD of $C_3$ mean | SD of $C_4$ mean | SD of Bean mean | SD of Terrestrial meat mean |
|---|---|---|---|---|---|
| **Conde Duque** | 6 | 4.1% | 5.4% | 3.5% | 2.9% |
| **Coyo** | 58 | 5.9% | 7.9% | 5.9% | 3.4% |
| **Larache** | 14 | 6.4% | 10.2% | 5.1% | 3.7% |
| **Quitor** | 33 | 7.0% | 9.5% | 6.5% | 3.9% |
| **Solcor** | 48 | 6.2% | 7.5% | 5.0% | 3.7% |
| **Tchecar** | 8 | 3.6% | 5.6% | 4.7% | 3.7% |
| **Levene's test for equality of variances** | | | | | |
| **F-statistic** | | 0.9 | 1.2 | 1.8 | 0.51 |
| **p** | | 0.49 | 0.31 | 0.12 | 0.77 |

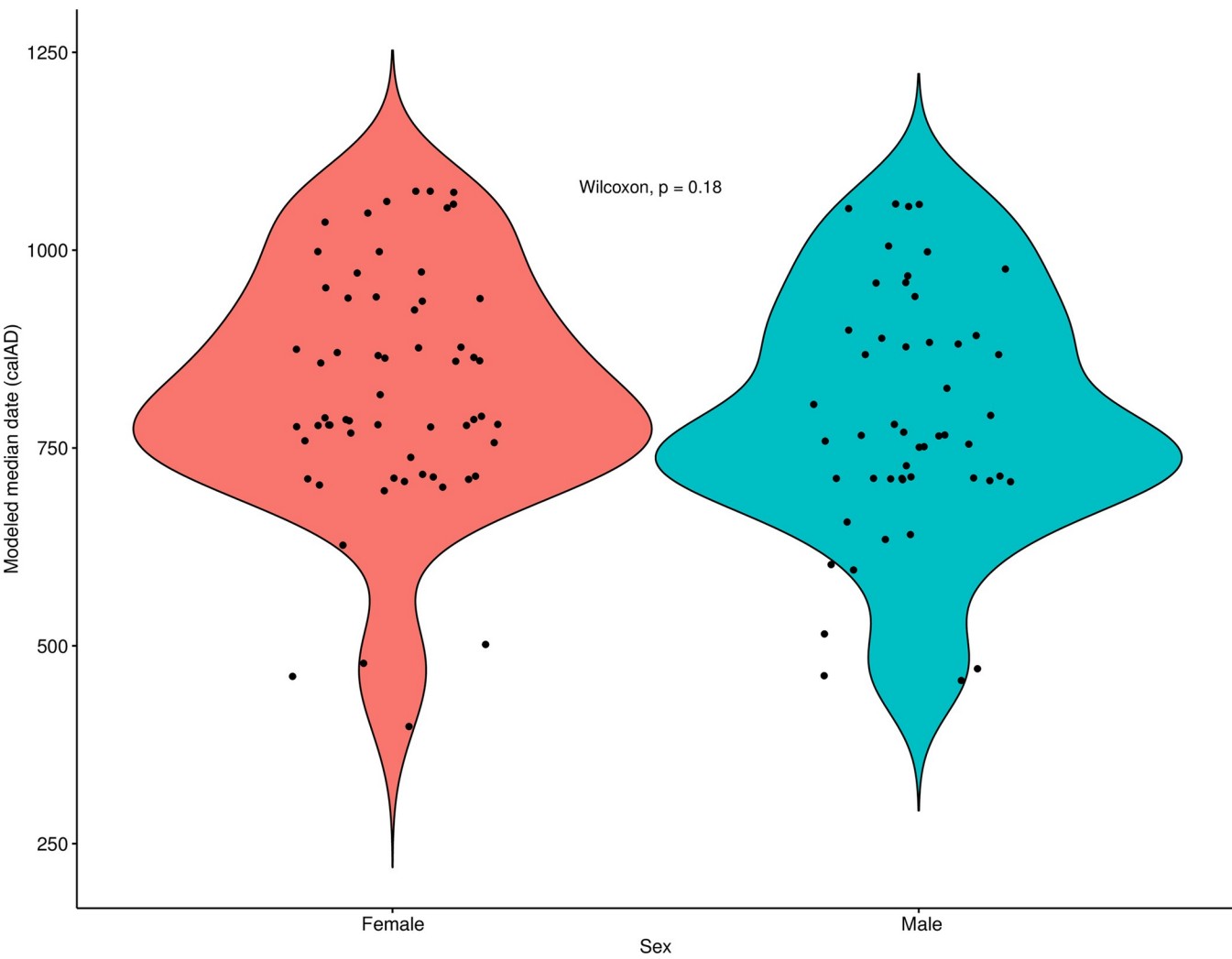

**Fig 13. Violin plot of modeled median calibrated radiocarbon dates by sex.**

for one of the four modeled food groups (beans). Indeed, for $C_3$ plants, $C_4$/CAM plants, and terrestrial meat, there was substantial overlap in male and female consumption and insignificant sex-based differences in the modeled outcomes. On the other hand, females appear to have consumed significantly more beans (26.3±6.2%) than did their male counterparts (23.2 ±7.3%). This difference was not exclusive, however, as is evident in the Fig 14, which shows the substantial overlap between the proportion of beans consumed by females and males.

Acknowledging that substantial variability in sex-based difference might have been obscured at the previous scale of analysis by temporal differences among the *ayllus*, the next level of analysis was conducted on an intra-*ayllu* scale, with male and female modeled diet from each *ayllu* compared with one-another. A sufficient number of sexed individuals were available from only four *ayllus*: Coyo, Larache, Quitor, and Solcor (Table 11). Conde Duque was excluded from this analysis as it comprised just seven individuals for whom we could determine sex, while Tchecar, which had fifteen sexed individuals, only included five males. We chose to include Larache in this analysis because, while the sample size was smaller-than-preferred (n = 18), that number did include an equal number of females and males, and

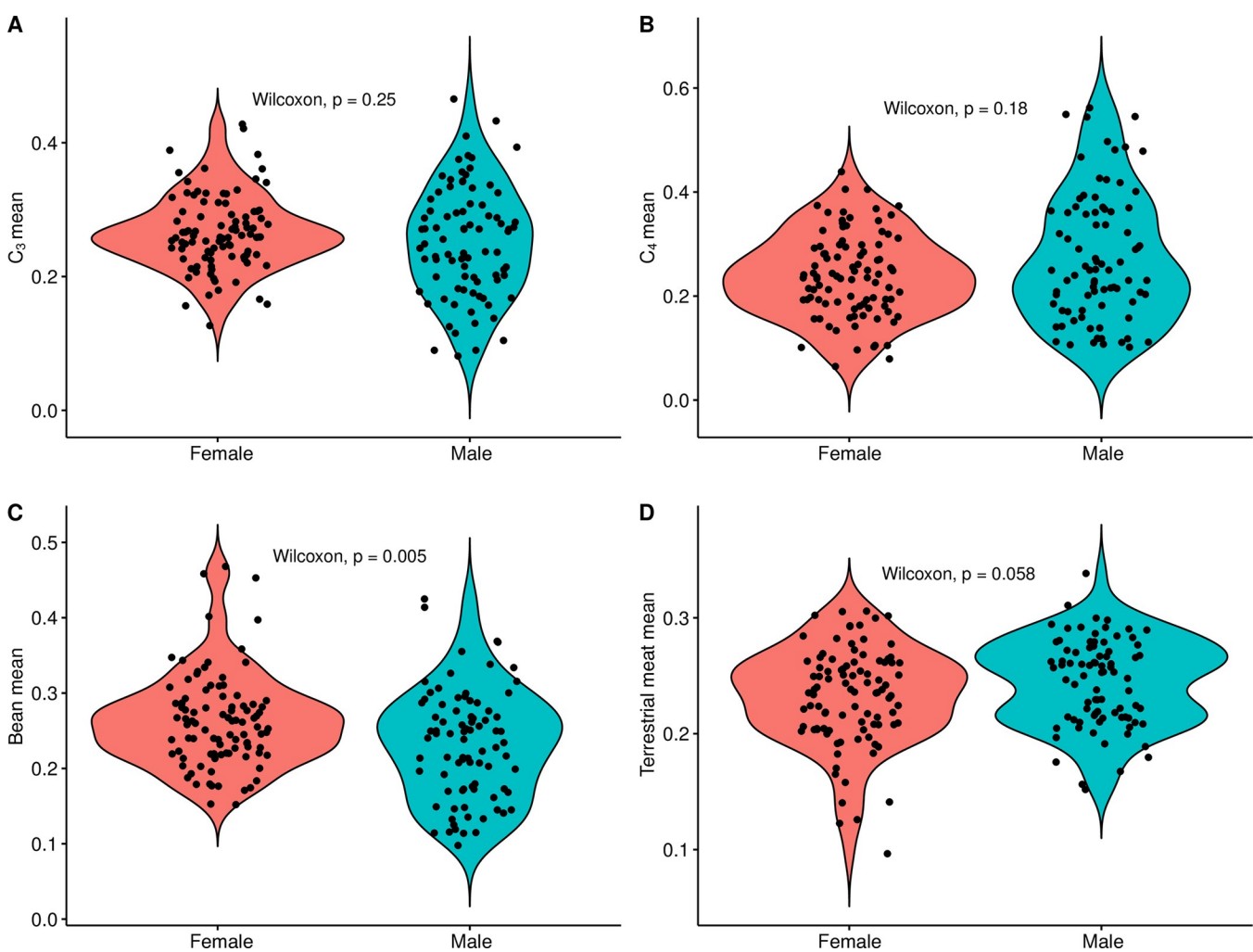

**Fig 14. Violin plot matrix of FRUITS mean modeled foodstuff contributions by sex.**

Larache is a cemetery with a number of intriguing contextual elements, upon which many arguments about the Middle Period in San Pedro have been built [102–104].

For three of these four *ayllus*, there were no significant differences in modeled consumption between females and males. Among the Coyo individuals (n = 54), the p-values for comparisons (Wilcoxon rank sum tests) of male and female diet ranged from 0.22–0.66. For Quitor

**Table 11. Number of sexed individuals by *ayllu*.**

| Ayllu | Female | Male | Indeterminate | Total |
|---|---|---|---|---|
| Conde Duque | 3 | 4 | 4 | 11 |
| Coyo | 38 | 36 | 12 | 86 |
| Larache | 9 | 9 | 2 | 20 |
| Quitor | 19 | 12 | 17 | 48 |
| Solcor | 29 | 25 | 12 | 66 |
| Tchecar | 10 | 5 | 11 | 26 |
| *Total* | 108 | 91 | 58 | 257 |

**Table 12. Pairwise comparisons of modeled foodstuff contributions by *ayllu* and sex, with raw values above the diagonal and radiocarbon date-regressed residuals below the diagonal.**

| | Conde Duque_F | Conde Duque_M | Coyo_F | Coyo_M | Larache_F | Larache_M | Quitor_F | Quitor_M | Solcor_F | Solcor_M | Tchecar_F | Tchecar_M |
|---|---|---|---|---|---|---|---|---|---|---|---|---|
| Conde Duque_F | ■ | | | | | | | | | | | |
| Conde Duque_M | | ■ | Beans, TM | TM | | | Beans, TM | | Beans | | | |
| Coyo_F | | | ■ | | | C3, C4, Beans | | | | | | TM |
| Coyo_M | | | | ■ | | C3, C4, Beans | Beans | | | | | TM |
| Larache_F | | | | | ■ | C3, C4, Beans | | | | | | |
| Larache_M | | | C3, C4, Beans | C3, C4 | C3, C4, Beans | ■ | C3, C4, Beans, TM | C3, C4, Beans | C3, C4, Beans | C3, C4, Beans | C3, C4, Beans | C3, C4, Beans |
| Quitor_F | | | | Beans | C4 | C3, C4, Beans | ■ | | TM | TM | TM | TM |
| Quitor_M | | | | | | Beans | Beans | ■ | | | | |
| Solcor_F | | C4, Beans | | C4 | C4 | C3, C4, Beans | | C3, C4, Beans | ■ | | | |
| Solcor_M | | | | | | C3, C4, Beans | TM | C4 | | ■ | | |
| Tchecar_F | | | | C4 | C4, Beans | | | Beans | | | ■ | |
| Tchecar_M | | | | | | | | | | | | ■ |

Only significant differences noted.

(n = 31), p-values for food group comparisons ranged from 0.07–0.21. Finally, for Solcor (n = 54), the p-values were similarly unimpressive, ranging from 0.43–0.58.

It was only at Larache (which has a relatively small sample size, n = 18) that any significant sex-based differences were observed. In this cemetery, apart from terrestrial meat (W = 22, p = 0.11), consumption of the other three food groups differed significantly between females and males. Specifically, females consumed a significantly greater amount (26.4±6.8% vs. 14.9 ±7.7%) of $C_3$ plants than males (W = 68, p = 0.01), whereas males consumed significantly more $C_4$/CAM plants than females (44.1±14.1% vs. 25.7±6.7%; W = 14, p = 0.02). Finally, females at Larache consumed significantly more beans than did their male counterparts (26.1 ±7.6% vs. 15.4±6.7%; W = 70, p<0.01).

The final iteration of this sex-based analysis combined *ayllu* and sex, forming twelve groups, as to determine the effects of sex on diet across *ayllus*. *Ayllu*/sex pairs with significant differences are noted in Table 12 above the diagonal. For $C_3$ plants, the Kruskal-Wallis test found an overall significant difference among these twelve groups ($\chi^2 = 21.2$, df = 11, and p = 0.03), and significant pairwise differences (Wilcoxon rank sum test) for nine of the sixty-six pairwise comparisons, all involving males from Larache. In the case of $C_4$/CAM plants, aggregate differences among the twelve groups were significant ($\chi^2 = 20.3$, df = 11, and p = 0.04), and pairwise comparison found significant differences for the same nine pairings as the $C_3$ plant analysis (Larache males being significantly different from the same eight other subgroups).

For beans, the comparison of the twelve groups again found an aggregate significant difference ($\chi^2 = 23.2$, df = 11, and p = 0.02), and thirteen significant pairwise differences of the possible sixty-six, as noted in Table 12 above the diagonal. Finally, for terrestrial meat, we again

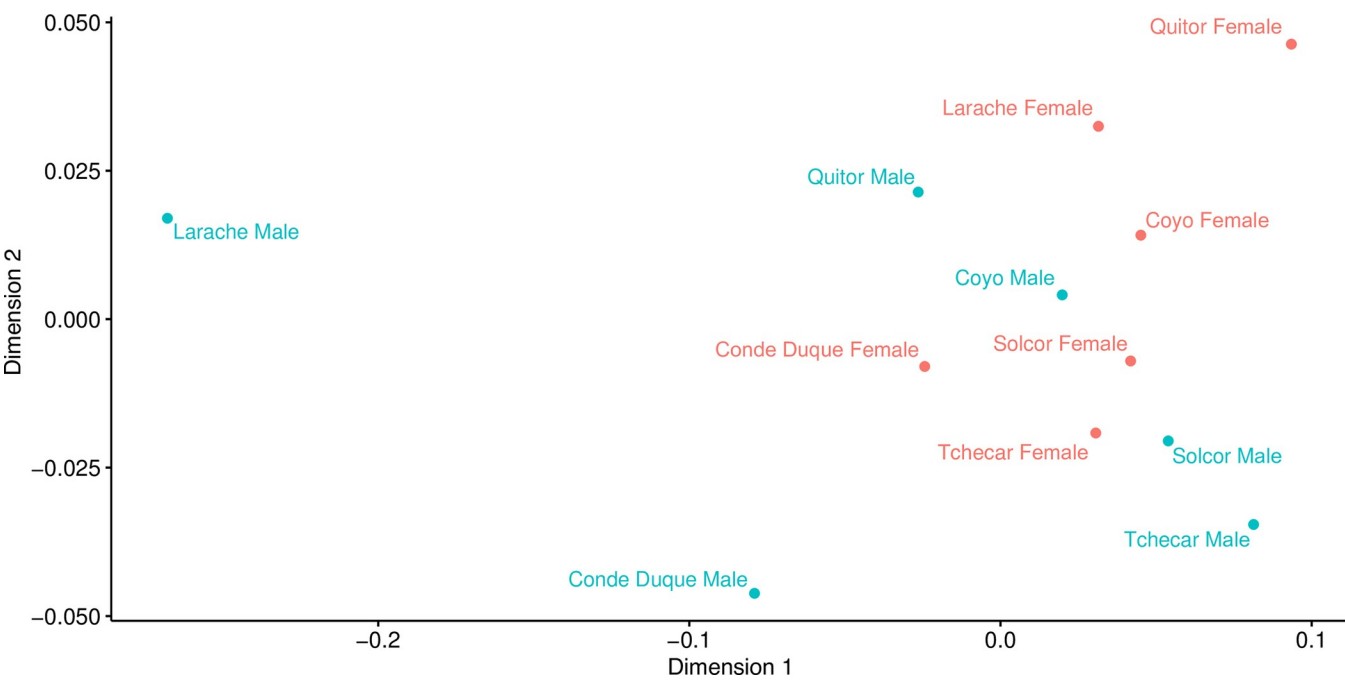

**Fig 15. MDS plot of FRUITS modeled foodgroup contributions, by *ayllu* and sex.**

found a significant aggregate difference ($\chi^2$ = 22.3, df = 11, and p = 0.02), and significant differences for ten of the sixty-six pairs.

A graphical summation of these differences is presented in Fig 15, in which we present the results of a multidimensional scaling analysis of the twelve *ayllu*-sex groups that accounts for all four modeled food groups. As presaged by the analysis presented above, there are some differences between the Quitor females and Conde Duque males, with the Larache males appearing most distinctive of all.

As before, given that the different *ayllus* belong to different portions of the Middle Period, we augmented this comparison of the "raw" modeled foodstuff proportions with a second level of analysis considering the residuals of modeled contributions for the *ayllu*-sex groups after regression against Inter-*ayllu* modeled calibrated radiocarbon dates. It should be noted that, for this level of analysis, the total number of groups decreased to eleven, as there were no dated males from Tchecar. In spite of accounting for time, some significant differences remained at this level of the analysis. *Ayllu*/sex pairs with significant differences in residual values are noted in Table 12 below the diagonal.

For $C_3$ plant residuals, the Kruskal-Wallis test did not find an overall significant difference among the eleven *ayllu*-sex groups ($\chi^2$ = 17.8, df = 10, and p = 0.06). Nonetheless, there were significant pairwise differences for seven of fifty-five pairs. Similarly, for terrestrial meat, the aggregate differences among the eleven groups was not significant ($\chi^2$ = 6.8, df = 10, and p = 0.74), and there was only one pairwise difference. For beans, however, the comparison of the *ayllu*-sex groups did find an aggregate significant difference ($\chi^2$ = 22.6, df = 10, and p = 0.01), and twelve significant pairwise differences of a possible fifty-five. Likewise, for $C_4$/CAM plants, a significant aggregate difference was again found ($\chi^2$ = 23.6, df = 10, and p<0.01), and significant differences were found for fourteen of the fifty-five possible pairs, as noted below the diagonal in Table 12.

A graphical summation of these differences is presented in Fig 16, in which we present the results of a multidimensional scaling analysis of the eleven *ayllu*-sex groups that accounts for

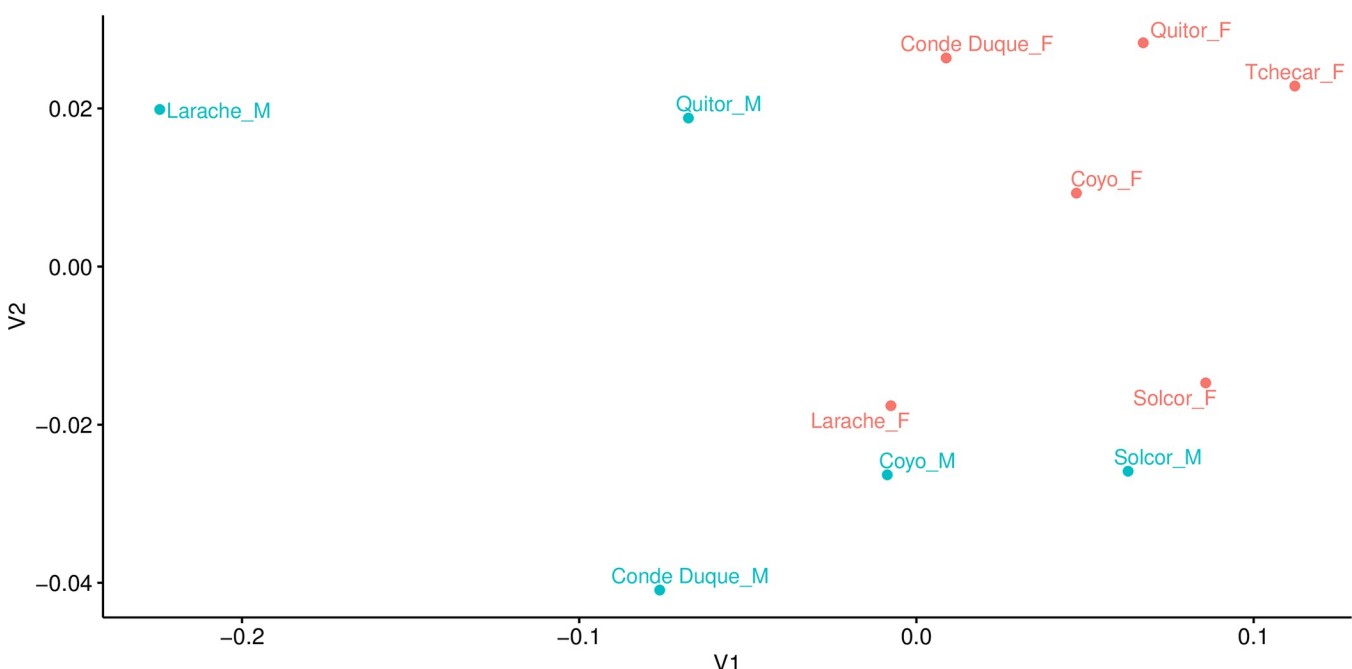

**Fig 16. MDS plot of radiocarbon date-regressed modeled foodgroup contributions by *ayllu* and sex.**

the residuals of all four modeled food groups. As above, the Larache males stand out as the most notably distinct subgroup, followed by the Conde Duque males. Additionally, it is noteworthy, as seen in Fig 17, that the females from the various *ayllus* form a rough "cluster" and the males from the *ayllus* exhibit greater variety. This suggests a greater homogeneity of time-controlled diet among females and a broader range of dietary practices among males.

## Discussion

In this work, we have used diet as a proxy for underlying social dynamics, acknowledging that any observed changes in consumption likely flow from, or reflect changes in, the structures of the societies under study (and, conversely, that food could have been used to affect certain social changes). In our consideration of the three research axes structuring this work (time, space, and sex), we highlight how the gastro-political lens of the study of paleodiet suggests new insights into the social dynamics of the Middle Period societies of San Pedro de Atacama.

Beginning with the considerations of time, our analysis revealed that: 1) over the 600-year period represented by our sample, there were significant changes in consumption patterns that may evince broad diachronic changes in the structure of Atacameño society, and 2) at/near 600 calAD, there probably was an episode of social discontinuity that manifested in significant changes in consumption practices.

Changes over time have been at the core of the archaeological discussion of the occupation of the San Pedro de Atacama oases [19, 20, 24, 25, 68, 105–112], given the long-term dynamics of local human occupation and their connections to neighboring regions. Despite this focus on changing social and biological aspects of life in the past, there has been a strong tendency to define local prehistory through discrete (non-overlapping) periods. In that context, much of the previous discussion about diet and health among past Atacameños has focused on comparisons between such periods [43, 44, 113, 114]. In recent years, however, there has been an important shift away from typological views of the past, with archaeological [115–120] and

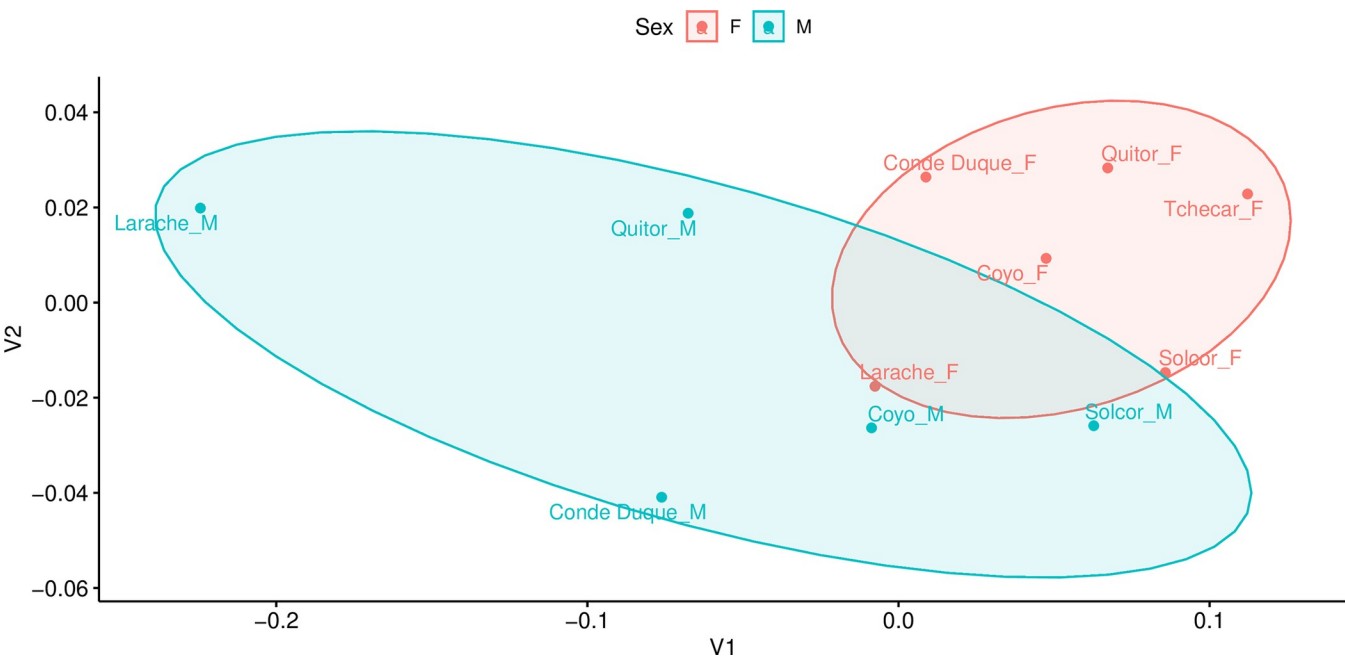

**Fig 17. MDS plot of radiocarbon date-regressed modeled foodgroup contributions by *ayllu* and sex with "cluster" of female values and greater dispersion of male values noted by ellipses.**

bioarchaeological work [42, 46, 60, 61, 120–122], which has highlighted both synchronic variability and gradual historical processes that were products of intrinsic social developments, like population growth, emerging social complexity, and local cultural developments, as well as the exchange of people, resources, and ideas from other regions.

This new lens on the local past presents a more detailed and nuanced view of local prehistory, allocating agency to local actors, and situating the oases as important nodes inside the social network (or meshwork) that organized and influenced the South-Central Andes (rather than just a peripheral society within the sphere of influence of larger polities). While there is a long-term engagement in the scholarly literature with the important networks and polities of the Middle Period, our data reveals the import of local traditions and resources in shaping the human experience in the San Pedro de Atacama oases. In other words, this type of work creates a space where the discussion of the past can be seen as the result of the interplay between local and regional. The results we observe in diet over time in the Middle Period fit well within this discussion, as they show a gradual change in diet in the oases, stressing the variation that existed internal to this period.

As pertains to the observed long-term diachronic changes, the increased consumption of $C_4$/CAM plants and terrestrial meat over time could both be argued to reflect a broad increase in the status and/or wealth of individuals across the oases. It is noteworthy that these classes of food have been associated with certain categories of power/privilege. While likely holding a variety of economic, ritual, political, military, and social functions throughout Andean prehistory, in later periods (in particular the Inka and colonial ages), *chicha* (a maize beer) is imbued with particular social and political power [123–126]. Interestingly, some of the higher $C_4$ values in our analyses come from Larache, where some of these individuals are interred with gold *keros*, the name given to a type of waisted drinking vessel. *Keros* are found in both the Middle Period Tiwanaku polity and the later Inca and are frequently understood as chicha drinking vessels [127]. While we cannot say with certainty that the observed increase in $C_4$/CAM

consumption is the result of the wider availability or distribution of *chicha*—indeed it could simply represent an increase in cultivation of $C_4$ plants/maize in the oases over the period of study—the possibility cannot be discounted, especially given the importance that *chicha* has as a ritual beverage across the Andes.

Similarly, there is a broad anthropological literature (from non-human primates [128] to complex human societies [129]) that associates greater meat consumption with higher status. The increasing role that both of these classes of (potentially) high-status food played over the centuries in Atacameño diet could appear to confirm the long-held narrative that describes the Middle Period as one of unparalleled prosperity [19, 43, 44, 130] from which all or many of the inhabitants of the oases benefitted. The change in diet may reflect the impact of the growing networks of interregional exchange (e.g., [43, 120–122, 131]), bringing not only wealth, but different dietary practices to the oases. As we have previously argued [60], incorporation into these exchange networks likely served to accelerate or accentuate intra-societal dietary variability.

This notion of significant change over the course of the Middle Period itself has and should continue to be explored via the material culture and settlement patterns of the area. In that vein, the observed dietary disjunction at/near 600 calAD would appear to have correlates in other categories of archaeological material. Both Stovel [132] and Gallardo and colleagues [133] have identified significant coeval shifts in patterns of ceramic production and decoration at/near this time that should be further explored in other forms of material culture. These parallel findings illuminate new foci for future research; it would be interesting to examine, for instance, whether 600 calAD represents an inflection point in the intensity, level, or mechanism of foreign presence in, or interactions with, the people of the San Pedro de Atacama oases.

Turning to socio-spatial considerations, our analysis of dietary makeup by *ayllu* revealed several interesting phenomena. First, while dietary variability was greater at Larache and Quitor and lower at Conde Duque, Coyo, and Tchecar, differences in internal variability among the *ayllus* decreased to insignificance once temporal differences were accounted for. Put differently, while there were some differences in the level of internal dietary variability among the *ayllus*, once time was fully considered, none of the *ayllus* stood out for having a more (or less) clearly internally differentiated cuisine. This could be a consequence of limitations of diet variability at any given time in what is, after all, an ecologically constrained environment. However, the lack of internal dietary variability (a truly internally differentiated cuisine) could also be a feature of emergent, rather than established systems of inequality. Thus, a true internally differentiated internal cuisine might only be expected in locales/cultures with established social hierarchies, such as at Moundville [41], Cahokia [134], Classic Period Oaxaca [135], or during the height of the Inca Empire [136]. Or, as Goody [10] noted to be possible, it could be the case that internally differentiated cuisine failed to appear in spite of the existence of bona fide institutionalized inequality, the former not always requiring the latter.

In contrast, at the level of both raw foodstuff contribution values and date-regressed residuals, the inhabitants of Larache (and to a lesser degree Conde Duque) stand apart from the individuals from the other four *ayllus*. Larache's distinctiveness in terms of dietary practice is, perhaps, attributable to the greater presence of non-local individuals among those people interred there. Previous strontium isotope studies [104, 137], have identified Larache as an *ayllu* with a high percentage of non-local persons, with five of eighteen analyzed individuals possessing strontium signatures indicative of birth/youth spent outside of the San Pedro oases. That four of the five presumptively "foreign" individuals interred at Larache were male, and that the diet of the Larache males is the most distinctive of any *ayllu*-combination, supports this notion that it is the place of origin and/or continuing maintenance of non-local dietary

traditions among a subset of Larache individuals that is driving the observed *ayllu*-based difference(s).

Larache's distinctiveness has long been noted, and it was once considered the site of a colony of elite Tiwanaku settlers in the Atacama [102, 138]. While this speaks to an earlier perspective that gave primacy to the role of Tiwanaku in shaping the Atacama oases, our bioarchaeological data more clearly support Larache as "the burial place for a diverse yet culturally integrated and potentially elite and well-connected segment of the Atacameño population" [104], and not a space and people inherently tied to Tiwanaku. Given the interesting results seen in the modeling data, it is perhaps worth considering Martínez's [139] idea of interdigitated populations instead of assuming a pattern of ethnic enclaving. Using data from the *Revisita de Atacama* of 1683, Martínez details the interdigitated relationships between Atacama, López, Tucumán, and Chicha regions, noting the significant movement between, and occasionally the permanent settlement of, members of these different groups in each of the other areas [35, 139, 140]. In some cases, the *Revisita* records over 40% of a population being from outside a given region. While the patterns we note in both strontium and carbon and nitrogen data from San Pedro de Atacama are not so pronounced, the data from Larache do add credibility to the idea that there was an interdigitation of the Atacama oases with neighboring areas long before the colonial incursion. Similarly, it supports the argument that important cultural interactions were not limited to a monolithic engagement with the Tiwanaku polity, but rather with numerous groups scattered across the highlands. If we were to consider the idea of interdigitation more systematically, we might be able to explore hypotheses grounded in dietary and morphological variability and diversity. This cursory consideration of our paleodietary data bear this out with evidence of variety internal to local diet, and more specifically, a varied diet at Larache.

In reference to our third structuring question, sex does not appear to have been a particularly salient driver of observed dietary differences among the Middle Period inhabitants of San Pedro de Atacama. This finding contradicts earlier studies that suggested important differences in access to resources according to sex during the Middle Period [43, 44]. The lack of observed sex-based differences is surprising given that reconstructions of human movement in this period are tied to interregional exchange driven by llama caravans which have been understood, based on ethnographic and ethnohistoric evidence, to be primarily led by male kin [141–143], which would lead one to predict that the diets of males should show greater variability (see below). The lack of definitive sex-based differences raises the possibility that diet on these excursions was closely related to diet in the oases, at least as visible isotopically. This is especially true given that most exchange seems to have been focused on other highland communities and not on the more dietarily variable coast.

Importantly, our data suggest that diet—or at least access to particular foodstuffs as we cannot explore differences in quantities of foods—was not necessarily inherently differentiated by sex, notwithstanding our understanding of the breadth of Andean prehistory, which suggests the importance of duality in Andean cosmology. Our data suggest that males and females were not counterpointed in terms of their access to dietary resources in these Atacama populations despite some established sex differences in other practices and material culture distributions [121, 144, 145]. We would, however, note that our results do not entirely preclude the possibility of sex-linked dietary differences on a broader scale, in that females show greater homogeneity of dietary residuals across different *ayllus*, with their male counterparts exhibiting greater variety and consequently a possible range of different dietary practices (see Fig 17). As concerns this, our analyses are noteworthy, in that only Larache has significant male-female differences in modeled dietary contributions (observed in three of four food groups). Given the foreign origins of many of that *ayllu*'s males, it is not clear whether the observed modeled

differences are a consequence of dietary practices tied to sex or they speak more clearly to distinctions based on place of origin.

## Conclusion

In closing, we return to Appadurai's [9] notion of gastro-politics, and ask whether any of these results could be taken as revealing the dietary dimensions of the construction or maintenance of meaningful institutional social difference within the Middle Period societies of San Pedro de Atacama. While we do not see any *de facto* evidence for complete dietary differentiation (as there is always overlap in consumption between/among individuals, *ayllus*, and time periods, and as isotopic analysis is not capable of pinpointing specifically different foods or preparations), there are broad, potentially status-linked, changes in diet over time, with foreignness (and perhaps male-ness) playing a role in determining dietary composition.

The three levels of analysis presented here contribute directly to the recent archaeological and bioarchaeological debates about social organization in San Pedro de Atacama populations during the Middle Period. On a larger level, the analyses across *ayllu* show a clear diachronic trend in consumption (more $C_4$/CAM plants and more terrestrial meat with time), suggesting a rather uniform increase in status-linked diet, if the literature regarding the sumptuary dimensions of these foods are to be believed. This pattern fits well with the traditional view of the Middle Period in San Pedro de Atacama as a time of increasing social complexity [19, 29, 68], highlighting the regional commonalities among the different *ayllus*. This temporal shift in diet militates against the existence of a form of zero-sum gastro-politics, in that all diets are improving, irrespective of the inequalities that exist within local societies.

The more focused levels of analysis presented here, however, demonstrate the complexities of trying to define the impact that the emergent local social complexity had on dietary composition. Across all the *ayllus*, the results do not support a clear pattern of sexual differentiation of diet composition, with a suggestion that male diet may have been more diverse, contrasting with some previous studies of specific cemeteries [43, 60, 61]. Nevertheless, there are exceptions to this on an *ayllu* by *ayllu* basis, which adds support to recent discussions about the importance of local identity between *ayllus* and larger regional heterogeneities [46, 117, 120, 121].

Along this line, in the males of Larache we see potential evidence for the intersection of a sex and origin-based difference in diet given that males, who were more often non-local in origin, consumed more $C_4$/CAM plants and terrestrial meat. Again, if the literature on these classes of foods is to be believed, the distinctiveness of the Larache male diet could be seen as a testament to their higher status/access to prestige resources. That these were non-local individuals, who presumably had more direct access to the long-distance exchange networks for which the Middle Period is known, speaks to their differentiation being a possible manifestation of a network based strategy (a la Blanton et al. [2]), whereby status gained through network connections beyond the local conferred status upon aspirant leaders, who signified/reinforced their positioning through the consumption of different kinds/combinations of food. The richness of the material assemblage from a number of Larache burials (which includes gold *keros* and other rarities), provides a confirmation of the uniqueness and prestige of this subset of individuals. This may stand as the clearest example of the kinds of "gastro-politics" we set out to identify.

## Supporting information

**S1 File. R Script for calculation of bivariate correlations when either/both variables have associated uncertainties.**
(R)

## Acknowledgments

This work would not have been possible without the continued support of the staff of the *Instituto de Arqueología y Antropología* and the *Museo Arqueológico Le Paige* of the Universidad Católica del Norte, in particular, M. Arturo Torres and Jimena Cruz. We thank Mariana Ugarte for the design and execution of Fig 1. Finally, the authors wish to acknowledge the efforts of our colleague Sarah Schrader as well as our thoughtful and hard-working student research assistants: Cameron Beason, Brianne Herrera, Rocío López-Barrales, Eva Mann, Evan Mann, Camila Morales-Zuñiga, and Erin K. Smith.

## Author Contributions

**Conceptualization:** William J. Pestle, Mark Hubbe, Christina Torres-Rouff, Gonzalo Pimentel.

**Data curation:** William J. Pestle, Mark Hubbe, Christina Torres-Rouff.

**Formal analysis:** William J. Pestle, Mark Hubbe, Christina Torres-Rouff.

**Funding acquisition:** William J. Pestle, Mark Hubbe, Christina Torres-Rouff, Gonzalo Pimentel.

**Investigation:** William J. Pestle, Mark Hubbe, Christina Torres-Rouff, Gonzalo Pimentel.

**Methodology:** William J. Pestle, Mark Hubbe, Christina Torres-Rouff.

**Project administration:** William J. Pestle, Mark Hubbe, Christina Torres-Rouff.

**Resources:** William J. Pestle, Mark Hubbe, Christina Torres-Rouff, Gonzalo Pimentel.

**Software:** William J. Pestle, Mark Hubbe.

**Supervision:** Christina Torres-Rouff.

**Validation:** William J. Pestle, Mark Hubbe, Christina Torres-Rouff.

**Visualization:** William J. Pestle, Mark Hubbe.

**Writing – original draft:** William J. Pestle, Mark Hubbe, Christina Torres-Rouff.

**Writing – review & editing:** William J. Pestle, Mark Hubbe, Christina Torres-Rouff, Gonzalo Pimentel.

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
