## [Decision Letter · Decision Letter 0]

8 Feb 2021

PONE-D-20-41004

Temporal, Spatial, and Gender-based Dietary Differences in Middle Period San Pedro de Atacama, Chile: A Model-based Approach.

PLOS ONE

Dear Dr. Pestle,

Thank you for submitting your manuscript to PLOS ONE. After careful consideration, we feel that it has merit but does not fully meet PLOS ONE’s publication criteria as it currently stands. Therefore, we invite you to submit a revised version of the manuscript that addresses the points raised during the review process.

The manuscript of Pestle and co-authors analyzed a rather large sample of human skeletal remains from the broad Middle Period (AD 500-1000) of the San Pedro de Atacama (Chile) oases. The main focus of the paper is to sort out a possible signal of social inequality through the changes in diet (access to food resources) across space, time and gender.

The two reviewers are positive about this manuscript and I share most of their opinion. Nevertheless, there are some important points, raised by the reviewers and by myself, to be clarified and the manuscript should be changed accordingly, before acceptance.

1. the manuscript is not formatted according to Plos One instructions, nor the text nor the reference section. Please, follow in detail the Plos One guidelines for authors.

2. There is a substantial lack of information about the archaeology of the graveyards. From which kind of funerary assemblages is the skeletal record coming from? What about the representativeness, in term of demographic profiles, of the single necropoles. Is the skeletal record a reasonable proxy of the ayllus society? Are the skeletal samples from single ayllus comparable? We perfectly know how the Osteological Paradox and the always non-random selection of the funerary record limit our biocultural reconstructions. The authors have to supply a concise but exhaustive description of the funerary contexts and if it is homogeneous through time and space. A better description of the "incredibly well preserved and contextualized human skeletons" should be made available for the reader.

3.Similarly,  the bioarchaeological record is not enough reported, and few lines of the basic methods used for sex and age at death determination should be provided. Are all the individuals fully adult ones? so subadults? If " we recently completed intensive bioarchaeological and biogeochemical analysis of a large set (over 600 individuals) of incredibly well preserved and contextualized human skeletons from the Middle Period of the San Pedro oases" what was the criterion used to select the 288 individuals?

4. The authors use the MDS for some data analysis, please specify more in detail which flavour and the R packege used.

5. " while inequality of some sort is inherent in all human societies", page 2, is a too deterministic and maybe not necessary sentence

6. page 32 " A sufficient number of sexed individuals were available from only four ayllus: Coyo, Larache, Quitor, and Solcor." then in page 32 "The final iteration of this sex-based analysis combined *ayllu *and sex, forming twelve groups..", then in figure 16 5 male groups and 6 female, same in figure 17. According to Table 1 the sexed individuals are

Conde Duque

7

Coyo

74

Larache

18

Quitor

31

Solcor

54

Tchecar

15

18 individuals from Larache aren’t much more than the 15 in Tchear. I suggest to fix this point and to use (if possible) all the subsample or to better justify the exclusions.

Finally, a table in the text summarizing the number of individuals by sex and age at death for the six sub-samples could  be useful.

We look forward to receiving your revised manuscript.

Kind regards,

Luca Bondioli, M.D.

Academic Editor

PLOS ONE

Journal Requirements:

2. Please ensure that you include a title page within your main document. You should list all authors and all affiliations as per our author instructions and clearly indicate the corresponding author.

4. Please include your tables as part of your main manuscript and remove the individual files. Please note that supplementary tables (should remain/ be uploaded) as separate "supporting information" files

5. We note that Figure 1 in your submission contains map/satellite images which may be copyrighted. All PLOS content is published under the Creative Commons Attribution License (CC BY 4.0), which means that the manuscript, images, and Supporting Information files will be freely available online, and any third party is permitted to access, download, copy, distribute, and use these materials in any way, even commercially, with proper attribution. For these reasons, we cannot publish previously copyrighted maps or satellite images created using proprietary data, such as Google software (Google Maps, Street View, and Earth). For more information, see our copyright guidelines: http://journals.plos.org/plosone/s/licenses-and-copyright.

(1) You may seek permission from the original copyright holder of Figure 1 to publish the content specifically under the CC BY 4.0 license. 

Reviewers' comments:

Reviewer's Responses to Questions

**Comments to the Author**

1. Is the manuscript technically sound, and do the data support the conclusions?

Reviewer #1: Yes

Reviewer #2: Yes

2. Has the statistical analysis been performed appropriately and rigorously? 

Reviewer #1: Yes

Reviewer #2: Yes

3. Have the authors made all data underlying the findings in their manuscript fully available?

Reviewer #1: Yes

Reviewer #2: Yes

4. Is the manuscript presented in an intelligible fashion and written in standard English?

Reviewer #1: Yes

Reviewer #2: Yes

5. Review Comments to the Author

Reviewer #1: Very well-written and analyzed paper with a lot of information packed into a small space. Great sample size and background. Very nice introduction and convincing narrative.

Here are the few questions that arose as I read through it:

1. The paper explores the existence of inequalities in San Pedro de Atacama during the Middle Period. Are scholars directly addressing this question by using other bioarchaeological methods?

2. Three main potential vectors of difference (time, context and sex) are explored. Why age is not explored as a potential vector of difference?

3. Model-based paleodietary reconstruction is an interesting approach, and can help address differences among and within populations. However, many archaeologists may be unaware of the complexities and pitfalls of stable isotope mixing models. I do not think it is outside of the scope of this research to briefly discuss the idea that mixing models in paleodietary reconstructions can hardly provide an exact calculation of food group contributions, despite their potential to provide new lines of enquiry.

Reviewer #2: This is an excellent study that expands significantly beyond the approach usually used in archaeology by using a large data set and by applying statistical analysis to the results. My main concern with this paper is that there is almost no mention of where the skeletal and dental material comes from. If these samples are from archaeological sites, please provide some information about them. It is also unclear how the samples have been classified into ayllu.

6. PLOS authors have the option to publish the peer review history of their article (what does this mean?). If published, this will include your full peer review and any attached files.

Reviewer #1: No

Reviewer #2: **Yes: **Ian Moffat

---

## [Author Response · Author response to Decision Letter 0]

16 Apr 2021

Please see responses to individual reviewer comments/suggestions below, in red.

PONE-D-20-41004

Temporal, Spatial, and Gender-based Dietary Differences in Middle Period San Pedro de Atacama, Chile: A Model-based Approach.

Done.

Done

Done

We will look into the possibility of depositing our laboratory protocols in protocols.io.

Journal Requirements:

Manuscript (text and references) has been reformatted to PLoS One style requirements.

2. Please ensure that you include a title page within your main document. You should list all authors and all affiliations as per our author instructions and clearly indicate the corresponding author.

Completed as requested.

Corresponding author (Pestle) noted.

4. Please include your tables as part of your main manuscript and remove the individual files. Please note that supplementary tables (should remain/ be uploaded) as separate "supporting information" files

All tables have been added as Microsoft Excel objects in the main manuscript.

 5. We note that Figure 1 in your submission contains map/satellite images which may be copyrighted. All PLOS content is published under the Creative Commons Attribution License (CC BY 4.0), which means that the manuscript, images, and Supporting Information files will be freely available online, and any third party is permitted to access, download, copy, distribute, and use these materials in any way, even commercially, with proper attribution. For these reasons, we cannot publish previously copyrighted maps or satellite images created using proprietary data, such as Google software (Google Maps, Street View, and Earth). For more information, see our copyright guidelines: http://journals.plos.org/plosone/s/licenses-and-copyright.

We have modified the Figure 1 basemap layer to use imagery from the USGS National Map Viewer, which is in the public domain.

1. the manuscript is not formatted according to Plos One instructions, nor the text nor the reference section. Please, follow in detail the Plos One guidelines for authors.

Manuscript (text and references) has been reformatted to PLoS One style requirements.

2. There is a substantial lack of information about the archaeology of the graveyards. From which kind of funerary assemblages is the skeletal record coming from? What about the representativeness, in term of demographic profiles, of the single necropoles. Is the skeletal record a reasonable proxy of the ayllus society? Are the skeletal samples from single ayllus comparable? We perfectly know how the Osteological Paradox and the always non-random selection of the funerary record limit our biocultural reconstructions. The authors have to supply a concise but exhaustive description of the funerary contexts and if it is homogeneous through time and space. A better description of the "incredibly well preserved and contextualized human skeletons" should be made available for the reader.

A new section (skeletal collections) has been added to Material and Methods to offer more information on the skeletal samples provenience and representativeness. 

3.Similarly, the bioarchaeological record is not enough reported, and few lines of the basic methods used for sex and age at death determination should be provided. Are all the individuals fully adult ones? so subadults? If " we recently completed intensive bioarchaeological and biogeochemical analysis of a large set (over 600 individuals) of incredibly well preserved and contextualized human skeletons from the Middle Period of the San Pedro oases" what was the criterion used to select the 288 individuals?

See the new section in Material and Methods for details on bioarchaeological methods, and the added information in the Sampling section that clarifies the selection criteria for isotopic samples.

4. The authors use the MDS for some data analysis, please specify more in detail which flavour and the R packege used.

Details were added to M&M about the MDS. 

5. " while inequality of some sort is inherent in all human societies", page 2, is a too deterministic and maybe not necessary sentence

This sentence has been modified to remove the phrase noted.

6. page 32 " A sufficient number of sexed individuals were available from only four ayllus: Coyo, Larache, Quitor, and Solcor." then in page 32 "The final iteration of this sex-based analysis combined ayllu and sex, forming twelve groups..", then in figure 16 5 male groups and 6 female, same in figure 17. According to Table 1 the sexed individuals are

Conde Duque 7

Coyo 74

Larache 18

Quitor 31

Solcor 54

Tchecar 15

 18 individuals from Larache aren’t much more than the 15 in Tchecar. I suggest to fix this point and to use (if possible) all the subsample or to better justify the exclusions.

We have attempted to address in the text both the reasons for the exclusion of Tchecar and our decision to justify the focus on Larache.

Finally, a table in the text summarizing the number of individuals by sex and age at death for the six sub-samples could be useful.

A new Table (Table 11) is now in the text with sex-by-ayllu data. Age data were not supplied for reasons discussed below (lack of subadults and imprecision of adult age estimates).

Reviewers' comments:

Reviewer #1: Very well-written and analyzed paper with a lot of information packed into a small space. Great sample size and background. Very nice introduction and convincing narrative.

Here are the few questions that arose as I read through it:

1. The paper explores the existence of inequalities in San Pedro de Atacama during the Middle Period. Are scholars directly addressing this question by using other bioarchaeological methods?

This is detailed in the new section “Skeletal Collections” in Materials and Methods.

2. Three main potential vectors of difference (time, context and sex) are explored. Why age is not explored as a potential vector of difference?

The reasons for omission of age as a vector or difference are noted in the Research questions stcion. The sample of 257 well-preserved individuals includes only 15 subadults (5.8%), and as many of the adult individuals are represented only by cranial remains, precise age determination was not possible. 

3. Model-based paleodietary reconstruction is an interesting approach, and can help address differences among and within populations. However, many archaeologists may be unaware of the complexities and pitfalls of stable isotope mixing models. I do not think it is outside of the scope of this research to briefly discuss the idea that mixing models in paleodietary reconstructions can hardly provide an exact calculation of food group contributions, despite their potential to provide new lines of enquiry.

A discussion of the limitations of model-based approaches to paleodietary estimation has been added as requested/.

Reviewer #2: This is an excellent study that expands significantly beyond the approach usually used in archaeology by using a large data set and by applying statistical analysis to the results. My main concern with this paper is that there is almost no mention of where the skeletal and dental material comes from. If these samples are from archaeological sites, please provide some information about them. It is also unclear how the samples have been classified into ayllu.

See new section on Skeletal Collections that offers the information requested.

---

## [Decision Letter · Decision Letter 1]

10 May 2021

Temporal, Spatial, and Gender-based Dietary Differences in Middle Period San Pedro de Atacama, Chile: A Model-based Approach.

PONE-D-20-41004R1

Dear Dr. Pestle,

We’re pleased to inform you that your manuscript has been judged scientifically suitable for publication and will be formally accepted for publication once it meets all outstanding technical requirements.

Kind regards,

Luca Bondioli, M.D.

Academic Editor

PLOS ONE

Additional Editor Comments (optional):

Reviewers' comments:

Reviewer's Responses to Questions

**Comments to the Author**

1. If the authors have adequately addressed your comments raised in a previous round of review and you feel that this manuscript is now acceptable for publication, you may indicate that here to bypass the “Comments to the Author” section, enter your conflict of interest statement in the “Confidential to Editor” section, and submit your "Accept" recommendation.

Reviewer #1: All comments have been addressed

Reviewer #2: All comments have been addressed

2. Is the manuscript technically sound, and do the data support the conclusions?

Reviewer #1: Yes

Reviewer #2: Yes

3. Has the statistical analysis been performed appropriately and rigorously? 

Reviewer #1: Yes

Reviewer #2: Yes

4. Have the authors made all data underlying the findings in their manuscript fully available?

Reviewer #1: Yes

Reviewer #2: Yes

5. Is the manuscript presented in an intelligible fashion and written in standard English?

Reviewer #1: Yes

Reviewer #2: Yes

6. Review Comments to the Author

Reviewer #1: The authors have well addressed my concerns and I think the manuscript is acceptable for publication

Reviewer #2: (No Response)

7. PLOS authors have the option to publish the peer review history of their article (what does this mean?). If published, this will include your full peer review and any attached files.

Reviewer #1: No

Reviewer #2: **Yes: **Ian Moffat

---

## [Editor Report · Acceptance letter]

12 May 2021

PONE-D-20-41004R1 

Temporal, Spatial, and Gender-based Dietary Differences in Middle Period San Pedro de Atacama, Chile: A Model-based Approach 

Dear Dr. Pestle:

I'm pleased to inform you that your manuscript has been deemed suitable for publication in PLOS ONE. Congratulations! Your manuscript is now with our production department. 

Kind regards, 

on behalf of

Dr. Luca Bondioli 

Academic Editor

PLOS ONE